# WFR-FM: Simulation-Free Dynamic Unbalanced Optimal Transport

**Qiangwei Peng**[1]∗, **Zihan Wang**[2]∗, **Junda Ying**[3]∗, **Yuhao Sun**[4]∗,
**Qing Nie**[5,6,7]†, **Lei Zhang**[2,3,4]†, **Tiejun Li**[1,4,8]† & **Peijie Zhou**[2,4,8,9]†

[1]LMAM and School of Mathematical Sciences, Peking University
[2]Center for Quantitative Biology, Peking University
[3]Beijing International Center for Mathematical Research, Peking University
[4]Center for Machine Learning Research, Peking University
[5]NSF-Simons Center for Multiscale Cell Fate Research, University of California, Irvine
[6]Department of Developmental and Cell Biology, University of California, Irvine
[7]Department of Mathematics, University of California, Irvine
[8]National Engineering Laboratory for Big Data Analysis and Applications, Beijing
[9]AI for Science Institute, Beijing

## Abstract

The Wasserstein–Fisher–Rao (WFR) metric extends dynamic optimal transport (OT) by coupling displacement with change of mass, providing a principled geometry for modeling unbalanced snapshot dynamics. Existing WFR solvers, however, are often unstable, computationally expensive, and difficult to scale. Here we introduce **WFR Flow Matching (WFR-FM)**, a simulation-free training algorithm that unifies flow matching with dynamic unbalanced OT. Unlike classical flow matching which regresses only a transport vector field, WFR-FM simultaneously regresses a vector field for displacement and a scalar growth rate function for birth–death dynamics, yielding continuous flows under the WFR geometry. Theoretically, we show that minimizing the WFR-FM loss exactly recovers WFR geodesics. Empirically, WFR-FM yields more accurate and robust trajectory inference in single-cell biology, reconstructing consistent dynamics with proliferation and apoptosis, estimating time-varying growth fields, and applying to generative dynamics under imbalanced data. It outperforms state-of-the-art baselines in efficiency, stability, and reconstruction accuracy. Overall, WFR-FM establishes a unified and efficient paradigm for learning dynamical systems from unbalanced snapshots, where not only states but also mass evolve over time. The Python code is available at `https://github.com/QiangweiPeng/WFR-FM`.

## 1 Introduction

Reconstructing continuous dynamics from limited observations is a central challenge in many scientific domains (Chen et al., 2018). In single-cell transcriptomics, destructive sequencing protocols and experimental costs typically restrict measurements to a small number of temporal snapshots (Zheng et al., 2017; Chen et al., 2022a). The problem of trajectory inference is therefore to recover the underlying cellular dynamics from such sparse single-cell RNA sequencing (scRNA-seq) snapshots (Schiebinger et al., 2019; Tong et al., 2020). Optimal transport (OT) has then emerged as a powerful framework for modeling single-cell population across time-series scRNA-seq data (Bunne et al., 2024; Zhang et al., 2025b). Existing approaches can be broadly divided into two categories: methods based on static OT maps (Schiebinger et al., 2019; Klein et al., 2025; Halmos et al., 2025), which align distributions at different time points without explicitly modeling intermediate dynamics, and methods based on dynamic OT (Tong et al., 2020; Huguet et al., 2022; Zhang et al., 2024a;

---

∗Equal contribution
†Corresponding authors: qnie@uci.edu, zhangl@math.pku.edu.cn, {tieli, pjzhou}@pku.edu.cn

Peng et al., 2024; Tong et al., 2023; Gu et al., 2025), which reconstruct continuous flows between snapshots. While dynamic OT methods provide richer dynamical insights, they are often realized via neural ordinary differential equations (ODEs) and continuous normalizing flows, leading to high computational cost and training instabilities due to repeated ODE simulations (Yan et al., 2019).

To accelerate learning of continuous normalizing flows, flow matching (FM) (Lipman et al., 2023; Liu et al., 2022; Pooladian et al., 2023; Albergo & Vanden-Eijnden, 2022) has been proposed as a simulation-free training paradigm. By directly regressing the drift field of an ODE, FM enables stable and scalable training without the need for integration. This motivated several works combining OT with FM, which approximate dynamic OT in a simulation-free manner (Tong et al., 2024a; Klein et al., 2024; Rohbeck et al., 2025; Eyring et al., 2024). Such approaches enable the efficient recovery of transport-based dynamics, avoiding the prohibitive cost of repeated ODE solvers.

However, single-cell dynamics are not mass-conserving: cells can proliferate and undergo apoptosis, resulting in unbalanced distributions across time (Sha et al., 2024). The Wasserstein–Fisher–Rao (WFR) metric (Liero et al., 2016; Chizat et al., 2018a; Kondratyev et al., 2016) extends dynamic OT by coupling transport with mass creation and annihilation, providing a principled geometry for such unbalanced dynamics (Sha et al., 2024; Peng et al., 2024; Zhang et al., 2025a; Sun et al., 2025). This has inspired several extensions of FM to unbalanced settings (Eyring et al., 2024; Cao et al., 2025; Corso et al., 2025; Wang et al., 2025). However, existing approaches typically focus on regressing only velocity fields and neglecting explicit modeling of growth dynamics (Eyring et al., 2024; Corso et al., 2025; Cao et al., 2025). The recent VGFM (Wang et al., 2025) introduces an innovative flow matching framework that jointly learns both transport and growth. Its formulation separates birth–death dynamics from transport and then uses modified dynamics to approximate simultaneous evolution, which may depart from the geometry of dynamic unbalanced OT. In addition, while being effective, VGFM still relies on ODE simulation during post-training refinement and therefore is not fully simulation-free. Action Matching (AM) (Neklyudov et al., 2023) proposes a novel simulation-free approach that can handle unbalanced dynamics. However, it assumes the access to a continuous curve of densities, which might be challenging in scRNA-seq experiments. The Wasserstein Lagrangian Flows (WLF) framework (Neklyudov et al., 2024) addresses the issue with an inner-outer optimization design, albeit at the cost of introducing additional computational costs.

To address these challenges, we introduce WFR-FM, which combines the core ideas of unbalanced OT and FM to solve dynamic unbalanced OT under the WFR geometry in a simulation-free manner. WFR-FM jointly regresses both a transport vector field and a growth rate function, yielding continuous flows that explicitly model displacement and birth–death dynamics. We show that this approach produces more accurate WFR-OT flows than existing ODE-based solvers Sha et al. (2024); Zhang et al. (2025a); Huguet et al. (2022); Sun et al. (2025), balanced-distribution flow matching methods Tong et al. (2024b); Kapusniak et al. (2024); Rohbeck et al. (2025), and previous unbalanced FM variants Eyring et al. (2024); Wang et al. (2025). Our contributions can be summarized as follows:

- We propose WFR-FM, a novel framework that extends flow matching to unbalanced distributions by jointly regressing transport velocity fields and growth rates. WFR-FM is entirely simulation-free, eliminating costly ODE integration and providing a scalable, robust, and OT-principled method for modeling unbalanced dynamics of population snapshots.

- We establish theoretical guarantees for WFR-FM: minimizing its loss exactly recovers dynamic unbalanced optimal transport under the WFR metric, i.e. the constructed cell population trajectories follow WFR geodesics.

- We apply WFR-FM to handle multiple time points, making it particularly suitable for single-cell transcriptomics. Empirically, this yields more efficient and robust trajectory inference compared to existing neural differential equation-based and OT-based baselines.

## 2 RELATED WORKS

**Unbalanced Optimal Transport and WFR Distance.** Unbalanced optimal transport extends classical OT Kantorovich (1942); Benamou & Brenier (2000) by either penalizing deviations from marginals (Benamou, 2003; Figalli, 2010; Caffarelli & McCann, 2010) or in its dynamic form by adding a penalty on source terms. A widely used variant is the Wasserstein–Fisher–Rao (WFR) distance Liero et al. (2016); Chizat et al. (2018a); Kondratyev et al. (2016), which penalizes the

squared growth rates and admits closed-form solutions in special cases Chizat et al. (2018a), with dynamic and static formulations shown to be equivalent (Chizat et al., 2018b; Liero et al., 2018). Entropic regularization enables efficient solvers for static problem (Liero et al., 2018; Chapel et al., 2021). Clancy & Suarez (2022) studied interpolating splines on the WFR space. Recently, deep learning approaches to dynamic WFR have been applied to single-cell dynamics (Sha et al., 2024; Peng et al., 2024; Zhang et al., 2025a; Sun et al., 2025), but they depend on costly ODE simulation. Action Matching (Neklyudov et al., 2023) and WLF (Neklyudov et al., 2024) propose the non-flow-matching simulation-free method for WFR problem. Here WFR-FM introduces the **simulation-free flow-matching-based WFR solver** that eliminates expensive trajectory integration in training.

**Flow Matching and Unbalanced Extensions.** Flow matching (Lipman et al., 2023; Liu et al., 2022; Pooladian et al., 2023; Albergo & Vanden-Eijnden, 2022) is a simulation-free framework for learning velocity fields via conditional probability paths in generative models, avoiding explicit trajectory simulation. Subsequent work has expanded with OT coupling (Tong et al., 2024a; Klein et al., 2024), conditional generation (Rohbeck et al., 2025), multi-marginal formulations (Rohbeck et al., 2025; Lee et al., 2025), stochastic dynamics (Tong et al., 2024b; Lee et al., 2025), and other generalizations (Kapusniak et al., 2024; Zhang et al., 2024b; Atanackovic et al., 2025; Petrović et al., 2025), yet most assume **normalized distributions**, thereby conserving mass. Motivated by real-world applications such as cell differentiation with proliferation and apoptosis, unbalanced flow matching has recently emerged (Eyring et al., 2024; Cao et al., 2025; Corso et al., 2025; Wang et al., 2025), but existing approaches either regress only velocity fields (Eyring et al., 2024; Corso et al., 2025; Cao et al., 2025) or, when jointly modeling velocity and growth (Wang et al., 2025), still rely on ODE simulation for post-training and the coupling mechanisms are not consistent with unbalanced optimal transport. WFR-FM overcomes these limitations by providing an **OT-principled formulation of unbalanced flow matching**, which recovers the geodesics of the WFR metric.

**Single-cell Trajectory Inference.** Many algorithms have been proposed for single-cell trajectory inference in the multiple-time-points setup, including static optimal transport maps Schiebinger et al. (2019); Klein et al. (2025); Halmos et al. (2025); Banerjee et al. (2024), continuous formulations via neural ODEs Tong et al. (2020); Huguet et al. (2022); Zhang et al. (2024a); Sha et al. (2024); Tong et al. (2023); Gu et al. (2025); Choi & Choi (2024), and stochastic variants Neklyudov et al. (2023; 2024); Albergo et al. (2023); Maddu et al. (2024); Yeo et al. (2021); Chizat et al. (2022); Ventre et al. (2023); Lavenant et al. (2024); Shi et al. (2024); Koshizuka & Sato (2023); Bunne et al. (2023); Chen et al. (2022b); Jiang & Wan (2024); Zhang et al. (2025a); Sun et al. (2025). More recently, flow matching has been applied in this domain to improve scalability and training stability (Tong et al., 2024a; Rohbeck et al., 2025; Klein et al., 2024; Tong et al., 2024b; Lee et al., 2025; Kapusniak et al., 2024; Atanackovic et al., 2025). The biological importance of proliferation and apoptosis has further motivated extensions to unbalanced settings (Neklyudov et al., 2023; 2024; Eyring et al., 2024; Wang et al., 2025). Notably, VGFM (Wang et al., 2025) explicitly incorporating growth estimation into the flow-matching framework. However, existing unbalanced approaches either rely on static approximations or require costly ODE simulations, or they lack a truly simulation-free flow matching framework that can **jointly recover velocity and growth** in single-cell dynamics.

## 3 MATHEMATICAL BACKGROUND

**Setup.** Consider the distribution density function of two measures $\mu_0(\boldsymbol{x})$ and $\mu_1(\boldsymbol{x})$ (also denoted $\mu_0, \mu_1$) at times $t = 0$ and $t = 1$ respectively defined over $\mathcal{X} \subseteq \mathbb{R}^d$. Let $\mathcal{M}_+(\mathcal{X})$ represent the set of all absolutely continuous finite measures support on $\mathcal{X}$. The total mass of $\mu_0$ and $\mu_1$ may be different. We aim to learn a dynamical WFR-OT path connecting $\mu_0$ and $\mu_1$ (defined below) via a simulation-free framework. In this section and Section 4, we focus on the case of two measures. In Section 5, we will turn to the dynamic WFR problem with observation at multiple time points.

**Dynamic Optimal Transport via Flow Matching.** Dynamic OT provides a principled framework for interpolating between probability distributions by minimizing the kinetic energy of transport paths (Benamou & Brenier, 2000). In this setting, the transport is represented by a time-dependent probability path $\{\rho_t\}_{t \in [0,1]}$ and a velocity field $\boldsymbol{u}_t$, which jointly satisfy the continuity equation $\partial_t \rho_t + \nabla \cdot (\rho_t \boldsymbol{u}_t) = 0$. This dynamic formulation yields both natural interpolation between distributions and a notion of geodesic distance. Recent work (Tong et al., 2024a) connects this formulation to FM, which learns a parameterized velocity field $\boldsymbol{v}_\theta(t, \boldsymbol{x})$ to approximate the OT flow. The training

objective is a simple regression against the target field,

$$\mathcal{L}_{\mathrm{FM}}(\theta) = \mathbb{E}_{t,\boldsymbol{x}\sim\rho_t}\big[\|\boldsymbol{v}_\theta(t,\boldsymbol{x}) - \boldsymbol{u}_t(\boldsymbol{x})\|_2^2\big],$$

avoiding costly backpropagation through ODE solvers. A key ingredient is the use of conditional probability paths. Instead of directly constructing the marginal path $\rho_t$, one defines conditionals $\rho_t(\cdot|\boldsymbol{z})$ with tractable vector fields $\boldsymbol{u}_t(\cdot|\boldsymbol{z})$. For example, given endpoint samples $(\boldsymbol{x}_0,\boldsymbol{x}_1)$, a Gaussian bridge $\rho_t(\boldsymbol{x}|\boldsymbol{x}_0,\boldsymbol{x}_1)$ yields a closed-form vector field, and the marginal regression loss becomes

$$\mathcal{L}_{\mathrm{CFM}}(\theta) = \mathbb{E}_{t,\boldsymbol{z},\boldsymbol{x}\sim\rho_t(\cdot|\boldsymbol{z})}\big[\|\boldsymbol{v}_\theta(t,\boldsymbol{x}) - \boldsymbol{u}_t(\boldsymbol{x}|\boldsymbol{z})\|_2^2\big].$$

When conditioning pairs $(\boldsymbol{x}_0,\boldsymbol{x}_1)$ are drawn from an OT coupling $\gamma(\boldsymbol{x}_0,\boldsymbol{x}_1)$, the resulting objective recovers the dynamic OT flow. Thus, the **conditional path**, **conditional loss**, and **coupling** (i.e. conditions) designs jointly determine if learned dynamics match OT geodesics or general bridges.

**Continuous measure flows.** Consider a time-dependent measure path $\rho : [0,1]\times\mathbb{R}^d \to \mathbb{R}_+$, a time dependent vector field $\boldsymbol{u} : [0,1]\times\mathbb{R}^d \to \mathbb{R}^d$, and a time dependent rate function $g : [0,1]\times\mathbb{R}^d \to \mathbb{R}$. Assuming enough regularity, this vector field and rate function defines a unique time dependent flow $\boldsymbol{\phi} : [0,1]\times\mathbb{R}^d \to \mathbb{R}^d$, $m : [0,1]\times\mathbb{R}^d \to \mathbb{R}$ defined via an ODE:

$$\frac{d}{dt}\begin{pmatrix}\boldsymbol{\phi}_t(\boldsymbol{x})\\ \ln m_t(\boldsymbol{x})\end{pmatrix} = \begin{pmatrix}\boldsymbol{u}_t(\boldsymbol{\phi}_t(\boldsymbol{x}))\\ g_t(\boldsymbol{\phi}_t(\boldsymbol{x}))\end{pmatrix}, \quad \boldsymbol{\phi}_0(\boldsymbol{x}) = \boldsymbol{x}, \;\; m_0(\boldsymbol{x}) \text{ given.} \tag{3.1}$$

which defines a 'push-forward' mapping with changing mass $(\boldsymbol{\phi}_t, m_t)_\# : \mathcal{M}_+(\mathbb{R}^d) \to \mathcal{M}_+(\mathbb{R}^d)$ for each $t \in [0,1]$ such that

$$\rho_t(\boldsymbol{x}) = (\boldsymbol{\phi}_t, m_t)_\#\rho_0(\boldsymbol{x}) := \rho_0(\boldsymbol{\phi}_t^{-1}(\boldsymbol{x}))\,\big|\det\nabla_{\boldsymbol{x}}\boldsymbol{\phi}_t^{-1}(\boldsymbol{x})\big|\,\frac{m_t(\boldsymbol{x})}{m_0(\boldsymbol{\phi}_t^{-1}(\boldsymbol{x}))}$$

and satisfies the continuity equation with source term

$$\partial_t\rho_t(\boldsymbol{x}) + \nabla_{\boldsymbol{x}}\cdot(\boldsymbol{u}_t(\boldsymbol{x})\rho_t(\boldsymbol{x})) = g_t(\boldsymbol{x})\rho_t(\boldsymbol{x}) \tag{3.2}$$

**WFR Optimal transport.** The dynamical unbalanced OT described by WFR metric (Chizat et al., 2018a;b; Liero et al., 2018) is defined as

$$\mathrm{WFR}_\delta^2(\mu_0,\mu_1) = \inf_{\rho,g,\boldsymbol{u}}\int_0^1\int_{\mathcal{X}}\frac{1}{2}(\|\boldsymbol{u}(\boldsymbol{x},t)\|_2^2 + \delta^2\|g(\boldsymbol{x},t)\|_2^2)\rho_t(\boldsymbol{x})\mathrm{d}\boldsymbol{x}\mathrm{d}t \tag{3.3}$$
$$\text{s.t.} \qquad \partial_t\rho + \nabla_{\boldsymbol{x}}\cdot(\rho\boldsymbol{u}) = \rho g, \; \rho_0 = \mu_0, \; \rho_1 = \mu_1,$$

where $\delta$ is a hyperparameter in the WFR metric that balances the contributions of transport and birth–death (or mass creation/annihilation) terms in the action. To design the conditional path of flow matching for WFR, it is essential to study the geodesics between two mass points under such a metric. Indeed, the closed-form WFR distance between two Dirac measures $m_0\delta_{\boldsymbol{x}_0}$ and $m_1\delta_{\boldsymbol{x}_1}$ is given by Chizat et al. (2018a); Liero et al. (2018)

$$\mathrm{WFR\text{-}DD}_\delta^2(m_0\delta_{\boldsymbol{x}_0}, m_1\delta_{\boldsymbol{x}_1}) = 2\delta^2\big(m_0 + m_1 - 2\sqrt{m_0 m_1}\,\overline{\cos}(\frac{\|\boldsymbol{x}_0 - \boldsymbol{x}_1\|_2}{2\delta})\big) \tag{3.4}$$

where $\overline{\cos}(\boldsymbol{x}) = \cos(\min\{\boldsymbol{x}, \frac{\pi}{2}\})$. When $\|\boldsymbol{x}_0 - \boldsymbol{x}_1\|_2 < \pi\delta$, the corresponding geodesic curve i.e. optimal transport path, named **traveling Dirac** is described by

$$m(t) = At^2 - 2Bt + m_0, \quad \boldsymbol{u}(t)m(t) = \boldsymbol{\omega}_0 \tag{3.5}$$

where $A, B, \boldsymbol{\omega}_0$ can be computed as follows

$$\begin{cases} A = m_0 + m_1 - 2\sqrt{\dfrac{m_0 m_1}{1+\tau^2}} \;\;,\;\; B = m_0 - \sqrt{\dfrac{m_0 m_1}{1+\tau^2}} \\[3mm] \boldsymbol{\omega}_0 = 2\delta\tau\sqrt{\dfrac{m_0 m_1}{1+\tau^2}}\,\boldsymbol{l} \;\;,\;\; \tau = \tan(\dfrac{\|\boldsymbol{x}_1 - \boldsymbol{x}_0\|_2}{2\delta}) \end{cases} \tag{3.6}$$

where $\boldsymbol{l}$ is the unit vector pointing from $\boldsymbol{x}_0$ to $\boldsymbol{x}_1$.

To construct the coupling in flow matching training, we also refer to the equivalent static Kantorovich form of WFR problem, which is also studied by Chizat et al. (2018b); Liero et al. (2018)

$$\text{WFR}_\delta^2(\mu_0, \mu_1) = \inf_{(\gamma_0, \gamma_1) \in (\mathcal{M}_+(\mathcal{X}^2))^2} \int_{\mathcal{X}^2} \text{WFR-DD}_\delta^2(\gamma_0(\boldsymbol{x}, \boldsymbol{y})\delta_{\boldsymbol{x}}, \gamma_1(\boldsymbol{x}, \boldsymbol{y})\delta_{\boldsymbol{y}})\mathrm{d}\boldsymbol{x}\mathrm{d}\boldsymbol{y}, \quad (3.7)$$

where $(\gamma_0, \gamma_1)$ is called semi-coupling, and it satisfies the constraints

$$\Gamma(\mu_0, \mu_1) \stackrel{\text{def.}}{=} \left\{ (\gamma_0, \gamma_1) \in \left(\mathcal{M}_+(\mathcal{X}^2)\right)^2 : \int_{\mathcal{X}} \gamma_0(\boldsymbol{x}, \boldsymbol{y})\mathrm{d}\boldsymbol{y} = \mu_0(\boldsymbol{x}), \int_{\mathcal{X}} \gamma_1(\boldsymbol{x}, \boldsymbol{y})\mathrm{d}\boldsymbol{x} = \mu_1(\boldsymbol{y}) \right\}.$$

For fixed pair $(\boldsymbol{x}, \boldsymbol{y})$, $\gamma_0(\boldsymbol{x}, \boldsymbol{y})$ represents the mass at the initial time sent from $\boldsymbol{x}$, while $\gamma_1(\boldsymbol{x}, \boldsymbol{y})$ represents the corresponding mass at the final time received by $\boldsymbol{y}$. Since the transport is unbalanced, the mass sent from $\boldsymbol{x}$ may not be equal to the mass received by $\boldsymbol{y}$. To efficiently solve such WFR coupling, it is proved that the WFR problem is also equivalent to an optimal entropy-transport (OET) problem (Chizat et al., 2018b; Liero et al., 2018)

$$\begin{aligned}
\text{WFR}_\delta^2(\mu_0, \mu_1) = 2\delta^2 \inf_{\gamma \in \mathcal{M}_+(\mathcal{X}^2)} \Big\{ &\int_{\mathcal{X}^2} -2\ln\overline{\cos}(\frac{\|\boldsymbol{x} - \boldsymbol{y}\|_2}{2\delta})\gamma(\boldsymbol{x}, \boldsymbol{y})\mathrm{d}\boldsymbol{x}\mathrm{d}\boldsymbol{y} \\
&+ \text{KL}(\int_{\mathcal{X}} \gamma(\boldsymbol{x}, \boldsymbol{y})\mathrm{d}\boldsymbol{y}\|\mu_0(\boldsymbol{x})) + \text{KL}(\int_{\mathcal{X}} \gamma(\boldsymbol{x}, \boldsymbol{y})\mathrm{d}\boldsymbol{x}\|\mu_1(\boldsymbol{y})) \Big\}
\end{aligned} \quad (3.8)$$

The relation between $(\gamma_0, \gamma_1)$ and $\gamma$ has been studied previously under general condition (Liero et al., 2018). In this work, we also give a straightforward proof (see Appendix A.1) for the WFR problem that the optimal semi-coupling can be constructed from the OET coupling $\gamma$ as following.

**Theorem 3.1.** *Let $\gamma$ be the optimal coupling of the OET problem (3.8), then the semi-coupling $\gamma_0(\boldsymbol{x}, \boldsymbol{y}) = \frac{\gamma(\boldsymbol{x}, \boldsymbol{y})}{\int_{\mathcal{X}} \gamma(\boldsymbol{x}, \boldsymbol{z})\mathrm{d}\boldsymbol{z}}\mu_0(\boldsymbol{x}), \gamma_1(\boldsymbol{x}, \boldsymbol{y}) = \frac{\gamma(\boldsymbol{x}, \boldsymbol{y})}{\int_{\mathcal{X}} \gamma(\boldsymbol{z}, \boldsymbol{y})\mathrm{d}\boldsymbol{z}}\mu_1(\boldsymbol{y})$ solves the static WFR problem (3.7).*

# 4 SIMULATION-FREE TRAINING FOR WFR PROBLEM

Given a vector field $\boldsymbol{u}_t$ and a rate function $g_t$ that generate the measure path $\rho_t$ by (3.2), we aim to learn a time-dependent vector field $\boldsymbol{v}_{\boldsymbol{\theta}}(\boldsymbol{x}, t)$ and a time-dependent growth rate function $g_{\boldsymbol{\phi}}(\boldsymbol{x}, t)$ both parametrized by neural networks, with loss function specified by minimizing the **intractable** unbalanced flow matching objective

$$\mathcal{L}_{\text{UFM}}(\boldsymbol{\theta}, \boldsymbol{\phi}) = \int_0^1 \int_{\mathcal{X}} (\|\boldsymbol{v}_{\boldsymbol{\theta}}(\boldsymbol{x}, t) - \boldsymbol{u}_t(\boldsymbol{x})\|_2^2 + \kappa\|g_{\boldsymbol{\phi}}(\boldsymbol{x}, t) - g_t(\boldsymbol{x})\|_2^2)\rho_t(\boldsymbol{x})\mathrm{d}\boldsymbol{x}\mathrm{d}t \quad (4.1)$$

Next, we follow the idea of conditional flow matching (CFM) Lipman et al. (2023) to learn $\boldsymbol{u}_t(\boldsymbol{x})$ and $g_t(\boldsymbol{x})$ from conditional path, conditional loss and coupling guided by the WFR geometry.

## 4.1 CONDITIONAL PATH CONSTRUCTION

**Conditional measure path** Following the approach of defining a conditional probability path in analogy to flow matching (Lipman et al., 2023), we define a conditional measure path w.r.t the condition variable $\boldsymbol{z}$ such that $\rho_t(\boldsymbol{x}|\boldsymbol{z}) = m_t(\boldsymbol{z})\tilde{\rho}_t(\boldsymbol{x}|\boldsymbol{z})$ where the time-dependent conditional measure $\rho_t(\boldsymbol{x}|\boldsymbol{z})$ is decoupled into a time dependent mass $m_t(\boldsymbol{z})$ and a time-dependent conditional probability density $\tilde{\rho}_t(\boldsymbol{x}|\boldsymbol{z})$. The $\boldsymbol{u}_t(\boldsymbol{x}|\boldsymbol{z})$ and $g_t(\boldsymbol{x}|\boldsymbol{z})$ are conditional vector field and conditional rate function generating $\rho_t(\boldsymbol{x}|\boldsymbol{z})$ from $\rho_0(\boldsymbol{x}|\boldsymbol{z})$, i.e.

$$\partial_t \rho_t(\boldsymbol{x}|\boldsymbol{z}) + \nabla_{\boldsymbol{x}} \cdot (\boldsymbol{u}_t(\boldsymbol{x}|\boldsymbol{z})\rho_t(\boldsymbol{x}|\boldsymbol{z})) = \rho_t(\boldsymbol{x}|\boldsymbol{z})g_t(\boldsymbol{x}|\boldsymbol{z}) \quad (4.2)$$

**Marginal measure path** Assuming $\boldsymbol{z} \sim q(\boldsymbol{z})$, we define the marginal measure path from the conditional measure path

$$\rho_t(\boldsymbol{x}) = \int \rho_t(\boldsymbol{x}|\boldsymbol{z})q(\boldsymbol{z})\mathrm{d}\boldsymbol{z} \quad (4.3)$$

as well as the marginal vector field and rate function

$$\boldsymbol{u}_t(\boldsymbol{x}) = \int \boldsymbol{u}_t(\boldsymbol{x}|\boldsymbol{z})\frac{\rho_t(\boldsymbol{x}|\boldsymbol{z})q(\boldsymbol{z})}{\rho_t(\boldsymbol{x})}\mathrm{d}\boldsymbol{z}, \quad g_t(\boldsymbol{x}) = \int g_t(\boldsymbol{x}|\boldsymbol{z})\frac{\rho_t(\boldsymbol{x}|\boldsymbol{z})q(\boldsymbol{z})}{\rho_t(\boldsymbol{x})}\mathrm{d}\boldsymbol{z} \quad (4.4)$$

The following theorem sketches the relation between marginals $\boldsymbol{u}_t(\boldsymbol{x})$, $g_t(\boldsymbol{x})$ and conditionals $\boldsymbol{u}_t(\boldsymbol{x}|\boldsymbol{z})$ and $g_t(\boldsymbol{x}|\boldsymbol{z})$, with the proof present in Appendix A.2.

**Theorem 4.1.** *The marginal vector field and rate function (4.4) generates the marginal measure path (4.3) from $p_0(\boldsymbol{x})$ for any $q(\boldsymbol{z})$ independent of $\boldsymbol{x}$ and $t$.*

**Conditional Gaussian measure path** As a convenient choice, we introduce the conditional Gaussian measure path (CGMP)

$$\tilde{\rho}_t(\boldsymbol{x}|\boldsymbol{z}) = \mathcal{N}(\boldsymbol{x}|\boldsymbol{\eta}_t(\boldsymbol{z}), \boldsymbol{\sigma}_t^2(\boldsymbol{z})), \quad \rho_t(\boldsymbol{x}|\boldsymbol{z}) = m_t(\boldsymbol{z})\tilde{\rho}_t(\boldsymbol{x}|\boldsymbol{z}) \tag{4.5}$$

The conditional vector field and the rate function can be easily computed as follows.

**Proposition 4.1.** *For CGMP defined above (4.5), the conditional vector field and growth rate function are*

$$\boldsymbol{u}_t(\boldsymbol{x}|\boldsymbol{z}) = \frac{\boldsymbol{\sigma}_t'(\boldsymbol{z})}{\boldsymbol{\sigma}_t(\boldsymbol{z})}\big(\boldsymbol{x} - \boldsymbol{\eta}_t(\boldsymbol{z})\big) + \boldsymbol{\eta}_t'(\boldsymbol{z}), \quad g_t(\boldsymbol{x}|\boldsymbol{z}) = \partial_t \ln m_t(\boldsymbol{z}) \tag{4.6}$$

The proof is left to Appendix A.4. Note that the conditional vector field shares the same form as the conditional Gaussian path used in balanced FM (Lipman et al., 2023). This allows our algorithm to naturally include flow matching as a special case by setting $m_t(\boldsymbol{z}) \equiv 1$.

## 4.2 CONDITIONAL LOSS DESIGN

With designed $\boldsymbol{u}_t(\boldsymbol{x}|\boldsymbol{z})$ and $g_t(\boldsymbol{x}|\boldsymbol{z})$, we propose to regress them by minimizing the **conditional unbalanced flow matching (CUFM) objective**

$$\mathcal{L}_{\mathrm{CUFM}}(\boldsymbol{\theta}, \boldsymbol{\phi}) = \mathbb{E}_{t \sim \mathcal{U}[0,1], \boldsymbol{z} \sim q(\boldsymbol{z}), \boldsymbol{x} \sim \tilde{\rho}_t(\boldsymbol{x}|\boldsymbol{z})}(\|\boldsymbol{v}_{\boldsymbol{\theta}} - \boldsymbol{u}_t(\boldsymbol{x}|\boldsymbol{z})\|_2^2 + \kappa \|g_{\boldsymbol{\phi}} - g_t(\boldsymbol{x}|\boldsymbol{z})\|_2^2)m_t(\boldsymbol{z}) \tag{4.7}$$

The crucial difference between the balanced CFM objective (Lipman et al., 2023) and CUFM objective (4.7) is the time dependent mass term $m_t(\boldsymbol{z})$. In the traditional FM, we only simulate the position of particles via the learned flow, while in unbalanced FM, the mass of the particle varies depending on time. The idea here is to weight the regression error by mass, which is assumed to be uniform in balanced setups. When $g_t(\boldsymbol{x}) \equiv 0$, the UFM objective (4.1) and CUFM (4.7) objective reduce to the well-known FM objective and CFM objective (Lipman et al., 2023).

The following theorem claims that the gradient of UFM and CUFM objective w.r.t parameters $(\boldsymbol{\theta}, \boldsymbol{\phi})$ are equal, justifying the designed conditional loss function, with the proof present in Appendix A.3.

**Theorem 4.2.** *If $\rho_t(\boldsymbol{x}) > 0$ for all $\boldsymbol{x} \in \mathcal{X}$ and $t \in [0, 1]$, and $q(\boldsymbol{z})$ is independent of $\boldsymbol{x}$ and $t$, then $\mathcal{L}_{\mathrm{UFM}}(\boldsymbol{\theta}, \boldsymbol{\phi}) = \mathcal{L}_{\mathrm{CUFM}}(\boldsymbol{\theta}, \boldsymbol{\phi}) + C$, where $C$ is independent to $(\boldsymbol{\theta}, \boldsymbol{\phi})$. Thus they have identical gradients w.r.t $(\boldsymbol{\theta}, \boldsymbol{\phi})$, i.e.*

$$\nabla_{\boldsymbol{\theta}, \boldsymbol{\phi}}\mathcal{L}_{\mathrm{UFM}}(\boldsymbol{\theta}, \boldsymbol{\phi}) = \nabla_{\boldsymbol{\theta}, \boldsymbol{\phi}}\mathcal{L}_{\mathrm{CUFM}}(\boldsymbol{\theta}, \boldsymbol{\phi}).$$

## 4.3 COUPLING IN WFR-FM SOLVES THE DYNAMICAL WFR PROBLEM

Now, we consider two measure $\mu_0(\boldsymbol{x})$ and $\mu_1(\boldsymbol{x})$ as source and target of a WFR problem, and the semi-coupling of the static form denoted as $(\gamma_0, \gamma_1)$. Without loss of generality, let $\mu_0(\boldsymbol{x})$ be a probability density on $\mathcal{X}$, then $\gamma_0(\boldsymbol{x}_0, \boldsymbol{x}_1)$ is a probability density on $\mathcal{X}^2$. The conditional variable $\boldsymbol{z} = (\boldsymbol{x}_0, \boldsymbol{x}_1) \sim \gamma_0(\boldsymbol{x}_0, \boldsymbol{x}_1)$, represents the position of the source and the target points. Since the solution of dynamical WFR of two Dirac is traveling Dirac Chizat et al. (2018a), we choose a special CGMP named **traveling Gaussian**

$$\tilde{\rho}_t(\boldsymbol{x}|\boldsymbol{x}_0, \boldsymbol{x}_1) = \mathcal{N}(\boldsymbol{x}|\boldsymbol{x}_0 + \boldsymbol{\omega}_0 \int_0^t \frac{\mathrm{d}s}{m_s(\boldsymbol{x}_0, \boldsymbol{x}_1)}, \sigma^2\mathbf{I}) := \mathcal{N}(\boldsymbol{x}|\boldsymbol{x}_0 + \boldsymbol{\omega}_0 \Lambda_t(\boldsymbol{x}_0, \boldsymbol{x}_1), \sigma^2\mathbf{I})$$

$$= \mathcal{N}(\boldsymbol{x}|\boldsymbol{x}_0 + \frac{\boldsymbol{\omega}_0}{\sqrt{m_0 A - B^2}}(\arctan(\frac{At - B}{\sqrt{m_0 A - B^2}}) - \arctan(\frac{-B}{\sqrt{m_0 A - B^2}})), \sigma^2\mathbf{I}). \tag{4.8}$$

where $\boldsymbol{\omega}_0$ and $m_t(\boldsymbol{x}_0, \boldsymbol{x}_1)$ are defined as (3.5) and (3.6) with boundary condition $m_0(\boldsymbol{x}_0, \boldsymbol{x}_1) = 1$, $m_1(\boldsymbol{x}_0, \boldsymbol{x}_1) = \frac{\gamma_1(\boldsymbol{x}_0, \boldsymbol{x}_1)}{\gamma_0(\boldsymbol{x}_0, \boldsymbol{x}_1)}$. We set $m_0 = 1$ because $m_t$ describes the mass evolution relative to the initial mass along each trajectory. Under this setting, we can show that the marginal boundary measures approach $\mu_0$ and $\mu_1$ respectively as $\sigma \to 0$, and the induced measure path follows the WFR geodesic (proof in Appendix A.5).

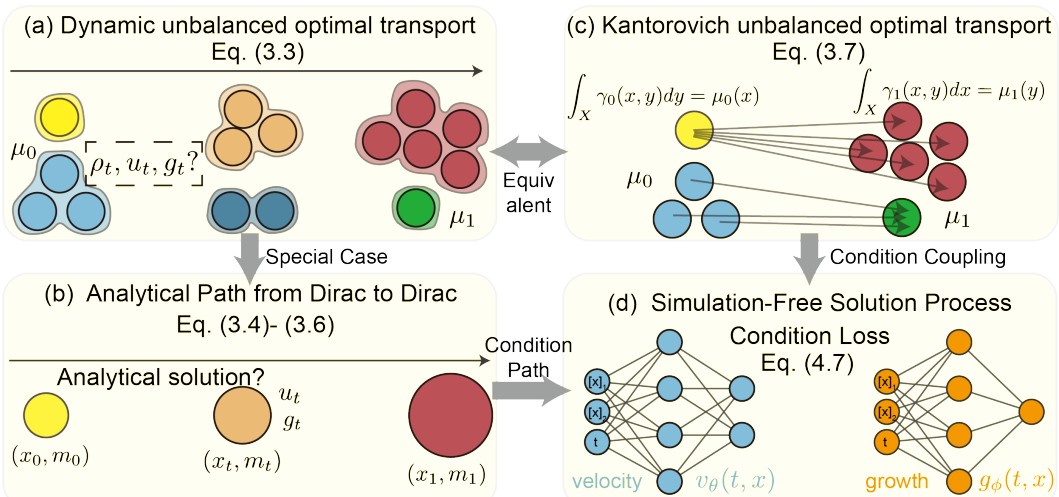

Figure 1: Overview of WFR-FM. **(a)** Dynamic unbalanced OT aims to interpolate the intermediate state in a way that minimizes the action. **(b)** Kantorovich unbalanced OT seeks the semi-coupling that minimizes total cost. **(c)** Analytical trajectory from a Dirac distribution to another Dirac distribution. **(d)** Jointly train the velocity field and the growth rate in the simulation-free manner.

**Proposition 4.2.** *Take travelling Gaussian (4.8) as CGMP, the marginal boundary measure $\rho_0(\boldsymbol{x}_0) \to \mu_0(\boldsymbol{x}_0)$, $\rho_1(\boldsymbol{x}_1) \to \mu_1(\boldsymbol{x}_1)$ as $\sigma \to 0$. Also, the flow generated by travelling Gaussian recover the optimal path of dynamical WFR problem (3.3), thus the marginal $\boldsymbol{u}_t(\boldsymbol{x})$ and $g_t(\boldsymbol{x})$ solve the dynamical WFR problem between $\mu_0$ and $\mu_1$.*

Notably, even for balanced masses $m_0 = m_1 = m$, the WFR mass trajectory $m(t) = 2m(1 - \frac{1}{\sqrt{1+\tau^2}})t(1-t) + m$ remains non-constant. Mass conservation is only recovered as the growth penalty $\delta \to \infty$, leading to the balanced OT limit characterized below (proof in Appendix A.6).

**Proposition 4.3.** *With balanced source $\mu_0$ and target $\mu_1$, let $\delta \to \infty$ (3.3), then the travelling Dirac (3.5) reduces to uniform rectilinear motion with conserved mass, and the travelling Gaussian (4.8) reduces to $\boldsymbol{u}_t(x|z) = \boldsymbol{x}_1 - \boldsymbol{x}_0$, $g_t(\boldsymbol{x}|\boldsymbol{z}) \equiv 0$, thus we recover OT-CFM (Tong et al., 2024a).*

## 5 WFR-FM WORKFLOW FOR TRAJECTORY INFERENCE

In this section, we discussed the setup and workflow of WFR-FM in actual calculations (Figure 1), such as reconstructing dynamics from scRNA-seq data with multiple time points.

**Multi-time point WFR Formulation.** Given samples from distributions $\mu_{t_i}$ at $K+1$ discrete time points, $t = t_0, t_1, \ldots, t_K$, our goal is to reconstruct the continuous evolution of these distributions. To this end, we solve the WFR problem, which entails minimizing the WFR action functional subject to the constraint that $\rho(\boldsymbol{x}, t)$ coincides with the observed distributions $\mu_{t_i}$ at the specified time points.

$$\mathrm{WFR}_\delta^2(\{\mu_{t_0}, \ldots, \mu_{t_K}\}) = \frac{t_K - t_0}{2} \inf_{\rho, g, \boldsymbol{u}} \int_{t_0}^{t_K} \int_{\mathcal{X}} \Big(\|\boldsymbol{u}(\boldsymbol{x}, t)\|_2^2 + \delta^2 \|g(\boldsymbol{x}, t)\|_2^2\Big) \rho_t(\boldsymbol{x}) \, \mathrm{d}\boldsymbol{x} \mathrm{d}t \tag{5.1}$$

s.t. $\qquad \partial_t \rho + \nabla_{\boldsymbol{x}} \cdot (\rho \boldsymbol{u}) = \rho g, \ \rho_{t_0} = \mu_{t_0}, \ldots, \rho_{t_K} = \mu_{t_K}$

The multi-time problem (5.1) reduces to a sequence of WFR steps, as established by the following proposition (proof in Appendix A.7). In practice, we share vector fields and growth rates across all time intervals to enhance generalization.

**Proposition 5.1.** *The solution to the multi-time WFR problem (5.1) is equivalent to the concatenation of the solutions of successive time points.*

**WFR-FM based on Mini-batch WFR-OET.** When solving the coupling, one needs to pre-compute the WFR-OET problem defined in (3.8). However, computing the full mapping directly on large-scale datasets is expensive. Following OT-CFM (Tong et al., 2024a), we adopt a mini-batch strategy.

Let $\mu = \sum_j \delta_{\boldsymbol{x}_j}$ denote the empirical distribution supported on the observed data points. We partition $\mu_0$ and $\mu_1$ into the same number of mini-batches $B$, denoted by $\{\mu_0^{(b)}\}_{b=1}^B$ and $\{\mu_1^{(b)}\}_{b=1}^B$, respectively. For each pair $(\mu_0^{(b)}, \mu_1^{(b)})$, we solve the WFR-OET problem to obtain an optimal coupling $\gamma^{(b)}$. The global coupling $\gamma$ is constructed by concatenating these mini-batch solutions, i.e., $\gamma = \bigoplus_{b=1}^B \gamma^{(b)}$, which approximates to the full coupling between $\mu_0$ and $\mu_1$. We include further discussions on the intuition, limitations, and practical considerations of using mini-batch WFR-OET in Appendix D and provide empirical sensitivity analysis in Table 8.

**Complete Workflow.** We described the training workflow of WFR-FM for trajectory inference problems with multiple time points through Algorithm 1 and inference workflow in Appendix E.

---

**Algorithm 1** WFR-FM Training Workflow

---

**Require:** Sample-able distributions $\mu_{t_0}, \mu_{t_1}, \ldots, \mu_{t_K}$, bandwidth $\sigma$, training batch size $b$, batch size of the OET problem $B$, WFR penalty coefficient $\delta$, vector field $\boldsymbol{v}_{\boldsymbol{\theta}}(\boldsymbol{x}, t)$, growth rate $g_{\boldsymbol{\phi}}(\boldsymbol{x}, t)$.

1: **for** $k = 0 \to K - 1$ **do**
2:      $\gamma^{(k)} \leftarrow \text{OET}(\mu_{t_k}, \mu_{t_{k+1}})$ or mini-batch $\text{OET}(\mu_{t_k}, \mu_{t_{k+1}}; B)$ (Defined in 3.8)
3:      $\gamma_0^{(k)}(\boldsymbol{x}, \boldsymbol{y}) \leftarrow \frac{\gamma^{(k)}(\boldsymbol{x}, \boldsymbol{y})}{\int_{\mathcal{X}} \gamma^{(k)}(\boldsymbol{x}, \boldsymbol{z}) \mathrm{d}\boldsymbol{z}} \mu_{t_k}(\boldsymbol{x}), \gamma_1^{(k)}(\boldsymbol{x}, \boldsymbol{y}) \leftarrow \frac{\gamma^{(k)}(\boldsymbol{x}, \boldsymbol{y})}{\int_{\mathcal{X}} \gamma^{(k)}(\boldsymbol{z}, \boldsymbol{y}) \mathrm{d}\boldsymbol{z}} \mu_{t_{k+1}}(\boldsymbol{y})$ (Theorem 3.1)
4: **end for**
5: **while** Training **do**
6:      **for** $k = 0 \to K - 1$ **do**
7:          $(\boldsymbol{x}_{t_k}, \boldsymbol{x}_{t_{k+1}}) \sim \gamma_0^{(k)}$ /* Sample batches of size $b$ i.i.d. from the datasets */
8:          Calculate the constants $A, B, \boldsymbol{\omega}_0$ and $\tau$ that path depends on. (Defined in 3.6)
9:          $t \sim \mathcal{U}(0, 1), t^{(k)} \leftarrow t_k + (t_{k+1} - t_k)t$
10:          $\boldsymbol{\eta}_{t^{(k)}} \leftarrow \boldsymbol{x}_{t_k} + \boldsymbol{\omega}_0 \Lambda_t(\boldsymbol{x}_{t_k}, \boldsymbol{x}_{t_{k+1}})$ (Defined in 3.5 and 4.8)
11:          $\boldsymbol{x}^{(k)} \sim \mathcal{N}(\boldsymbol{\eta}_{t^{(k)}}, \sigma^2 \mathbf{I})$ (Defined in 3.5 and 4.8)
12:          $\boldsymbol{u}^{(k)} \leftarrow \boldsymbol{\omega}_0 / (m_t(\boldsymbol{x}_{t_k}, \boldsymbol{x}_{t_{k+1}})(t_{k+1} - t_k))$ (Defined in 3.5 and 3.1)
13:          $g^{(k)} \leftarrow \frac{\mathrm{d}}{\mathrm{d}t} \ln m_t(\boldsymbol{x}_{t_k}, \boldsymbol{x}_{t_{k+1}}) / (t_{k+1} - t_k)$ (Defined in 3.5 and 3.1)
14:          $m^{(k)} \leftarrow m_t(\boldsymbol{x}_{t_k}, \boldsymbol{x}_{t_{k+1}}) / \gamma_0^{(k)}(\boldsymbol{x}_{t_k}, \boldsymbol{x}_{t_{k+1}})$ (Defined in 3.5 and 4.8)
15:      **end for**
16:      Concatenate $\{\boldsymbol{x}^{(i)}, t^{(i)}, \boldsymbol{u}^{(i)}, g^{(i)}, m^{(i)}\}_{i=1}^K$ into batch tensors $\{\boldsymbol{x}^c, t^c, \boldsymbol{u}^c, g^c, m^c\}$
17:      $\mathcal{L}_{\text{CUFM}}(\boldsymbol{\theta}, \boldsymbol{\phi}) \leftarrow (\|\boldsymbol{v}_{\boldsymbol{\theta}}(\boldsymbol{x}^c, t^c) - \boldsymbol{u}^c\|_2^2 + \kappa \|g_{\boldsymbol{\phi}}(\boldsymbol{x}^c, t^c) - g^c\|_2^2) m^c$ (Theorem 4.2)
18:      $\boldsymbol{\theta}, \boldsymbol{\phi} \leftarrow \text{Update}((\boldsymbol{\theta}, \boldsymbol{\phi}), (\nabla_{\boldsymbol{\theta}} \mathcal{L}_{\text{CUFM}}(\boldsymbol{\theta}, \boldsymbol{\phi}), \nabla_{\boldsymbol{\phi}} \mathcal{L}_{\text{CUFM}}(\boldsymbol{\theta}, \boldsymbol{\phi}))$
19: **end while**
20: **return** $\boldsymbol{v}_{\boldsymbol{\theta}}$ and $g_{\boldsymbol{\phi}}$

---

## 6 EXPERIMENT RESULTS

We evaluate WFR-FM with key questions **Q1:** Can WFR-FM transport $\mu_{t_0}$ to $\mu_{t_i}$? **Q2:** Does it approximate the dynamic WFR solution? **Q3:** How accurate is interpolation of unobserved data? **Q4:** How does its scalability compare to existing methods? **Q5:** How well does it recover the underlying birth-death dynamics?

**Accuracy of Distribution and Mass Transportation (Q1)** We evaluate the distribution time-propagation task on three synthetic datasets: Simulation Gene (Gene), Dygen and the 1000D Gaussian Mixtures (Gaussian) using two metrics (Appendix B): the 1-Wasserstein distance ($\mathcal{W}_1$) to measure the similarity between normalized predicted and true distributions, and Relative Mass Error (RME) to assess how well the model captures cell population growth. WFR-FM consistently achieves superior performance across all datasets, with the lowest $\mathcal{W}_1$ and near-zero RME (Table 1). In particular, it improves distributional matching on low- and mid-dimensional datasets and remains the most accurate in the high-dimensional case. We also evaluate WFR-FM on real datasets in Appendix B. Visualization on the Gene dataset and the real-world embroyid bodies (EB) dataset (Moon et al., 2019) further confirms that WFR-FM captures accurate dynamics (Figure 2).

**Closeness with Action of Static WFR (Q2)** We next examine whether WFR-FM provides a valid solution to the dynamic WFR formulation by comparing the path action of inferred trajectories with

Table 1: Mean $\mathcal{W}_1$ and RME (only for unbalanced methods)on synthetic datasets. For the baselines that exhibit randomness in inference, we report the mean value and standard deviation over 5 runs.

| Method | Gene (2D) | | Dyngen (5D) | | Gaussian (1000D) | |
|---|---|---|---|---|---|---|
| | $\mathcal{W}_1$ ($\downarrow$) | RME ($\downarrow$) | $\mathcal{W}_1$ ($\downarrow$) | RME ($\downarrow$) | $\mathcal{W}_1$ ($\downarrow$) | RME ($\downarrow$) |
| MMFM (Rohbeck et al., 2025) | 0.298 | — | 1.371 | — | 2.833 | — |
| Metric FM (Kapusniak et al., 2024) | 0.311 | — | 1.767 | — | 3.794 | — |
| SF2M (Tong et al., 2024b) | $0.224_{\pm0.007}$ | — | $1.277_{\pm0.017}$ | — | $3.543_{\pm0.002}$ | — |
| MIOFlow (Huguet et al., 2022) | 0.148 | — | 0.965 | — | 2.858 | — |
| TIGON (Sha et al., 2024) | 0.045 | 0.014 | 0.512 | 0.047 | 2.263 | 0.127 |
| DeepRUOT (Zhang et al., 2025a) | $0.043_{\pm0.002}$ | $0.017_{\pm0.001}$ | $0.623_{\pm0.032}$ | $0.065_{\pm0.011}$ | $3.785_{\pm0.009}$ | $0.303_{\pm0.070}$ |
| Var-RUOT (Sun et al., 2025) | $0.079_{\pm0.003}$ | $0.008_{\pm0.002}$ | $0.522_{\pm0.008}$ | $0.177_{\pm0.007}$ | $2.813_{\pm0.004}$ | $0.041_{\pm0.006}$ |
| UOT-FM (Eyring et al., 2024) | 0.093 | 0.010 | 1.204 | 0.097 | 2.771 | **0.033** |
| VGFM (Wang et al., 2025) | 0.046 | 0.006 | 0.598 | 0.037 | 3.010 | 0.037 |
| WFR-FM (Ours) | **0.019** | **0.001** | **0.135** | **0.005** | **2.233** | 0.044 |

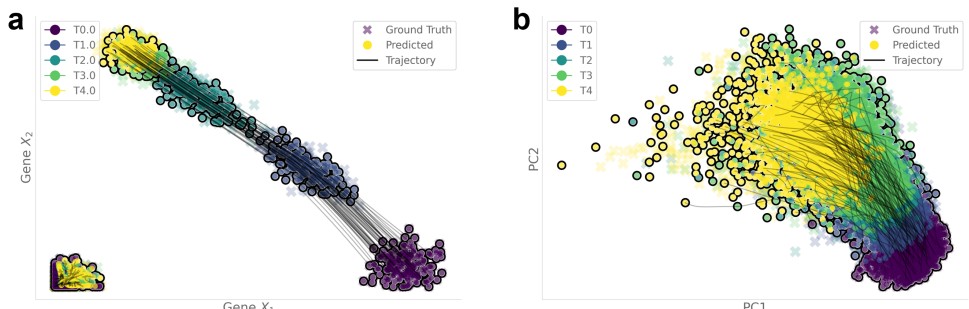

Figure 2: Learned trajectories on the (a) Simulation Gene dataset (b) EB dataset

the reference action obtained from the static WFR-OET solver (Table 2). Across all datasets, WFR-FM consistently yields the closest approximation to the true action. While some methods such as Var-RUOT produce actions that are lower than the reference action and also below those of WFR-FM, they indeed trade off distribution fidelity to achieve a lower action. These results show that WFR-FM faithfully approximates the WFR geodesics in distribution interpolation task.

**Interpolation Results under the Hold-One-Out Experiment (Q3)** We evaluate how well WFR-FM interpolates unobserved data in real multi-time scRNA-seq datasets by holding out one intermediate time point during training (Table 3). Across EMT (Cook & Vanderhyden, 2020), CITE-seq (Lance et al., 2022), and mouse hematopoiesis data (Mouse) (Weinreb et al., 2020), WFR-FM achieves the best interpolation accuracy, while comparable with top baselines on EB dataset. This suggests that unbalanced methods may better accommodate these datasets, possibly due to population imbalance from cell proliferation; see Appendix B.10 for further discussion.

**Scalability to Large-Scale Dataset (Q4)** We examine the scalability of WFR-FM on the large-scale EB dataset (100D) by comparing runtime, memory, and accuracy (Figure 3). WFR-FM achieves both high accuracy and efficiency, outperforming methods that only learn velocity fields

Table 2: Path action on synthetic datasets.

| Method | Gene | Dyngen | Gaussian |
|---|---|---|---|
| TIGON | 1.367 | 11.654 | 6.794 |
| DeepRUOT | $1.286_{\pm0.007}$ | $14.070_{\pm0.151}$ | $14.370_{\pm1.069}$ |
| Var-RUOT | $1.208_{\pm0.012}$ | $8.357_{\pm0.223}$ | $7.146_{\pm0.896}$ |
| WFR-FM | **1.305** | **9.410** | **7.529** |
| Static Reference | 1.333 | 9.569 | 10.531 |

Table 3: Mean $\mathcal{W}_1$ over held-out time points on EMT, EB, CITE and Mouse datasets.

| Method | EMT (10D) | EB (50D) | CITE (50D) | Mouse (50D) |
|---|---|---|---|---|
| MMFM | 0.323 | 11.213 | 38.521 | 8.263 |
| Metric FM | 0.314 | 10.726 | 37.342 | 7.753 |
| SF2M | $0.308_{\pm 0.001}$ | $10.986_{\pm 0.006}$ | $38.333_{\pm 0.002}$ | $8.646_{\pm 0.004}$ |
| MIOFlow | 0.325 | 10.960 | 39.574 | 7.779 |
| TIGON | 0.360 | 11.080 | 38.159 | 6.868 |
| DeepRUOT | $0.323_{\pm 0.002}$ | $\mathbf{10.075}_{\pm 0.004}$ | $37.892_{\pm 0.002}$ | $\underline{6.847}_{\pm 0.003}$ |
| Var-RUOT | $0.320_{\pm 0.003}$ | $11.035_{\pm 0.017}$ | $38.393_{\pm 0.029}$ | $8.672_{\pm 0.040}$ |
| UOT-FM | 0.322 | 11.344 | 38.649 | 9.332 |
| VGFM | $\underline{0.301}$ | 10.370 | 37.386 | 8.496 |
| **WFR-FM** | **0.298** | $\underline{10.157}$ | **37.221** | **6.586** |

in accuracy and ODE-based methods in efficiency. WFR-FM provides a favorable balance between computational cost and accuracy, making it well-suited for large-scale single-cell datasets.

Sensitivity analysi confirms WFR-FM's robustness to key hyperparameters, specifically the growth penalty $\delta$ (Table 5) and batch size for mini-batch WFR-OET (Table 8).

**Evaluation of Learned Growth Dynamics (Q5)** To confirm WFR-FM captures birth-death dynamics beyond marginal matching, we evaluate the learned growth rate $g_\phi(x,t)$ on the Simulation Gene dataset, where ground truth is provided by known SDEs. We calculate the Pearson correlation coefficient ($g_{\text{corr}}$) between the predicted growth rate and the ground truth growth rate $g = \alpha_g \frac{X_2^2}{1+X_2^2}$ on the data points from the dataset. Table 4 shows that WFR-FM outperforms baselines, validating its ability to reliably estimate growth rates.

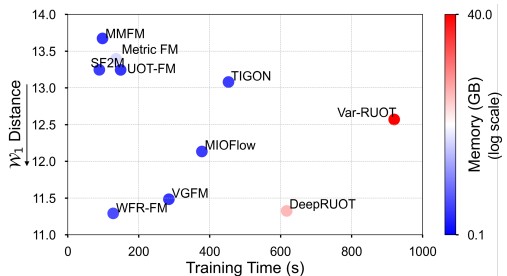

Figure 3: Efficiency on the 100D EB dataset

Table 4: Growth rate correlation ($g_{\text{corr}}$) on Simulation Gene dataset. Best results are in **bold**.

| Method | TIGON | DeepRUOT | Var-RUOT | Action Matching | **WFR-FM (Ours)** |
|---|---|---|---|---|---|
| **Correlation ($g_{\text{corr}}$)** | 0.9705 | 0.9688 | 0.9214 | 0.5851 | **0.9913** |

## 7 CONCLUSION, LIMITATION AND DISCUSSION

In this work, we introduce **WFR-FM**, a framework unifying the WFR metric with flow matching for simulation-free, dynamic unbalanced OT. By jointly learning learning velocity fields and growth rates, WFR-FM provides a principled formulation with theoretical guarantees that exactly solves the WFR problem. Extensive experiments demonstrate its effectiveness in reconstructing dynamics with proliferation and apoptosis, establishing it as a robust tool for biological trajectory inference.

Like other OT-FM methods, WFR-FM relies on solving a static OT problem, which can be costly for very large datasets. To address this, we adopt approximation strategies such as mini-batch OT and entropy-regularized Sinkhorn iterations. Incorporating uncertainty quantification in WFR-FM could also enhance the performance on noisy single-cell datasets. While our focus has been on WFR geometry, the framework established is indeed general: as long as the static OT problem can be efficiently solved and the Dirac-to-Dirac path admits a closed-form solution (e.g., when linear growth penalties are chosen, as in Chizat et al. (2018b)), our approach can be extended to other unbalanced transport functionals. We anticipate that this generality will enable new classes of flow-matching methods, broadening their applications across scientific domains where mass is not conserved or other machine learning application scenarios with unbalanced datasets.

## REPRODUCIBILITY STATEMENT

We have taken several measures to ensure the reproducibility of our work. All theoretical results are accompanied by complete proofs provided in the Appendix. Detailed descriptions of data processing steps are also included in the Appendix to facilitate replication of our experiments. The source code is available at `https://github.com/QiangweiPeng/WFR-FM`.

## AUTHOR CONTRIBUTIONS

Q.P., P.Z., and T.L.conceived the project. Q.P. and Z.W. implemented the algorithm and conducted data analysis. J.Y. and Y.S. performed the mathematical framework design and theoretical analysis. Q.P., Z.W., J.Y., and Y.S. contributed equally to this work. P.Z., T.L., L.Z., and Q.N. supervised the research. P.Q., Z.W., J.Y., Y.S., and P.Z. wrote the manuscript. All authors revised and approved the manuscript.

## ACKNOWLEDGMENTS

This work was supported by the National Key R&D Program of China (No. 2021YFA1003301 to T.L., and 2024YFA0919500 to L.Z.) and National Natural Science Foundation of China (NSFC No. 12288101 to T.L. & P.Z. & L.Z., 12225102 to L.Z., 8206100646 to P.Z., and T2321001 to P.Z. & L.Z.). We acknowledge the support from the High-performance Computing Platform of Peking University for computation. We thank the anonymous referees for their valuable feedback and constructive suggestions.

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

# A PROOFS

## A.1 PROOF OF THEOREM 3.1

**Lemma A.1.** *The static form or WFR problem 3.7 is convex with respect to $\gamma_0(\boldsymbol{x}, \boldsymbol{y})$ and $\gamma_1(\boldsymbol{x}, \boldsymbol{y})$.*

*Proof.*

First, we prove that the integrand:

$$\mathscr{J}(\gamma_0, \gamma_1, \boldsymbol{x}, \boldsymbol{y}) = \gamma_0 + \gamma_1 - 2\sqrt{\gamma_0\gamma_1}\overline{\cos}\left(\frac{\|\boldsymbol{x} - \boldsymbol{y}\|_2}{2\delta}\right) \tag{A.1}$$

is convex with respect to $\gamma_0$ and $\gamma_1$. To do this, we compute the Hessian matrix:

$$H = \begin{bmatrix} \frac{1}{2}\overline{\cos}\left(\frac{\|\boldsymbol{x}-\boldsymbol{y}\|_2}{2\delta}\right)\gamma_0^{-\frac{3}{2}}\gamma_1^{\frac{1}{2}} & -\frac{1}{2}\overline{\cos}\left(\frac{\|\boldsymbol{x}-\boldsymbol{y}\|_2}{2\delta}\right)\gamma_0^{-\frac{1}{2}}\gamma_1^{-\frac{1}{2}} \\ -\frac{1}{2}\overline{\cos}\left(\frac{\|\boldsymbol{x}-\boldsymbol{y}\|_2}{2\delta}\right)\gamma_0^{-\frac{1}{2}}\gamma_1^{-\frac{1}{2}} & \frac{1}{2}\overline{\cos}\left(\frac{\|\boldsymbol{x}-\boldsymbol{y}\|_2}{2\delta}\right)\gamma_0^{\frac{1}{2}}\gamma_1^{-\frac{3}{2}} \end{bmatrix} \tag{A.2}$$

Next, we check its principal minors. Since $\gamma_0, \gamma_1 \geq 0$ and $\overline{\cos}\left(\frac{|\boldsymbol{x}-\boldsymbol{y}|}{2\delta}\right) \geq 0$, the diagonal elements of the matrix are non-negative. And for the determinant:

$$\det(H) = \frac{1}{4}\overline{\cos}^2\left(\frac{\|\boldsymbol{x} - \boldsymbol{y}\|_2}{2\delta}\right)\gamma_0^{-1}\gamma_1^{-1} - \frac{1}{4}\overline{\cos}^2\left(\frac{\|\boldsymbol{x} - \boldsymbol{y}\|_2}{2\delta}\right)\gamma_0^{-1}\gamma_1^{-1} = 0 \tag{A.3}$$

Therefore, $H$ is a positive semidefinite matrix, and $\mathscr{J}$ is convex with respect to $\gamma_0$ and $\gamma_1$. Our optimization objective:

$$\mathscr{F}[\gamma_0, \gamma_1] = \int_{\mathcal{X}^2} \mathscr{J}(\gamma_0(\boldsymbol{x}, \boldsymbol{y}), \gamma_1(\boldsymbol{x}, \boldsymbol{y}), \boldsymbol{x}, \boldsymbol{y})\mathrm{d}\boldsymbol{x}\mathrm{d}\boldsymbol{y} \quad \left(\frac{1}{2\delta^2}\mathrm{WFR}_\delta^2(\mu_0, \mu_1) = \inf_{\gamma_0, \gamma_1}\mathscr{F}[\gamma_0, \gamma_1]\right) \tag{A.4}$$

is a functional of $\gamma_0$ and $\gamma_1$. Consider any $\gamma_{0,a}, \gamma_{0,b}, \gamma_{1,a}, \gamma_{1,b} \in \mathcal{M}_+(\mathcal{X}^2)$ and $t \in [0, 1]$:

$$\begin{aligned} \mathscr{F}[t\gamma_{0,a} + (1-t)\gamma_{0,b}, t\gamma_{1,a} + (1-t)\gamma_{1,b}] &= \int_{\mathcal{X}^2} \mathscr{J}(t\gamma_{0,a} + (1-t)\gamma_{0,b}, t\gamma_{1,a} + (1-t)\gamma_{1,b})\mathrm{d}\boldsymbol{x}\mathrm{d}\boldsymbol{y} \\ &\leq \int_{\mathcal{X}^2} (t\mathscr{J}(\gamma_{0,a}, \gamma_{1,a}) + (1-t)\mathscr{J}(\gamma_{0,b}, \gamma_{1,b}))\,\mathrm{d}\boldsymbol{x}\mathrm{d}\boldsymbol{y} \\ &= t\mathscr{F}[\gamma_{0,a}, \gamma_{1,a}] + (1-t)\mathscr{F}[\gamma_{0,b}, \gamma_{1,b}] \end{aligned} \tag{A.5}$$

Therefore, the problem is convex with respect to $\gamma_0$ and $\gamma_1$.

**Theorem 3.1.** *Let $\gamma^\star$ be the optimal coupling of the OET problem (3.8), then the semi-coupling $\gamma_0(\boldsymbol{x}, \boldsymbol{y}) = \frac{\gamma^\star(\boldsymbol{x}, \boldsymbol{y})}{\int_{\mathcal{X}} \gamma^\star(\boldsymbol{x}, \boldsymbol{z})\mathrm{d}\boldsymbol{z}}\mu_0(\boldsymbol{x}), \gamma_1(\boldsymbol{x}, \boldsymbol{y}) = \frac{\gamma^\star(\boldsymbol{x}, \boldsymbol{y})}{\int_{\mathcal{X}} \gamma^\star(\boldsymbol{z}, \boldsymbol{y})\mathrm{d}\boldsymbol{z}}\mu_1(\boldsymbol{y})$ solves the static form of WFR problem (3.7).*

*Proof.*

Let

$$\mathscr{J}(\gamma_0(\boldsymbol{x}, \boldsymbol{y}), \gamma_1(\boldsymbol{x}, \boldsymbol{y}), \boldsymbol{x}, \boldsymbol{y}) = \frac{1}{2\delta^2}\mathrm{WFR}_\delta^2(\gamma_0(\boldsymbol{x}, \boldsymbol{y})\delta_{\boldsymbol{x}}, \gamma_1(\boldsymbol{x}, \boldsymbol{y})\delta_{\boldsymbol{y}})$$

The computation of the WFR distance between two distributions can be formulated as the optimization problem (3.8):

$$\frac{1}{2\delta^2}\mathrm{WFR}_\delta^2(\mu_0, \mu_1) = \inf_{(\gamma_0, \gamma_1) \in (\mathcal{M}_+(\mathcal{X}^2))^2} \int_{\mathcal{X}^2} \mathscr{J}(\gamma_0(\boldsymbol{x}, \boldsymbol{y}), \gamma_1(\boldsymbol{x}, \boldsymbol{y}), \boldsymbol{x}, \boldsymbol{y})\mathrm{d}\boldsymbol{x}\mathrm{d}\boldsymbol{y} \tag{A.6}$$

$$\text{s.t.} \int_{\mathcal{X}} \gamma_0(\boldsymbol{x}, \boldsymbol{y})\mathrm{d}\boldsymbol{y} = \mu_0(\boldsymbol{x}), \int_{\mathcal{X}} \gamma_1(\boldsymbol{x}, \boldsymbol{y})\mathrm{d}\boldsymbol{x} = \mu_1(\boldsymbol{y}) \tag{A.7}$$

By introducing the dual variables $\phi(\boldsymbol{x}), \psi(\boldsymbol{y})$, it can be transformed into the dual problem:

$$\frac{1}{2\delta^2}\text{WFR}_\delta^2(\mu_0, \mu_1) = \sup_{(\phi,\psi)\in C(\mathcal{X})^2} \inf_{(\gamma_0,\gamma_1)\in(\mathcal{M}_+(\mathcal{X}^2))^2} \int_{\mathcal{X}^2} \mathscr{J}(\gamma_0(\boldsymbol{x},\boldsymbol{y}), \gamma_1(\boldsymbol{x},\boldsymbol{y}), \boldsymbol{x}, \boldsymbol{y})\mathrm{d}\boldsymbol{x}\mathrm{d}\boldsymbol{y}$$

$$- \int_{\mathcal{X}} \phi(\boldsymbol{x}) \left( \int_{\mathcal{X}} \gamma_0(\boldsymbol{x},\boldsymbol{y})\mathrm{d}\boldsymbol{y} - \mu_0(\boldsymbol{x}) \right) \mathrm{d}\boldsymbol{x} - \int_{\mathcal{X}} \psi(\boldsymbol{y}) \left( \int_{\mathcal{X}} \gamma_1(\boldsymbol{x},\boldsymbol{y})\mathrm{d}\boldsymbol{x} - \mu_1(\boldsymbol{y}) \right) \mathrm{d}\boldsymbol{y}$$

$$= \sup_{(\phi,\psi)\in C(\mathcal{X})^2} \int_{\mathcal{X}} \phi(\boldsymbol{x})\rho_0(\boldsymbol{x})\mathrm{d}\boldsymbol{x} + \int_{\mathcal{X}} \psi(\boldsymbol{y})\rho_1(\boldsymbol{y})\mathrm{d}\boldsymbol{y}$$

$$\text{s.t.} \quad \phi(\boldsymbol{x}) \le 1, \psi(\boldsymbol{y}) \le 1, (1 - \phi(\boldsymbol{x}))(1 - \psi(\boldsymbol{y})) \ge \overline{\cos}^2\left( \frac{\|\boldsymbol{x} - \boldsymbol{y}\|_2}{2\delta} \right) \tag{A.8}$$

By a change of variables, let $u = -\log(1 - \phi), v = -\log(1 - \psi)$, the above dual problem becomes:

$$\frac{1}{2\delta^2}\text{WFR}_\delta^2(\mu_0, \mu_1) = \sup_{(u,v)\in C(\mathcal{X})^2} \int_\Omega (1 - \exp(-u(\boldsymbol{x})))\rho_0(\boldsymbol{x})\mathrm{d}\boldsymbol{x} + \int (1 - \exp(-v(\boldsymbol{y})))\rho_1(\boldsymbol{y})\mathrm{d}\boldsymbol{y}$$
$$\tag{A.9}$$

$$\text{s.t.} \quad u(\boldsymbol{x}) + v(\boldsymbol{y}) \le -\log\left( \overline{\cos}^2\left( \frac{\|\boldsymbol{x} - \boldsymbol{y}\|_2}{2\delta} \right) \right) \tag{A.10}$$

The optimal solution $\gamma_0^\star(\boldsymbol{x},\boldsymbol{y}), \gamma_1^\star(\boldsymbol{x},\boldsymbol{y})$ of this optimization problem and the optimal dual variables $u^\star(\boldsymbol{x}), v^\star(\boldsymbol{y})$ should satisfy the following KKT conditions:

(1) Complementary slackness condition:

$$\begin{cases} (1 - \exp(-u^\star(\boldsymbol{x}))) \left( \int_{\mathcal{X}} \gamma_0^\star(\boldsymbol{x},\boldsymbol{y})\mathrm{d}\boldsymbol{y} - \mu_0(\boldsymbol{x}) \right) = 0 \\ (1 - \exp(-v^\star(\boldsymbol{y}))) \left( \int_{\mathcal{X}} \gamma_1^\star(\boldsymbol{x},\boldsymbol{y})\mathrm{d}\boldsymbol{x} - \mu_1(\boldsymbol{y}) \right) = 0 \end{cases} \tag{A.11}$$

(2) stationary condition:

$$\begin{cases} \partial_{\gamma_0}\mathscr{J}(\gamma_0, \gamma_1, \boldsymbol{x}, \boldsymbol{y})|_{\gamma_0^\star} = (1 - \exp(u^\star(\boldsymbol{x}))) \\ \partial_{\gamma_1}\mathscr{J}(\gamma_0, \gamma_1, \boldsymbol{x}, \boldsymbol{y})|_{\gamma_1^\star} = (1 - \exp(v^\star(\boldsymbol{y}))) \end{cases} \tag{A.12}$$

The solution we constructed, $\gamma_0(\boldsymbol{x},\boldsymbol{y}) = \dfrac{\gamma^\star(\boldsymbol{x},\boldsymbol{y})}{\int_{\mathcal{X}} \gamma^\star(\boldsymbol{x},\boldsymbol{z})\mathrm{d}\boldsymbol{z}}\mu_0(\boldsymbol{x}), \gamma_1(\boldsymbol{x},\boldsymbol{y}) = \dfrac{\gamma^\star(\boldsymbol{x},\boldsymbol{y})}{\int_{\mathcal{X}} \gamma^\star(\boldsymbol{z},\boldsymbol{y})\mathrm{d}\boldsymbol{z}}\mu_1(\boldsymbol{y})$,
satisfies the complementary slackness condition. Next, we need to verify that it satisfies the stationary condition. Consider the equivalent unconstrained optimization problem (3.9):

$$\frac{1}{2\delta^2}\text{WFR}_\delta^2(\mu_0, \mu_1) = 2\delta^2 \inf_{\gamma\in\mathcal{M}_+(\mathcal{X}^2)} \left( \int_{\mathcal{X}^2} -2\ln\overline{\cos}\left( \frac{\|\boldsymbol{x} - \boldsymbol{y}\|_2}{2\delta} \right) \gamma(\boldsymbol{x},\boldsymbol{y})\mathrm{d}\boldsymbol{x}\mathrm{d}\boldsymbol{y} \right.$$

$$\left. + \text{KL}\left( \int_{\mathcal{X}} \gamma(\boldsymbol{x},\boldsymbol{y})\mathrm{d}\boldsymbol{y}\|\mu_0(\boldsymbol{x}) \right) + \text{KL}\left( \int_{\mathcal{X}} \gamma(\boldsymbol{x},\boldsymbol{y})\mathrm{d}\boldsymbol{x}\|\mu_1(\boldsymbol{y}) \right) \right)$$

Transform it into a constrained optimization problem:

$$\frac{1}{2\delta^2}\text{WFR}_\delta^2(\mu_0, \mu_1) = 2\delta^2 \inf_{\gamma\in\mathcal{M}_+(\mathcal{X}^2)} \left( \int_{\mathcal{X}^2} -2\ln\overline{\cos}\left( \frac{\|\boldsymbol{x} - \boldsymbol{y}\|_2}{2\delta} \right) \gamma(\boldsymbol{x},\boldsymbol{y})\mathrm{d}\boldsymbol{x}\mathrm{d}\boldsymbol{y} \right.$$
$$\tag{A.13}$$
$$\left. + \text{KL}\left( \rho_0(\boldsymbol{x})\|\mu_0(\boldsymbol{x}) \right) + \text{KL}\left( \rho_1(\boldsymbol{x})\|\mu_1(\boldsymbol{y}) \right) \right)$$

$$\text{s.t.} \int_{\mathcal{X}} \gamma(\boldsymbol{x},\boldsymbol{y})\mathrm{d}\boldsymbol{y} = \rho_0(\boldsymbol{x}), \int_{\mathcal{X}} \gamma(\boldsymbol{x},\boldsymbol{y})\mathrm{d}\boldsymbol{x} = \rho_1(\boldsymbol{y}) \tag{A.14}$$

Introduce dual variables $u(\boldsymbol{x}), v(\boldsymbol{y})$ to obtain the dual problem:

$$\frac{1}{2\delta^2}\text{WFR}_\delta^2(\mu_0, \mu_1) = \sup_{(u,v)\in C(\mathcal{X})^2} \inf_{\rho_0,\rho_1\in\mathcal{M}_+(\mathcal{X}),\gamma\in\mathcal{M}_+(\mathcal{X}^2)} \left( \int_{\mathcal{X}^2} -2\ln\overline{\cos}\left(\frac{\|\boldsymbol{x}-\boldsymbol{y}\|_2}{2\delta}\right)\gamma(\boldsymbol{x},\boldsymbol{y})\mathrm{d}\boldsymbol{x}\mathrm{d}\boldsymbol{y} \right.$$

$$+ \text{KL}\left(\rho_0(\boldsymbol{x})\|\mu_0(\boldsymbol{x})\right) + \text{KL}\left(\rho_1(\boldsymbol{x})\|\mu_1(\boldsymbol{y})\right) \Big)$$

$$- \int_{\mathcal{X}} u(\boldsymbol{x})\left(\int_{\mathcal{X}}\gamma(\boldsymbol{x},\boldsymbol{y})\mathrm{d}\boldsymbol{y} - \rho_0(\boldsymbol{x})\right)\mathrm{d}\boldsymbol{x} - \int_{\mathcal{X}} v(\boldsymbol{y})\left(\int_{\mathcal{X}}\gamma(\boldsymbol{x},\boldsymbol{y})\mathrm{d}\boldsymbol{x} - \rho_1(\boldsymbol{y})\right)\mathrm{d}\boldsymbol{y}$$

$$= \sup_{(u,v)\in C(\mathcal{X})^2} \inf_{\rho_0,\rho_1\in\mathcal{M}_+(\mathcal{X}),\gamma\in\mathcal{M}_+(\mathcal{X}^2)} \int_{\mathcal{X}^2}\left(-2\ln\overline{\cos}\left(\frac{\|\boldsymbol{x}-\boldsymbol{y}\|_2}{2\delta}\right) - u(\boldsymbol{x}) - v(\boldsymbol{y})\right)$$

$$+ \left(\text{KL}(\rho_0\|\mu_0) + \int_{\mathcal{X}} u(\boldsymbol{x})\rho_0(\boldsymbol{x})\mathrm{d}\boldsymbol{x}\right) + \left(\text{KL}(\rho_1\|\mu_1) + \int_{\mathcal{X}} u(\boldsymbol{x})\rho_1(\boldsymbol{x})\mathrm{d}\boldsymbol{x}\right) \tag{A.15}$$

Now, $\rho_0, \rho_1, \gamma$ are in three separate terms, so when solving the inner optimization problem, we can minimize these three terms separately. From the first term, we can obtain the constraint on the dual variables:

$$\text{s.t.}\quad u(\boldsymbol{x}) + v(\boldsymbol{y}) \le -\log\left(\overline{\cos}^2\left(\frac{\|\boldsymbol{x}-\boldsymbol{y}\|_2}{2\delta}\right)\right) \tag{A.16}$$

By solving the inner optimization problem, we can obtain the relationship between the dual variables and the primal variables:

$$\forall(\boldsymbol{x},\boldsymbol{y})\in\text{supp}(\gamma(\boldsymbol{x},\boldsymbol{y})), -\log\left(\overline{\cos}^2\left(\frac{\|\boldsymbol{x}-\boldsymbol{y}\|_2}{2\delta}\right)\right) - u(\boldsymbol{x}) - v(\boldsymbol{y}) = 0 \tag{A.17}$$

$$\frac{\rho_0(\boldsymbol{x})}{\mu_0(\boldsymbol{x})} = \exp(-u(\boldsymbol{x})), \frac{\rho_1(\boldsymbol{y})}{\mu_1(\boldsymbol{y})} = \exp(-v(\boldsymbol{y})) \tag{A.18}$$

Substituting these relationships into the dual problem of (3.9), we get:

$$\frac{1}{2\delta^2}\text{WFR}_\delta^2(\mu_0,\mu_1) = \sup_{(u,v)\in C(\mathcal{X})^2}\int_\Omega (1-\exp(-u(\boldsymbol{x})))\rho_0(\boldsymbol{x})\mathrm{d}\boldsymbol{x} + \int (1-\exp(-v(\boldsymbol{y})))\rho_1(\boldsymbol{y})\mathrm{d}\boldsymbol{y} \tag{A.19}$$

This is exactly the form of the dual problem of (3.8) after the change of variables. Therefore, (3.8) and (3.9) have the same dual problem, and the $u(\boldsymbol{x}), v(\boldsymbol{y})$ we introduced when constructing the dual problem of (3.9) are the same $u(\boldsymbol{x}), v(\boldsymbol{y})$ in the dual problem of (3.8). Consider the relationship between the primal and dual variables when both are optimal in (3.9):

$$\forall(\boldsymbol{x},\boldsymbol{y})\in\text{supp}(\gamma(\boldsymbol{x},\boldsymbol{y})), -\log\left(\overline{\cos}^2\left(\frac{\|\boldsymbol{x}-\boldsymbol{y}\|_2}{2\delta}\right)\right) - u^\star(\boldsymbol{x}) - v^\star(\boldsymbol{y}) = 0 \tag{A.20}$$

$$\frac{\int_{\mathcal{X}}\gamma^\star(\boldsymbol{x},\boldsymbol{y})\mathrm{d}\boldsymbol{y}}{\mu_0(\boldsymbol{x})} = \exp(-u^\star(\boldsymbol{x})), \frac{\int_{\mathcal{X}}\gamma^\star(\boldsymbol{x},\boldsymbol{y})\mathrm{d}\boldsymbol{x}}{\mu_1(\boldsymbol{y})} = \exp(-v^\star(\boldsymbol{y})) \tag{A.21}$$

Since:

$$\partial_{\gamma_0}\mathscr{I}|_{\gamma_0^\star} = 1 - \sqrt{\frac{\gamma_1^\star(\boldsymbol{x},\boldsymbol{y})}{\gamma_0^\star(\boldsymbol{x},\boldsymbol{y})}}\overline{\cos}\left(\frac{\|\boldsymbol{x}-\boldsymbol{y}\|_2}{2\delta}\right) \tag{A.22}$$

Substituting the solution we constructed, $\gamma_0(\boldsymbol{x},\boldsymbol{y}) = \dfrac{\gamma^\star(\boldsymbol{x},\boldsymbol{y})}{\int_{\mathcal{X}}\gamma^\star(\boldsymbol{x},\boldsymbol{z})\mathrm{d}\boldsymbol{z}}\mu_0(\boldsymbol{x}), \gamma_1(\boldsymbol{x},\boldsymbol{y}) = \dfrac{\gamma^\star(\boldsymbol{x},\boldsymbol{y})}{\int_{\mathcal{X}}\gamma^\star(\boldsymbol{z},\boldsymbol{y})\mathrm{d}\boldsymbol{z}}\mu_1(\boldsymbol{y})$, and the above relationship between primal and dual variables, we can verify:

$$\partial_{\gamma_0}\mathscr{I}|_{\gamma_0^\star} = 1 - \exp\left(\frac{1}{2}(v^\star(\boldsymbol{y}) - u^\star(\boldsymbol{x}))\right)\exp\left(-\frac{1}{2}(u^\star(\boldsymbol{x}) + v^\star(\boldsymbol{y}))\right)$$

$$= 1 - \exp(-u^\star(\boldsymbol{x}))$$

Thus, the stationary condition is satisfied. The other stationary condition can be verified by substitution in the same way. Therefore, the solution we constructed satisfies the KKT conditions of problem 3.7. Accroding to Lemma A.1, problem 3.7 is convex with respect to $\gamma_0, \gamma_1$, the solution we constructed is the optimal solution of 3.7.

### A.2 PROOF OF THEOREM 4.1

**Theorem 4.1.** *The marginal vector field and rate function (4.4) generates the marginal measure path (4.3) from $p_0(\boldsymbol{x})$ for any $q(\boldsymbol{z})$ independent of $\boldsymbol{x}$ and $t$.*

*Proof.* To verify this, it is sufficient to show that the marginal measure $\rho_t(\boldsymbol{x})$ satisfies the continuity equation with source term (3.2).

$$
\begin{aligned}
\partial_t \rho_t(\boldsymbol{x}) &= \frac{d}{dt} \int \rho_t(\boldsymbol{x}|\boldsymbol{z}) q(\boldsymbol{z}) \mathrm{d}\boldsymbol{z} \\
&= \int \partial_t \rho_t(\boldsymbol{x}|\boldsymbol{z}) q(\boldsymbol{z}) \mathrm{d}\boldsymbol{z} \\
&\overset{(1)}{=} \int -\nabla_{\boldsymbol{x}} \cdot (\boldsymbol{u}_t(\boldsymbol{x}|\boldsymbol{z}) \rho_t(\boldsymbol{x}|\boldsymbol{z})) q(\boldsymbol{z}) \mathrm{d}\boldsymbol{z} + \int g_t(\boldsymbol{x}|\boldsymbol{z}) \rho_t(\boldsymbol{x}|\boldsymbol{z}) q(\boldsymbol{z}) \mathrm{d}\boldsymbol{z} \\
&= -\nabla_{\boldsymbol{x}} \cdot \int \boldsymbol{u}_t(\boldsymbol{x}|\boldsymbol{z}) \rho_t(\boldsymbol{x}|\boldsymbol{z}) q(\boldsymbol{z}) \mathrm{d}\boldsymbol{z} + \int g_t(\boldsymbol{x}|\boldsymbol{z}) \rho_t(\boldsymbol{x}|\boldsymbol{z}) q(\boldsymbol{z}) \mathrm{d}\boldsymbol{z} \\
&= -\nabla_{\boldsymbol{x}} \cdot (\rho_t(\boldsymbol{x}) \int \boldsymbol{u}_t(\boldsymbol{x}|\boldsymbol{z}) \frac{\rho_t(\boldsymbol{x}|\boldsymbol{z}) q(\boldsymbol{z})}{\rho_t(\boldsymbol{x})} \mathrm{d}\boldsymbol{z}) + \rho_t(\boldsymbol{x}) \int g_t(\boldsymbol{x}|\boldsymbol{z}) \frac{\rho_t(\boldsymbol{x}|\boldsymbol{z}) q(\boldsymbol{z})}{\rho_t(\boldsymbol{x})} \mathrm{d}\boldsymbol{z} \\
&\overset{(2)}{=} -\nabla_{\boldsymbol{x}} \cdot (\rho_t(\boldsymbol{x}) \boldsymbol{u}_t(\boldsymbol{x})) + \rho_t(\boldsymbol{x}) g_t(\boldsymbol{x})
\end{aligned}
$$

In (1), we use the continuity equation with source term of the conditional measure $\rho_t(\boldsymbol{x}|\boldsymbol{z})$ (4.2). In (2), we use the definition of marginal vector field and rate function (4.4). Thus, the marginal measure $\rho_t(\boldsymbol{x})$ satisfies the continuity equation with source term (3.2). The proof is completed. □

### A.3 PROOF OF THEOREM 4.2

**Theorem 4.2.** *If $\rho_t(\boldsymbol{x}) > 0$ for all $\boldsymbol{x} \in \mathcal{X}$ and $t \in [0,1]$, and $q(\boldsymbol{z})$ is independent of $\boldsymbol{x}$ and $t$, then $\mathcal{L}_{UFM}(\boldsymbol{\theta}, \boldsymbol{\phi}) = \mathcal{L}_{CUFM}(\boldsymbol{\theta}, \boldsymbol{\phi}) + C$, where $C$ is independent to $(\boldsymbol{\theta}, \boldsymbol{\phi})$. Thus they have identical gradients w.r.t $(\boldsymbol{\theta}, \boldsymbol{\phi})$, i.e.*

$$
\nabla_{\boldsymbol{\theta}, \boldsymbol{\phi}} \mathcal{L}_{\mathrm{UFM}}(\theta) = \nabla_{\boldsymbol{\theta}, \boldsymbol{\phi}} \mathcal{L}_{\mathrm{CUFM}}(\theta).
$$

*Proof.* First, we recall the two objective

$$
\mathcal{L}_{\mathrm{UFM}}(\boldsymbol{\theta}, \boldsymbol{\phi}) = \mathbb{E}_{t \sim \mathcal{U}[0,1]} \int_{\mathcal{X}} (\|\boldsymbol{v}_{\boldsymbol{\theta}}(\boldsymbol{x}, t) - \boldsymbol{u}_t(\boldsymbol{x})\|_2^2 + \kappa \|g_{\boldsymbol{\phi}}(\boldsymbol{x}, t) - g_t(\boldsymbol{x})\|_2^2) \rho_t(\boldsymbol{x}) \mathrm{d}\boldsymbol{x}
$$

$$
\mathcal{L}_{\mathrm{CUFM}}(\boldsymbol{\theta}, \boldsymbol{\phi}) = \mathbb{E}_{t \sim \mathcal{U}[0,1], \boldsymbol{z} \sim q(\boldsymbol{z}), \boldsymbol{x} \sim \tilde{\rho}_t(\boldsymbol{x}|\boldsymbol{z})} (\|\boldsymbol{v}_{\boldsymbol{\theta}}(\boldsymbol{x}, t) - \boldsymbol{u}_t(\boldsymbol{x}|\boldsymbol{z})\|_2^2 + \kappa \|g_{\boldsymbol{\phi}}(\boldsymbol{x}, t) - g_t(\boldsymbol{x}|\boldsymbol{z})\|_2^2) m_t(\boldsymbol{z})
$$

By direct computation, we have

$$
\begin{aligned}
\mathcal{L}_{\text{UFM}}(\boldsymbol{\theta}, \boldsymbol{\phi}) =& \mathbb{E}_{t \sim \mathcal{U}[0,1]} \int_{\mathcal{X}} (\|\boldsymbol{v_\theta}(\boldsymbol{x}, t)\|_2^2 - 2\langle \boldsymbol{v_\theta}(\boldsymbol{x}, t), \boldsymbol{u}_t(\boldsymbol{x}) \rangle) \rho_t(\boldsymbol{x}) \mathrm{d}\boldsymbol{x} \\
&+ \kappa \mathbb{E}_{t \sim \mathcal{U}[0,1]} \int_{\mathcal{X}} (\|g_\phi(\boldsymbol{x}, t)\|_2^2 - 2\langle g_\phi(\boldsymbol{x}, t), g_t(\boldsymbol{x}) \rangle) \rho_t(\boldsymbol{x}) \mathrm{d}\boldsymbol{x} + C \\
\overset{(1)}{=}& \mathbb{E}_{t \sim \mathcal{U}[0,1]} \{ \int_{\mathcal{X}} \int (\|\boldsymbol{v_\theta}(\boldsymbol{x}, t)\|_2^2 \rho_t(\boldsymbol{x}|\boldsymbol{z}) q(\boldsymbol{z}) \mathrm{d}\boldsymbol{z} \mathrm{d}\boldsymbol{x} \\
&- 2 \int_{\mathcal{X}} \langle \boldsymbol{v_\theta}(\boldsymbol{x}, t), \int \boldsymbol{u}_t(\boldsymbol{x}|\boldsymbol{z}) \rho_t(\boldsymbol{x}|\boldsymbol{z}) q(\boldsymbol{z}) \mathrm{d}\boldsymbol{z} \rangle \mathrm{d}\boldsymbol{x} \} \\
&+ \kappa \mathbb{E}_{t \sim \mathcal{U}[0,1]} \{ \int_{\mathcal{X}} \int (\|g_\phi(\boldsymbol{x}, t)\|_2^2 \rho_t(\boldsymbol{x}|\boldsymbol{z}) q(\boldsymbol{z}) \mathrm{d}\boldsymbol{z} \mathrm{d}\boldsymbol{x} \\
&- 2 \int_{\mathcal{X}} \langle g_\phi(\boldsymbol{x}, t), \int g_t(\boldsymbol{x}|\boldsymbol{z}) \rho_t(\boldsymbol{x}|\boldsymbol{z}) q(\boldsymbol{z}) \mathrm{d}\boldsymbol{z} \rangle \mathrm{d}\boldsymbol{x} \} + C \\
\overset{(2)}{=}& \mathbb{E}_{t \sim \mathcal{U}[0,1]} \int_{\mathcal{X}} \int (\|\boldsymbol{v_\theta}(\boldsymbol{x}, t)\|_2^2 - 2\langle \boldsymbol{v_\theta}(\boldsymbol{x}, t), \boldsymbol{u}_t(\boldsymbol{x}|\boldsymbol{z}) \rangle) m_t(\boldsymbol{z}) \tilde{\rho}_t(\boldsymbol{x}|\boldsymbol{z}) q(\boldsymbol{z}) \mathrm{d}\boldsymbol{z} \mathrm{d}\boldsymbol{x} \\
&+ \kappa \mathbb{E}_{t \sim \mathcal{U}[0,1]} \int_{\mathcal{X}} \int (\|g_\phi(\boldsymbol{x}, t)\|_2^2 - 2\langle g_\phi(\boldsymbol{x}, t), g_t(\boldsymbol{x}|\boldsymbol{z}) \rangle) m_t(\boldsymbol{z}) \tilde{\rho}_t(\boldsymbol{x}|\boldsymbol{z}) q(\boldsymbol{z}) \mathrm{d}\boldsymbol{z} \mathrm{d}\boldsymbol{x} + C \\
\overset{(3)}{=}& \mathbb{E}_{t \sim \mathcal{U}[0,1], \boldsymbol{z} \sim q(\boldsymbol{z}), \boldsymbol{x} \sim \tilde{\rho}_t(\boldsymbol{x}|\boldsymbol{z})} (\|\boldsymbol{v_\theta}(\boldsymbol{x}, t)\|_2^2 - 2\langle \boldsymbol{v_\theta}(\boldsymbol{x}, t), \boldsymbol{u}_t(\boldsymbol{x}) \rangle) m_t(\boldsymbol{z}) \\
&+ \kappa \mathbb{E}_{t \sim \mathcal{U}[0,1], \boldsymbol{z} \sim q(\boldsymbol{z}), \boldsymbol{x} \sim \tilde{\rho}_t(\boldsymbol{x}|\boldsymbol{z})} (\|g_\phi(\boldsymbol{x}, t)\|_2^2 - 2\langle g_\phi(\boldsymbol{x}, t), g_t(\boldsymbol{x}) \rangle) m_t(\boldsymbol{z}) + C \\
=& \mathcal{L}_{\text{CUFM}}(\boldsymbol{\theta}, \boldsymbol{\phi}) + C
\end{aligned}
$$

where $C$ represents a constant independent to $(\boldsymbol{\theta}, \boldsymbol{\phi})$. In (1) we expand the marginal measure $\rho_t(\boldsymbol{x})$ by its difinition (4.3). In (2), we decouple the measure $\rho_t(\boldsymbol{x}|\boldsymbol{z})$ into mass $m_t(\boldsymbol{z})$ and density $\tilde{\rho}_t(\boldsymbol{x}|\boldsymbol{z})$. In (3), we rewrite the integral in the form of expectation, moving the two probability density $q(\boldsymbol{z})$ and $\tilde{\rho}_t(\boldsymbol{x}|\boldsymbol{z})$ to the expectation operator.

Since the constant $C$ is independent to $(\boldsymbol{\theta}, \boldsymbol{\phi})$, we have

$$\nabla_{\boldsymbol{\theta}, \boldsymbol{\phi}} \mathcal{L}_{\text{UFM}}(\boldsymbol{\theta}, \boldsymbol{\phi}) = \nabla_{\boldsymbol{\theta}, \boldsymbol{\phi}} \mathcal{L}_{\text{CUFM}}(\boldsymbol{\theta}, \boldsymbol{\phi})$$

The proof is completed. □

### A.4 PROOF OF PROPOSITION 4.1

**Proposition 4.1.** *For CGMP defined as (4.5), if $m_t(\boldsymbol{z}) > 0$ for all $t$, then the conditional vector field and rate function are*

$$
\begin{cases}
\boldsymbol{u}_t(\boldsymbol{x}|\boldsymbol{z}) = \dfrac{\boldsymbol{\sigma}_t'(\boldsymbol{z})}{\boldsymbol{\sigma}_t(\boldsymbol{z})} (\boldsymbol{x} - \boldsymbol{\eta}_t(\boldsymbol{z})) + \boldsymbol{\eta}_t'(\boldsymbol{z}) \\
g_t(\boldsymbol{x}|\boldsymbol{z}) = \partial_t \ln m_t(\boldsymbol{z})
\end{cases}
\tag{A.23}
$$

*Proof.* The CMGP satisfies the following continuity equation with source term (4.2)

$$\partial_t \rho_t(\boldsymbol{x}|\boldsymbol{z}) + \nabla_{\boldsymbol{x}} \cdot (\boldsymbol{u}_t(\boldsymbol{x}|\boldsymbol{z}) \rho_t(\boldsymbol{x}|\boldsymbol{z})) = \rho_t(\boldsymbol{x}|\boldsymbol{z}) g_t(\boldsymbol{x}|\boldsymbol{z})$$

expand the conditional measure into mass and density, the equation become

$$m_t(\boldsymbol{z}) \partial_t \tilde{\rho}_t(\boldsymbol{x}|\boldsymbol{z}) + \partial_t m_t(\boldsymbol{z}) \tilde{\rho}_t(\boldsymbol{x}|\boldsymbol{z}) + m_t(\boldsymbol{z}) \nabla_{\boldsymbol{x}} \cdot (\boldsymbol{u}_t(\boldsymbol{x}|\boldsymbol{z}) \tilde{\rho}_t(\boldsymbol{x}|\boldsymbol{z})) = m_t(\boldsymbol{z}) \tilde{\rho}_t(\boldsymbol{x}|\boldsymbol{z}) g_t(\boldsymbol{x}|\boldsymbol{z}) \tag{A.24}$$

Since $m_t(\boldsymbol{z}) > 0$ for all $t$, we devide the both sides of the equation by $m_t(\boldsymbol{z})$ and get

$$\partial_t \tilde{\rho}_t(\boldsymbol{x}|\boldsymbol{z}) + \partial_t \ln m_t(\boldsymbol{z}) \tilde{\rho}_t(\boldsymbol{x}|\boldsymbol{z}) + \nabla_{\boldsymbol{x}} \cdot (\boldsymbol{u}_t(\boldsymbol{x}|\boldsymbol{z}) \tilde{\rho}_t(\boldsymbol{x}|\boldsymbol{z})) = \tilde{\rho}_t(\boldsymbol{x}|\boldsymbol{z}) g_t(\boldsymbol{x}|\boldsymbol{z}) \tag{A.25}$$

We further reorganize it to

$$\partial_t \tilde{\rho}_t(\boldsymbol{x}|\boldsymbol{z}) + \nabla_{\boldsymbol{x}} \cdot (\boldsymbol{u}_t(\boldsymbol{x}|\boldsymbol{z}) \tilde{\rho}_t(\boldsymbol{x}|\boldsymbol{z})) = \tilde{\rho}_t(\boldsymbol{x}|\boldsymbol{z}) (g_t(\boldsymbol{x}|\boldsymbol{z}) - \partial_t \ln m_t(\boldsymbol{z})) \tag{A.26}$$

Set $g_t(\boldsymbol{x}|\boldsymbol{z}) = \partial_t \ln m_t(\boldsymbol{z})$, the equation of the density $\tilde{\rho}_t(\boldsymbol{x}|\boldsymbol{z})$ reduces to the continuity equation

$$\partial_t \tilde{\rho}_t(\boldsymbol{x}|\boldsymbol{z}) + \nabla_{\boldsymbol{x}} \cdot (\boldsymbol{u}_t(\boldsymbol{x}|\boldsymbol{z})\tilde{\rho}_t(\boldsymbol{x}|\boldsymbol{z})) = 0 \tag{A.27}$$

Note that this equation is exactly the continuity equation of the conditional Gaussian path in conditional flow matching. Thus, the conditional vector field of CGMP shares the same form with the conditional vector field of conditional Gaussian path in balanced conditional flow matching, which is $\boldsymbol{u}_t(x|z) = \frac{\sigma'_t(\boldsymbol{z})}{\sigma_t(\boldsymbol{z})}(\boldsymbol{x} - \boldsymbol{\eta}_t(\boldsymbol{z})) + \boldsymbol{\eta}'_t(\boldsymbol{z})$. The proof is completed. $\qquad\square$

## A.5 PROOF OF PROPOSITION 4.2

**Proposition 4.2.** *Take traveling Gaussian (4.8) as CGMP, the marginal boundary measure $\rho_0(\boldsymbol{x}_0) \to \mu_0(\boldsymbol{x}_0)$), $\rho_1(\boldsymbol{x}_1) \to \mu_1(\boldsymbol{x}_1)$) as $\sigma \to 0$. Also, the flow generated by traveling Gaussian recover the optimal path of dynamical WFR problem (3.3), thus the marginal $\boldsymbol{u}_t(\boldsymbol{x})$ and $g_t(\boldsymbol{x})$ solve the dynamical WFR problem between $\mu_0$ and $\mu_1$.*

*Proof.* First, we expand the marginal measure

$$\rho_t(\boldsymbol{x}) = \int \rho_t(\boldsymbol{x}|\boldsymbol{z})q(\boldsymbol{z})\mathrm{d}\boldsymbol{z}$$

Set $t = 0$, we get the marginal boundary measure

$$\begin{aligned}
\rho_0(\boldsymbol{x}) &= \int \rho_0(\boldsymbol{x}|\boldsymbol{z})q(\boldsymbol{z})\mathrm{d}\boldsymbol{z} \\
&= \int_{\mathcal{X}^2} \tilde{\rho}_0(\boldsymbol{x}|\boldsymbol{z})m_0(\boldsymbol{x}_0, \boldsymbol{x}_1)q(\boldsymbol{x}_0, \boldsymbol{x}_1)\mathrm{d}\boldsymbol{x}_0\mathrm{d}\boldsymbol{x}_1 \\
&= \int_{\mathcal{X}^2} \mathcal{N}(\boldsymbol{x}|\boldsymbol{x}_0, \sigma^2\mathbf{I}) \cdot 1 \cdot \gamma_0(\boldsymbol{x}_0, \boldsymbol{x}_1)\mathrm{d}\boldsymbol{x}_0\mathrm{d}\boldsymbol{x}_1 \\
&= \int_{\mathcal{X}^2} \mathcal{N}(\boldsymbol{x}|\boldsymbol{x}_0, \sigma^2\mathbf{I})\gamma_0(\boldsymbol{x}_0, \boldsymbol{x}_1)\mathrm{d}\boldsymbol{x}_0\mathrm{d}\boldsymbol{x}_1 \\
&= \int_{\mathcal{X}} \mathcal{N}(\boldsymbol{x}|\boldsymbol{x}_0, \sigma^2\mathbf{I})\mu_0(\boldsymbol{x}_0)\mathrm{d}\boldsymbol{x}_0 \\
&= \mu_0 * \mathcal{N}(0, \sigma^2\mathbf{I})(\boldsymbol{x}) \overset{\sigma \to 0}{\to} \mu_0(\boldsymbol{x})
\end{aligned} \tag{A.28}$$

Set $t = 1$, we get the marginal boundary measure

$$\begin{aligned}
\rho_1(\boldsymbol{x}) &= \int \rho_1(\boldsymbol{x}|\boldsymbol{z})q(\boldsymbol{z})\mathrm{d}\boldsymbol{z} \\
&= \int_{\mathcal{X}^2} \tilde{\rho}_1(\boldsymbol{x}|\boldsymbol{z})m_1(\boldsymbol{x}_0, \boldsymbol{x}_1)q(\boldsymbol{x}_0, \boldsymbol{x}_1)\mathrm{d}\boldsymbol{x}_0\mathrm{d}\boldsymbol{x}_1 \\
&= \int_{\mathcal{X}^2} \mathcal{N}(\boldsymbol{x}|\boldsymbol{x}_1, \sigma^2)\frac{\gamma_1(\boldsymbol{x}_0, \boldsymbol{x}_1)}{\gamma_0(\boldsymbol{x}_0, \boldsymbol{x}_1)}\gamma_0(\boldsymbol{x}_0, \boldsymbol{x}_1)\mathrm{d}\boldsymbol{x}_0\mathrm{d}\boldsymbol{x}_1 \\
&= \int_{\mathcal{X}^2} \mathcal{N}(\boldsymbol{x}|\boldsymbol{x}_1, \sigma^2\mathbf{I})\gamma_1(\boldsymbol{x}_0, \boldsymbol{x}_1)\mathrm{d}\boldsymbol{x}_0\mathrm{d}\boldsymbol{x}_1 \\
&= \int_{\mathcal{X}} \mathcal{N}(\boldsymbol{x}|\boldsymbol{x}_1, \sigma^2\mathbf{I})\mu_1(\boldsymbol{x}_1)\mathrm{d}\boldsymbol{x}_1 \\
&= \mu_1 * \mathcal{N}(0, \sigma^2\mathbf{I})(\boldsymbol{x}) \overset{\sigma \to 0}{\to} \mu_1(\boldsymbol{x})
\end{aligned} \tag{A.29}$$

Thus the learned unbalanced flow matches the source measure $\mu_0$ and the target source $\mu_1$.

Next, we show that the induced marginal path $\{\rho_t\}$, which is the push forward of traveling Dirac on $\gamma_0(\boldsymbol{x}, \boldsymbol{y})$, indeed satisfies the geodesic of WFR metric. The WFR distance between the marginal measure $\rho_s$ and $\rho_t$ ($s < t$) can be estimated as follows.

$$\text{WFR}_\delta^2(\rho_s, \rho_t) = \inf_{(\gamma_0^{st}, \gamma_1^{st}) \in \Gamma(\rho_s, \rho_t)} \int_{\mathcal{X}^2} \text{WFR}_\delta^2(\gamma_0^{st}(\boldsymbol{x}, \boldsymbol{y})\delta_{\boldsymbol{x}}, \gamma_1^{st}(\boldsymbol{x}, \boldsymbol{y})\delta_{\boldsymbol{y}})\mathrm{d}\boldsymbol{x}\mathrm{d}\boldsymbol{y}$$

$$\overset{(1)}{\leq} \int_{\mathcal{X}^2} \text{WFR}_\delta^2(m_s(\boldsymbol{x}, \boldsymbol{y}, \gamma_0, \gamma_1)\delta_{\boldsymbol{x}_s(\boldsymbol{x}, \boldsymbol{y}, \gamma_0, \gamma_1)}, m_t(\boldsymbol{x}, \boldsymbol{y}, \gamma_0, \gamma_1)\delta_{\boldsymbol{x}_t(\boldsymbol{x}, \boldsymbol{y}, \gamma_0, \gamma_1)})\mathrm{d}\boldsymbol{x}\mathrm{d}\boldsymbol{y}$$

$$\overset{(2)}{=} \int_{\mathcal{X}^2} |t - s|^2 \text{WFR}_\delta^2(\gamma_0(\boldsymbol{x}, \boldsymbol{y})\delta_{\boldsymbol{x}}, \gamma_1(\boldsymbol{x}, \boldsymbol{y})\delta_{\boldsymbol{y}})\mathrm{d}\boldsymbol{x}\mathrm{d}\boldsymbol{y}$$

$$= |t - s|^2 \text{WFR}_\delta^2(\mu_0, \mu_1)$$

$$(A.30)$$

where $\boldsymbol{x}_t(\boldsymbol{x}, \boldsymbol{y}, \gamma_0, \gamma_1), m_t(\boldsymbol{x}, \boldsymbol{y}, \gamma_0, \gamma_1)$ are the position and mass at time $t$ induced by the traveling Dirac connecting $\gamma_0(\boldsymbol{x}, \boldsymbol{y})$ and $\gamma_1(\boldsymbol{x}, \boldsymbol{y})$. In (1) we replace the optimal semi-coupling between $\rho_s$ and $\rho_t$, namely $(\gamma_0^{st}, \gamma_1^{st})$, with a specific semi-coupling induced by the push forward of traveling Dirac on $\gamma_0(\boldsymbol{x}, \boldsymbol{y})$. In (2), we use the constant speed property of the traveling Dirac as a geodesic under WFR metric, which has been proved by (Chizat et al., 2018a) and (Liero et al., 2016). One can also check the constant speed property of traveling Dirac by direct calculation. Utilizing the inequality above, we have

$$\text{WFR}_\delta(\mu_0, \mu_1) \overset{(1)}{\leq} \text{WFR}_\delta(\mu_0, \rho_s) + \text{WFR}_\delta(\rho_s, \rho_t) + \text{WFR}_\delta(\rho_t, \mu_1)$$

$$\overset{(2)}{\leq} |s - 0|\,\text{WFR}_\delta(\mu_0, \mu_1) + |t - s|\,\text{WFR}_\delta(\mu_0, \mu_1) + |1 - t|\,\text{WFR}_\delta(\mu_0, \mu_1)$$

$$\overset{(3)}{=} \text{WFR}_\delta(\mu_0, \mu_1)$$

$$(A.31)$$

In (1) we use the triangle inequality of WFR, since it is a metric on $\mathcal{M}_+(\mathcal{X})$ (Chizat et al., 2018a). In (2) we use (A.30) on time intervals $[0, s]$, $[s, t]$ and $[t, 1]$. In (3) we simply use the fact that $0 \leq s < t \leq 1$. Thus all the inequalities in (A.31) are actually equalities. Specifically, we have

$$\text{WFR}_\delta(\rho_s, \rho_t) = |t - s|\text{WFR}_\delta(\mu_0, \mu_1) \qquad (A.32)$$

This indicates that the marginal measure path $\{\rho_t\}$ is a geodesic connecting the source $\mu_0$ and the target $\mu_1$ under the WFR metric. Thus, the corresponding marginal vector field $\boldsymbol{u}_t(\boldsymbol{x})$ and marginal rate function $g_t(\boldsymbol{x})$ solve the dynamical WFR problem. The proof is completed. $\qquad\square$

## A.6 PROOF OF PROPOSITION 4.3

**Proposition 4.3.** *For balanced cases, let $\delta \to \infty$ (3.3), then the travelling Dirac (3.5) reduces to uniform rectilinear motion with conserved mass, and the travelling Gaussian (4.8) simply reduces to $\boldsymbol{u}_t(x|z) = \boldsymbol{x}_1 - \boldsymbol{x}_0$, $g_t(\boldsymbol{x}|\boldsymbol{z}) \equiv 0$, thus we recover OT-CFM (Tong et al., 2024a).*

*Proof.* With conserved mass, and $\delta \to \infty$, the growth rate is forced to be 0 in order to make the cost finite. Thus, the minimizer of dynamical WFR (3.3) simply reduces to the minimizer of dynamical OT

$$\text{WFR}_\infty^2(\mu_0, \mu_1) = \inf_{\rho, g, \boldsymbol{u}} \int_0^1 \int_{\mathcal{X}} \frac{1}{2} \|\boldsymbol{u}(\boldsymbol{x}, t)\|_2^2 \rho_t(\boldsymbol{x})\mathrm{d}\boldsymbol{x}\mathrm{d}t$$

$$\text{s.t.} \qquad \partial_t \rho + \nabla_{\boldsymbol{x}} \cdot (\rho \boldsymbol{u}) = \rho g, \ g \equiv 0, \ \rho_0 = \mu_0, \ \rho_1 = \mu_1.$$

$$(A.33)$$

The corresponding OET (3.8) minimizer also reduces to the static OT minimizer

$$\begin{aligned}
\mathrm{WFR}_\delta^2(\mu_0, \mu_1) &= 2\delta^2 \inf_{\gamma \in \mathcal{M}_+(\mathcal{X}^2)} \{ \int_{\mathcal{X}^2} -2\ln\overline{\cos}(\frac{\|\boldsymbol{x}-\boldsymbol{y}\|_2}{2\delta})\gamma(\boldsymbol{x},\boldsymbol{y})\mathrm{d}\boldsymbol{x}\mathrm{d}\boldsymbol{y} \\
&\quad + \mathrm{KL}(\int_{\mathcal{X}} \gamma(\boldsymbol{x},\boldsymbol{y})\mathrm{d}\boldsymbol{y}\|\mu_0(\boldsymbol{x})) + \mathrm{KL}(\int_{\mathcal{X}} \gamma(\boldsymbol{x},\boldsymbol{y})\mathrm{d}\boldsymbol{x}\|\mu_1(\boldsymbol{y})) \} \\
&= 2\delta^2 \inf_{\gamma \in \mathcal{M}_+(\mathcal{X}^2)} \{ \int_{\mathcal{X}^2} -2\ln(1 - \frac{1}{2}(\frac{\|\boldsymbol{x}-\boldsymbol{y}\|_2}{2\delta})^2 + o(\delta^{-2}))\gamma(\boldsymbol{x},\boldsymbol{y})\mathrm{d}\boldsymbol{x}\mathrm{d}\boldsymbol{y} \\
&\quad + \mathrm{KL}(\int_{\mathcal{X}} \gamma(\boldsymbol{x},\boldsymbol{y})\mathrm{d}\boldsymbol{y}\|\mu_0(\boldsymbol{x})) + \mathrm{KL}(\int_{\mathcal{X}} \gamma(\boldsymbol{x},\boldsymbol{y})\mathrm{d}\boldsymbol{x}\|\mu_1(\boldsymbol{y})) \} \\
&= 2\delta^2 \inf_{\gamma \in \mathcal{M}_+(\mathcal{X}^2)} \{ \int_{\mathcal{X}^2} ((\frac{\|\boldsymbol{x}-\boldsymbol{y}\|_2}{2\delta})^2 + o(\delta^{-2}))\gamma(\boldsymbol{x},\boldsymbol{y})\mathrm{d}\boldsymbol{x}\mathrm{d}\boldsymbol{y} \qquad\text{(A.34)} \\
&\quad + \mathrm{KL}(\int_{\mathcal{X}} \gamma(\boldsymbol{x},\boldsymbol{y})\mathrm{d}\boldsymbol{y}\|\mu_0(\boldsymbol{x})) + \mathrm{KL}(\int_{\mathcal{X}} \gamma(\boldsymbol{x},\boldsymbol{y})\mathrm{d}\boldsymbol{x}\|\mu_1(\boldsymbol{y})) \} \\
&= \inf_{\gamma \in \mathcal{M}_+(\mathcal{X}^2)} \{ \int_{\mathcal{X}^2} \frac{1}{2}\|\boldsymbol{x}-\boldsymbol{y}\|_2^2\gamma(\boldsymbol{x},\boldsymbol{y})\mathrm{d}\boldsymbol{x}\mathrm{d}\boldsymbol{y} + o(1) \\
&\quad + 2\delta^2\,\mathrm{KL}(\int_{\mathcal{X}} \gamma(\boldsymbol{x},\boldsymbol{y})\mathrm{d}\boldsymbol{y}\|\mu_0(\boldsymbol{x})) + 2\delta^2\,\mathrm{KL}(\int_{\mathcal{X}} \gamma(\boldsymbol{x},\boldsymbol{y})\mathrm{d}\boldsymbol{x}\|\mu_1(\boldsymbol{y})) \} \\
&\overset{\delta\to\infty}{\to} \inf_{(\gamma,\gamma) \in \Gamma(\mu_0,\mu_1)} \int_{\mathcal{X}^2} \frac{1}{2}\|\boldsymbol{x}-\boldsymbol{y}\|_2^2\gamma(\boldsymbol{x},\boldsymbol{y})\mathrm{d}\boldsymbol{x}\mathrm{d}\boldsymbol{y}
\end{aligned}$$

The semi-coupling constructed from the OT coupling is just the OT coupling itself. Thus, we have $m_0 = m_1$ for each pair of points sampled from the semi-coupling. The mass is denoted as $m$ in the following.

The cost between two dirac (3.4) reduces to

$$\begin{aligned}
\mathrm{WFR\text{-}DD}_\delta^2(m\delta_{\boldsymbol{x}_0}, m\delta_{\boldsymbol{x}_1}) &= 2\delta^2(m + m - 2m\overline{\cos}(\frac{\|\boldsymbol{x}_0-\boldsymbol{x}_1\|_2}{2\delta})) \\
&= 4m\delta^2(1 - \cos(\frac{\|\boldsymbol{x}_0-\boldsymbol{x}_1\|_2}{2\delta})) \\
&= 4m\delta^2(\frac{1}{2}(\frac{\|\boldsymbol{x}_0-\boldsymbol{x}_1\|_2}{2\delta})^2 + o(\delta^{-2})) \qquad\text{(A.35)} \\
&= \frac{1}{2}m\|\boldsymbol{x}_0-\boldsymbol{x}_1\|_2^2 + o(1) \\
&\overset{\delta\to\infty}{\to} \frac{1}{2}m\|\boldsymbol{x}_0-\boldsymbol{x}_1\|_2^2
\end{aligned}$$

When $\delta \to \infty$, all pairs of $\boldsymbol{x}_0, \boldsymbol{x}_1$ satisfies $\|\boldsymbol{x}_0 - \boldsymbol{x}_1\|_2 < \pi\delta$. The travelling Dirac (3.5,3.6) reduces to uniform rectilinear motion with conserved mass as following

$$\begin{cases}
\tau = \tan(\frac{\|\boldsymbol{x}_1-\boldsymbol{x}_0\|_2}{2\delta}) = \frac{\|\boldsymbol{x}_1-\boldsymbol{x}_0\|_2}{2\delta} + o(\delta^{-1}) = o(1) \\[2mm]
\boldsymbol{\omega}_0 = 2\delta\tau\sqrt{\frac{m^2}{1+\tau^2}}\boldsymbol{l} = (\|\boldsymbol{x}_1-\boldsymbol{x}_0\|_2 + o(1))\frac{m}{\sqrt{1+o(1)}}\boldsymbol{l} \overset{\delta\to\infty}{\to} m\|\boldsymbol{x}_1-\boldsymbol{x}_0\|_2\boldsymbol{l} \\[2mm]
A = 2m - \frac{2m}{\sqrt{1+\tau^2}} = 2m(1 - \frac{1}{\sqrt{1+o(1)}}) \overset{\delta\to\infty}{\to} 0 \\[2mm]
B = m - \frac{m}{\sqrt{1+\tau^2}} = m(1 - \frac{1}{\sqrt{1+o(1)}}) \overset{\delta\to\infty}{\to} 0 \qquad\text{(A.36)} \\[2mm]
m(t) = At^2 - 2Bt + m \overset{\delta\to\infty}{\to} m \\[2mm]
\boldsymbol{u}(t) = \frac{\boldsymbol{\omega}_0}{m(t)} \overset{\delta\to\infty}{\to} \|\boldsymbol{x}_1-\boldsymbol{x}_0\|_2\boldsymbol{l} = \boldsymbol{x}_1 - \boldsymbol{x}_0
\end{cases}$$

Thus, the tavelling Gaussian (4.8) reduces to

$$\tilde{\rho}_t(\boldsymbol{x}|\boldsymbol{x}_0, \boldsymbol{x}_1) = \mathcal{N}(\boldsymbol{x}|\boldsymbol{x}_0 + t(\boldsymbol{x}_1 - \boldsymbol{x}_0), \sigma^2\mathbf{I})$$

The conditional velocity field and growth rate reduces to

$$\boldsymbol{u}_t(\boldsymbol{x}|\boldsymbol{z}) = \boldsymbol{x}_1 - \boldsymbol{x}_0, \quad g_t(\boldsymbol{x}|\boldsymbol{z}) = \partial_t \ln m = 0$$

The proof is completed. $\qquad\square$

### A.7 PROOF OF PROPOSITION 5.1

**Proposition 5.1.** *The solution to the multi-time WFR problem equation 5.1 is equivalent to the concatenation of the solutions of successive time points.*

*Proof.* Let $t_0 < t_1 < \cdots < t_K$ be the discrete time points. For each $k = 1, \ldots, K$, denote by $(\rho^{(k)}(\boldsymbol{x}, t), u^{(k)}(\boldsymbol{x}, t), g^{(k)}(\boldsymbol{x}, t))$ the optimal solution of the two-time WFR problem between $\mu_{t_{k-1}}$ and $\mu_{t_k}$:

$$\mathrm{WFR}_\delta^2(\mu_{t_{k-1}}, \mu_{t_k}) = \frac{t_k - t_{k-1}}{2} \inf_{\rho, u, g} \int_{t_{k-1}}^{t_k} \int_{\mathcal{X}} \left( \|u(\boldsymbol{x}, t)\|_2^2 + \delta^2 \|g(\boldsymbol{x}, t)\|_2^2 \right) \rho(\boldsymbol{x}, t) \, d\boldsymbol{x} \, dt \quad \text{(A.37)}$$

subject to

$$\partial_t \rho + \nabla_{\boldsymbol{x}} \cdot (\rho u) = \rho g, \quad \rho_{t_{k-1}} = \mu_{t_{k-1}}, \ \rho_{t_k} = \mu_{t_k}. \quad \text{(A.38)}$$

Now define the concatenated trajectory

$$\tilde{\rho}(\boldsymbol{x}, t) = \rho^{(k)}(\boldsymbol{x}, t), \quad \tilde{u}(\boldsymbol{x}, t) = u^{(k)}(\boldsymbol{x}, t), \quad \tilde{g}(\boldsymbol{x}, t) = g^{(k)}(\boldsymbol{x}, t), \quad \text{for } t \in [t_{k-1}, t_k].$$

Since $\rho_{t_k}^{(k)} = \mu_{t_k} = \rho_{t_k}^{(k+1)}$, the concatenated triple $(\tilde{\rho}, \tilde{u}, \tilde{g})$ is admissible for the multi-time WFR problem.

The corresponding action is

$$
\frac{t_K - t_0}{2} \int_{t_0}^{t_K} \int_{\mathcal{X}} \left( \|\tilde{\boldsymbol{u}}(\boldsymbol{x}, t)\|_2^2 + \delta^2 \|\tilde{g}(\boldsymbol{x}, t)\|_2^2 \right) \tilde{\rho}(\boldsymbol{x}, t) \, d\boldsymbol{x} \, dt
$$
$$
= \sum_{k=1}^K \frac{t_K - t_0}{t_k - t_{k-1}} \mathrm{WFR}_\delta^2(\mu_{t_{k-1}}, \mu_{t_k}). \quad \text{(A.39)}
$$

Hence

$$\mathrm{WFR}_\delta^2(\{\mu_{t_0}, \ldots, \mu_{t_K}\}) \leq \sum_{k=1}^K \frac{t_K - t_0}{t_k - t_{k-1}} \mathrm{WFR}_\delta^2(\mu_{t_{k-1}}, \mu_{t_k}). \quad \text{(A.40)}$$

To see that equality must hold, assume by contradiction that there exists another admissible solution $(\rho', u', g')$ such that

$$\frac{t_K - t_0}{2} \int_{t_0}^{t_K} \int_{\mathcal{X}} \left( \|u'(\boldsymbol{x}, t)\|_2^2 + \delta^2 \|g'(\boldsymbol{x}, t)\|_2^2 \right) \rho'(\boldsymbol{x}, t) \, d\boldsymbol{x} \, dt < \sum_{k=1}^K \frac{t_K - t_0}{t_k - t_{k-1}} \mathrm{WFR}_\delta^2(\mu_{t_{k-1}}, \mu_{t_k}).$$

Since the total action is a sum over the disjoint intervals $[t_{k-1}, t_k]$, this strict inequality implies that there must exist at least one index $k^*$ such that

$$\frac{t_K - t_0}{2} \int_{t_{k^*-1}}^{t_{k^*}} \int_{\mathcal{X}} \left( \|u'(\boldsymbol{x}, t)\|_2^2 + \delta^2 \|g'(\boldsymbol{x}, t)\|_2^2 \right) \rho'(\boldsymbol{x}, t) \, d\boldsymbol{x} \, dt < \frac{t_K - t_0}{t_{k^*} - t_{k^*-1}} \mathrm{WFR}_\delta^2(\mu_{t_{k^*-1}}, \mu_{t_{k^*}}).$$

Dividing both sides by the positive factor $\frac{t_K - t_0}{t_{k^*} - t_{k^*-1}}$ gives

$$\frac{t_{k^*} - t_{k^*-1}}{2} \int_{t_{k^*-1}}^{t_{k^*}} \int_{\mathcal{X}} \left( \|u'(\boldsymbol{x}, t)\|_2^2 + \delta^2 \|g'(\boldsymbol{x}, t)\|_2^2 \right) \rho'(\boldsymbol{x}, t) \, d\boldsymbol{x} \, dt < \mathrm{WFR}_\delta^2(\mu_{t_{k^*-1}}, \mu_{t_{k^*}}).$$

But this contradicts the definition of $\mathrm{WFR}_\delta^2(\mu_{t_{k^*-1}}, \mu_{t_{k^*}})$ as the minimal action between $\mu_{t_{k^*-1}}$ and $\mu_{t_{k^*}}$. Hence no admissible solution can have strictly smaller action than the concatenated one.

We conclude that the concatenated trajectory

$$(\rho^*(x, t), u^*(x, t), g^*(x, t)) = \bigcup_{k=1}^K (\rho^{(k)}, u^{(k)}, g^{(k)}), \quad t \in [t_{k-1}, t_k], \quad \text{(A.41)}$$

 achieves the infimum of the multi-time problem and is therefore optimal. $\qquad\square$

**Remark**: Proposition 5.1 holds true when the transport between adjacent time pairs $(t_{k-1}, t_k)$ is independent. However, in multi-marginal methods, such as MMFM (Rohbeck et al., 2025), 3MSBM (Theodoropoulos et al., 2025) and MMSFM (Lee et al., 2025), which enforce global coherence across all marginals, the dependencies between adjacent time pairs are introduced. As a result, the assumption of independence no longer holds, and Proposition 5.1 does not apply in the same form under a multi-marginal framework.

# B ADDITIONAL RESULTS

## B.1 EXPERIMENTAL DETAILS

The experiments were performed on a shared high-performance computing cluster with NVIDIA A100 GPU and 128 CPU cores. The architecture of the neural networks used to parameterize $\boldsymbol{v_\theta}(\boldsymbol{x}, t)$ and $g_\phi(\boldsymbol{x}, t)$ are Multilayer Perceptrons with 256 hidden channels and 5 layers. We use the LeakyReLU activation function. These networks were optimized using Pytorch (Paszke et al., 2017). The OET problem is solved using the Python Optimal Transport (POT) package (Flamary et al., 2021).

## B.2 EVALUATION METRICS

We use two metrics to evaluate model performance: the 1-Wasserstein distance ($\mathcal{W}_1$) to measure the similarity between normalized predicted and true distributions, and the Relative Mass Error (RME) to assess how well the model captures cell population growth.

The metrics are defined as:

$$\mathcal{W}_1(p, q) = \min_{\pi \in \Pi(p,q)} \int \|x - y\|_2 d\pi(x, y)$$

$$\text{RME}(t_k) = \frac{|\sum_i w_i(t_k) - n_k/n_0|}{n_k/n_0}$$

Here, $p$ and $q$ are the predicted and true empirical distributions, $w_i(t_k)$ is the inferred weight of cell $i$ at time $t_k$, and $n_k$ is the number of cells at time $k$.

To evaluate a model, we apply its learned dynamics to the initial cell data (where all weights are uniform, $w_i(0) = 1/n_0$) to predict cell trajectories and their corresponding weights at subsequent time points if the unbalanced term is included, otherwise the weights are kept uniform. We then compute the weighted $\mathcal{W}_1$ and RME by comparing these predictions to the observed cells from dataset. For some datasets, we also perform a hold-out experiment, training on all but one time point and using $\mathcal{W}_1$ to evaluate performance on the unseen time point. Specifically, to evaluate the performance of TIGON (Sha et al., 2024), we reimplemented their method to avoid numerical instabilities. For other methods, we mainly used their default settings if the datasets included were used in their original paper, otherwise we changed the network sizes of these models to ensure comparable parameter counts, tuned the training epochs and learning rates for these models to suit different datasets for fair comparisons. Specifically, for VGFM, as most datasets used in our work are included in their original codebase, we directly adopted their hyperparameters. For other simulation-free methods, we adjusted the training budget based on data complexity. For high-dimensional real-world datasets (such as EB, CITE, Mouse), we increased the training duration to 3,000 epochs and incorporated learning rate schedulers (Cosine Annealing) to ensure stability and convergence. For simulation-based methods (MIOFlow, TIGON and DeepRUOT), we adopted a distinct but unified hyperparameter configuration. To ensure a consistent and fair comparison among these methods, we set their hyperparameters to the values recommended by DeepRUOT, which represents the current state-of-the-art in this category.

### B.3 SIMULATION GENE DATA

We adopt the same dataset in Zhang et al. (2025a) that simulates a synthetic gene regulatory network. Its dynamics are governed by the following set of stochastic ordinary differential equations:

$$\frac{dX_1}{dt} = \frac{\alpha_1 X_1^2 + \beta}{1 + \alpha_1 X_1^2 + \gamma_2 X_2^2 + \gamma_3 X_3^2 + \beta} - \delta_1 X_1 + \eta_1 \xi_t$$

$$\frac{dX_2}{dt} = \frac{\alpha_2 X_2^2 + \beta}{1 + \gamma_1 X_1^2 + \alpha_2 X_2^2 + \gamma_3 X_3^2 + \beta} - \delta_2 X_2 + \eta_2 \xi_t$$

$$\frac{dX_3}{dt} = \frac{\alpha_3 X_3^2}{1 + \alpha_3 X_3^2} - \delta_3 X_3 + \eta_3 \xi_t$$

Here, $X_i(t)$ is the concentration of gene $i$. The model features a toggle switch between $X_1$ and $X_2$ (mutual inhibition and self-activation), which are further inhibited by $X_3$ and activated by an external signal $\beta$. The equations are parameterized by rates for self-activation ($\alpha_i$), inhibition ($\gamma_i$), and degradation ($\delta_i$), with an added stochastic noise term ($\eta_i \xi_t$).

The simulation included probabilistic cell division, where the instantaneous growth rate, $g$, depends on the expression of $X_2$ as $g = \alpha_g \frac{X_2^2}{1+X_2^2}$. Upon division, daughter cells inherit the parent's state plus a small perturbation. Data was recorded at time points [0, 8, 16, 24, 32] from two distinct initial cell populations. One initial population exhibits transition and growth, while another population is in a steady state. As shown in Figure 4, WFR-FM is able to recover the increasing growth rate of the population on the right side, eliminating false transition.

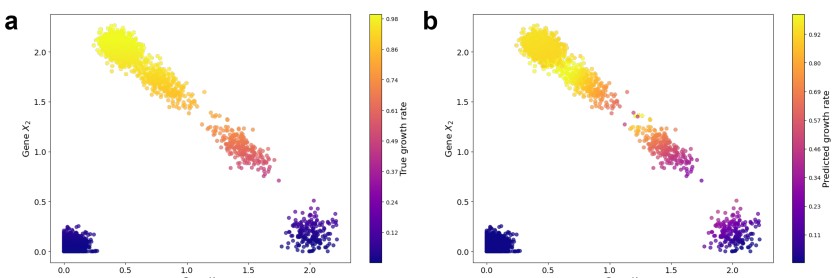

Figure 4: (a) True growth rate (b) Predicted growth rate by WFR-FM

**Choice of Growth Penalty $\delta$.** The parameter $\delta$ represents the penalty of growth in our algorithm. An excessively low $\delta$ would incorrectly incentivize solutions where cells tend to vanish and appear again to match the observed distributions, which may not be biologically plausible. In contrary, an overly high $\delta$ would excessively suppress cell proliferation. Therefore, a reasonable choice of $\delta$ is to balance transition and growth, which requires it to scale appropriately with the data. To evaluate the effect of $\delta$, we further conducted an ablation study on the simulation gene dataset. As shown in Table 5, both an overly high ($\delta$=3.0) and low ($\delta$=0.5) value result in sub-optimal performance. However, the results remain comparable for $\delta$ within a suitable range (1.0 to 2.0), demonstrating the robustness of our algorithm with respect to this parameter, provided it matches the scale of data. $\delta$ is set to 1.5 for the results of this dataset in the main text.

### B.4 DYGEN DATA

We adopt the same dataset as in Huguet et al. (2022); Wang et al. (2025) which was simulated by Dygen (Cannoodt et al., 2021) to model unbalanced bifurcation. The dataset contains 728 cells, and its dimension was reduced to 5 using PHATE (Moon et al., 2019). The data is challenging as it exhibits varying mass across different time points, and the number of cells in the lower branch is larger than the upper branch. As shown in Figure 5, WFR-FM accurately generates the bifurcating trajectories, and predicts higher growth rate in the lower branch, capturing the unbalancedness in the bifurcating process with the growth penalty set to 2. We also summarized the detailed quantitative results across different time points in Table 6. As shown, WFR-FM achieves the lowest $\mathcal{W}_1$ and RME in most of the time points.

Table 5: Sensitivity analysis for parameter $\delta$ on simulation gene dataset.

| Parameter | t=1 | | t=2 | | t=3 | | t=4 | |
|---|---|---|---|---|---|---|---|---|
| | $\mathcal{W}_1$ | RME | $\mathcal{W}_1$ | RME | $\mathcal{W}_1$ | RME | $\mathcal{W}_1$ | RME |
| $\delta = 0.5$ | 0.021 | 0.003 | 0.036 | 0.016 | 0.036 | 0.022 | 0.033 | 0.027 |
| $\delta = 1.0$ | 0.020 | 0.001 | 0.021 | 0.002 | 0.018 | 0.003 | 0.015 | 0.001 |
| $\delta = 1.5$ | 0.021 | 0.001 | 0.021 | 0.001 | 0.018 | 0.002 | 0.016 | 0.001 |
| $\delta = 2.0$ | 0.022 | 0.000 | 0.025 | 0.000 | 0.025 | 0.003 | 0.032 | 0.000 |
| $\delta = 2.5$ | 0.023 | 0.001 | 0.042 | 0.003 | 0.025 | 0.006 | 0.042 | 0.002 |
| $\delta = 3.0$ | 0.027 | 0.001 | 0.044 | 0.003 | 0.051 | 0.006 | 0.030 | 0.004 |

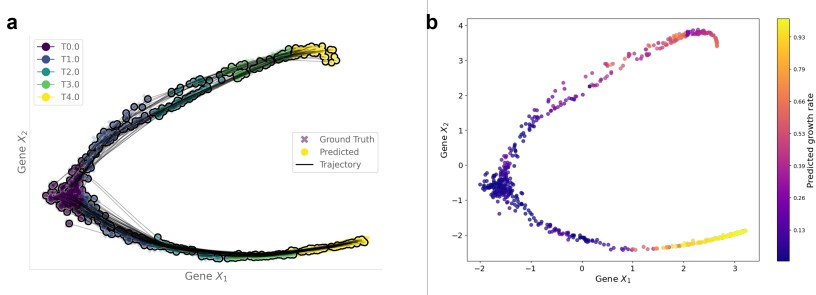

Figure 5: (a) Learned trajectories (b) growth rate on the dygen dataset

Table 6: Comparison of method performance over time on the Dygen dataset. Best results are in bold.

| Method | t=1 | | t=2 | | t=3 | | t=4 | |
|---|---|---|---|---|---|---|---|---|
| | $\mathcal{W}_1$ | RME | $\mathcal{W}_1$ | RME | $\mathcal{W}_1$ | RME | $\mathcal{W}_1$ | RME |
| MMFM | 0.574 | — | 1.704 | — | 1.499 | — | 1.706 | — |
| Metric FM | 0.892 | — | 2.347 | — | 2.030 | — | 1.799 | — |
| SF2M | 0.637 | — | 1.266 | — | 1.415 | — | 1.790 | — |
| MIOFlow | 0.420 | — | 0.640 | — | 1.537 | — | 1.263 | — |
| TIGON | 0.446 | 0.033 | 0.584 | 0.060 | 0.415 | 0.023 | 0.603 | 0.071 |
| DeepRUOT | 0.454 | 0.011 | 0.481 | 0.070 | 0.870 | 0.104 | 0.688 | 0.074 |
| Var-RUOT | 0.315 | 0.128 | 0.548 | 0.336 | 0.630 | 0.222 | 0.593 | 0.023 |
| UOT-FM | 0.652 | 0.008 | 0.780 | 0.077 | 1.252 | 0.090 | 2.130 | 0.213 |
| VGFM | 0.335 | **0.001** | 0.312 | 0.073 | 1.109 | 0.041 | 0.634 | 0.033 |
| WFR-FM | **0.110** | 0.003 | **0.098** | **0.007** | **0.211** | **0.008** | **0.121** | **0.002** |

### B.5 Gaussian Mixture Data

We adopt the 1000D Gaussian Mixture Data from Wang et al. (2025). Following their setup, an initial distribution of 500 samples (100 from an upper Gaussian, 400 from an lower Gaussian) and a final distribution of 1,400 samples (1,000 from the upper, 200 from each lower two Gaussians) were generated. This models cell proliferation in the upper region, which serves as a suitable example to evaluate the algorithm's ability of modeling unbalanced dynamics in high dimensional settings. As shown in Figure 6, WFR-FM correctly models the higher growth rate in the upper region, and yields correct trajectories with growth penalty as 1.4, showing its effectiveness.

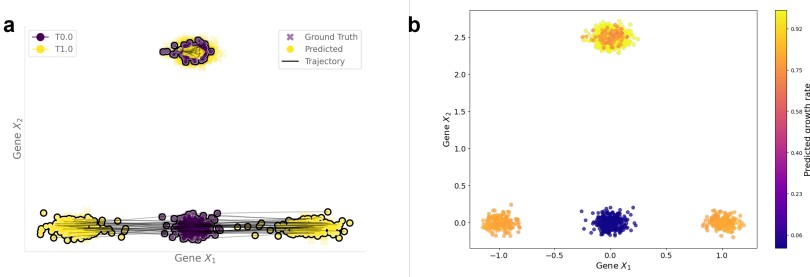

Figure 6: (a) Learned trajectories (b) growth rate on the Gaussian 1000D dataset

### B.6 Epithelial-Mesenchymal Transition Data

We adopt the dataset that captures the epithelial-mesenchymal transition (EMT) in A549 lung cancer cells from Cook & Vanderhyden (2020). The dataset includes samples collected at four distinct time points throughout this process. The dimension was reduced to 10 using an AutoEncoder (AE) as in Sha et al. (2024). As shown in Table 7, WFR-FM achieves the best performance in both distribution matching task and mass matching task across all time points with growth penalty set to 2. We plotted the learned trajectory and growth rate in Figure 7. As shown, WFR-FM predicts higher growth rate in initial and intermediate stages, which is in line with the results predicted by DeepRUOT (Zhang et al., 2025a) as these cells exhibit enhanced stemness.

Table 7: Comparison of method performance over time on the 10D EMT dataset. Best results are in bold.

| Method | t=1 | | t=2 | | t=3 | |
|---|---|---|---|---|---|---|
| | $\mathcal{W}_1$ | RME | $\mathcal{W}_1$ | RME | $\mathcal{W}_1$ | RME |
| MMFM | 0.2576 | — | 0.2874 | — | 0.3102 | — |
| Metric FM | 0.2605 | — | 0.2971 | — | 0.3050 | — |
| SF2M | 0.2566 | — | 0.2811 | — | 0.2900 | — |
| MIOFlow | 0.2439 | — | 0.2665 | — | 0.2841 | — |
| TIGON | 0.2433 | 0.002 | 0.2661 | 0.003 | 0.2847 | **0.001** |
| DeepRUOT | 0.2902 | **0.001** | 0.3193 | 0.011 | 0.3291 | 0.002 |
| Var-RUOT | 0.2540 | 0.075 | 0.2670 | 0.014 | 0.2683 | 0.041 |
| UOT-FM | 0.2538 | 0.002 | 0.2696 | 0.013 | 0.2771 | 0.010 |
| VGFM | 0.2350 | 0.016 | 0.2420 | 0.011 | 0.2450 | 0.018 |
| WFR-FM | **0.2099** | **0.001** | **0.2272** | **0.002** | **0.2346** | **0.001** |

### B.7 Embryoid Bodies Data

We adopt the dataset of 16,819 cells from a 27-day differentiation time course of human embryoid bodies (EB), sampled at five time points (Moon et al., 2019). This experimental system models early embryonic development. The original high-dimensional gene expression data was preprocessed by reducing it to lower dimensions using Principal Component Analysis (PCA) as in Wang et al. (2025).

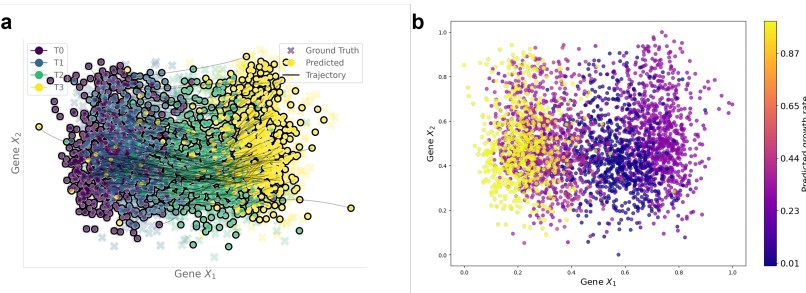

Figure 7: (a) Learned trajectories (b) growth rate on the EMT dataset

**Effects of Mini-batch WFR-OET on WFR-FM.** We employed the mini-batch WFR-OET to accelerate computation on this dataset. The effects of different batch sizes are summarized in Table 8. The mini-batch approach significantly reduces runtime compared to the full-batch method. we noticed that the runtime exhibits a non-monotonic trend, which may due to the higher number of OET calculations required for smaller batches. Performance generally improves as the batch size increases, eventually becoming comparable to the full-batch result. Therefore, we selected the batch size of 2000 as the default value to achieve a favorable balance between efficiency and accuracy.

Table 8: Sensitivity analysis for batch size of mini-batch WFR-OET on the 100D EB dataset.

| Batch Size | t=1 | | t=2 | | t=3 | | t=4 | | Time (s) |
|---|---|---|---|---|---|---|---|---|---|
| | $\mathcal{W}_1$ | RME | $\mathcal{W}_1$ | RME | $\mathcal{W}_1$ | RME | $\mathcal{W}_1$ | RME | |
| 500 | 9.980 | 0.010 | 11.075 | 0.008 | 11.567 | 0.008 | 12.850 | 0.004 | 159 |
| 1000 | 9.956 | 0.009 | 11.076 | 0.008 | 11.609 | 0.006 | 12.783 | 0.002 | 129 |
| 1500 | 9.937 | 0.009 | 11.046 | 0.008 | 11.583 | 0.005 | 12.771 | 0.001 | 125 |
| 2000 | 9.941 | 0.009 | 11.040 | 0.006 | 11.516 | 0.008 | 12.664 | 0.005 | 131 |
| 3000 | 9.914 | 0.009 | 11.049 | 0.009 | 11.529 | 0.006 | 12.687 | 0.002 | 139 |
| 4000 | 9.920 | 0.009 | 11.050 | 0.007 | 11.538 | 0.006 | 12.654 | 0.003 | 159 |
| w/o mini-batch | 9.926 | 0.009 | 11.045 | 0.007 | 11.545 | 0.006 | 12.657 | 0.002 | 269 |

**Effects of Data Dimensions on WFR-FM.** To evaluate the scability of WFR-FM, we compare the performance of WFR-FM with other methods on 5 dimensions (Table 10) with growth penalty set to 2, 50 dimensions (Table 11) with growth penalty set to 25 and 100 dimensions (Table 12) with growth penalty set to 35. WFR-FM outperforms other methods in in most of the time points on all three datasets, indicating the consistent performance of WFR-FM when scaling to higher dimensions with little increase in computational time (Table 9).

## B.8 CITE-SEQ DATA

We adopt the CITE-seq dataset from Lance et al. (2022) which contains 31,240 cells collected at four time points. Following the preprocess procedure in Wang et al. (2025), we selected only the gene expression matrix and projected it into 50 dimensions using PCA. We summarized the performance of distribution matching and mass matching in Table 13. We adopt the mini-batch strategy with the batch size of 2000 for this dataset, and set growth penalty to 30. As shown, WFR-FM achieves the

Table 9: Computational time across different dimensions on the EB dataset.

| Dimension | Time (s) |
|---|---|
| 5 | 123 |
| 50 | 126 |
| 100 | 131 |

Table 10: Comparison of method performance over time on the 5D EB dataset. Best results are in bold.

| Method | t=1 | | t=2 | | t=3 | | t=4 | |
|---|---|---|---|---|---|---|---|---|
| | $\mathcal{W}_1$ | RME | $\mathcal{W}_1$ | RME | $\mathcal{W}_1$ | RME | $\mathcal{W}_1$ | RME |
| MMFM | 0.477 | — | 0.554 | — | 0.781 | — | 0.872 | — |
| Metric FM | 0.449 | — | 0.552 | — | 0.583 | — | 0.597 | — |
| SF2M | 0.556 | — | 0.715 | — | 0.750 | — | 0.650 | — |
| MIOFlow | 0.442 | — | 0.585 | — | 0.651 | — | 0.670 | — |
| TIGON | 0.386 | **0.002** | 0.502 | 0.015 | 0.602 | 0.021 | 0.600 | 0.027 |
| DeepRUOT | 0.386 | 0.005 | 0.497 | 0.017 | 0.591 | 0.021 | 0.585 | 0.030 |
| Var-RUOT | 0.416 | 0.111 | 0.486 | 0.144 | 0.509 | 0.054 | 0.511 | 0.022 |
| UOT-FM | 0.544 | 0.032 | 0.670 | 0.029 | 0.729 | 0.016 | 0.852 | 0.041 |
| VGFM | 0.402 | 0.046 | 0.494 | 0.018 | 0.525 | 0.035 | 0.573 | 0.021 |
| WFR-FM | **0.324** | 0.003 | **0.401** | **0.001** | **0.431** | **0.005** | **0.510** | **0.005** |

Table 11: Comparison of method performance over time on the 50D EB dataset. Best results are in bold.

| Method | t=1 | | t=2 | | t=3 | | t=4 | |
|---|---|---|---|---|---|---|---|---|
| | $\mathcal{W}_1$ | RME | $\mathcal{W}_1$ | RME | $\mathcal{W}_1$ | RME | $\mathcal{W}_1$ | RME |
| MMFM | 9.124 | — | 10.474 | — | 11.022 | — | 11.480 | — |
| Metric FM | 8.506 | — | 9.795 | — | 10.621 | — | 12.042 | — |
| SF2M | 9.247 | — | 10.882 | — | 11.650 | — | 12.154 | — |
| MIOFlow | 8.447 | — | 9.229 | — | 9.436 | — | 10.123 | — |
| TIGON | 8.433 | 0.067 | 9.275 | 0.022 | 9.802 | 0.179 | 10.148 | 0.101 |
| DeepRUOT | 8.169 | **0.003** | 9.049 | 0.038 | 9.378 | 0.088 | 9.733 | **0.004** |
| Var-RUOT | 9.442 | 0.128 | 9.709 | 0.081 | 10.482 | 0.031 | 10.735 | 0.030 |
| UOT-FM | 8.717 | 0.063 | 10.858 | 0.009 | 11.813 | 0.022 | 12.733 | 0.018 |
| VGFM | 7.951 | 0.089 | 8.747 | 0.042 | 9.244 | 0.019 | **9.620** | 0.044 |
| WFR-FM | **7.664** | 0.008 | **8.659** | **0.006** | **9.182** | **0.004** | 9.914 | **0.004** |

highest accuracy in matching distributions across all time points, while maintaining a mass matching accuracy comparable to other method that model unbalancedness.

## B.9 MOUSE HEMATOPOIESIS DATA

We adopt the mouse blood hematopoiesis dataset (Mouse) from Weinreb et al. (2020). We selected 49,302 cells with lineage tracing information collected at three time points. The original gene expression data was reduced to 50 dimensions using PCA. This dataset was chosen for its substantial cell population growth, making it suitable for scalability testing. We sub-sampled the dataset to

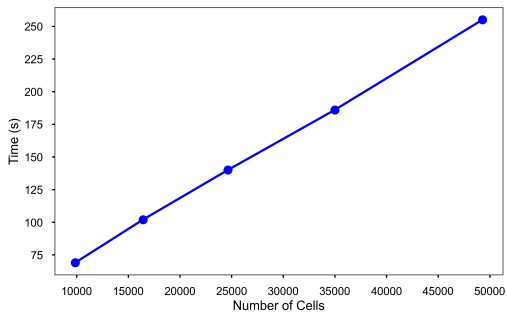

Figure 8: Scaling of runtime with respect to cell number

Table 12: Comparison of method performance over time on the 100D EB dataset. Best results are in bold.

| Method | t=1 | | t=2 | | t=3 | | t=4 | |
|--------|------|-----|------|-----|------|-----|------|-----|
| | $\mathcal{W}_1$ | RME | $\mathcal{W}_1$ | RME | $\mathcal{W}_1$ | RME | $\mathcal{W}_1$ | RME |
| MMFM | 11.460 | — | 13.879 | — | 14.441 | — | 14.907 | — |
| Metric FM | 10.806 | — | 12.348 | — | 13.622 | — | 16.801 | — |
| SF2M | 11.333 | — | 12.982 | — | 13.718 | — | 14.945 | — |
| MIOFlow | 11.387 | — | 12.331 | — | 11.905 | — | 12.908 | — |
| TIGON | 10.547 | 0.014 | 12.926 | 0.052 | 13.897 | 0.107 | 14.945 | 0.096 |
| DeepRUOT | 10.256 | 0.002 | 11.103 | 0.074 | 11.529 | 0.136 | **12.406** | 0.047 |
| Var-RUOT | 11.746 | 0.091 | 12.237 | 0.024 | 12.957 | 0.150 | 13.335 | 0.074 |
| UOT-FM | 10.757 | 0.056 | 12.799 | 0.037 | 13.761 | 0.044 | 15.657 | 0.022 |
| VGFM | 10.313 | 0.048 | 11.278 | 0.035 | 11.703 | 0.028 | 12.637 | 0.066 |
| WFR-FM | **9.941** | **0.009** | **11.040** | **0.006** | **11.516** | **0.008** | 12.664 | **0.005** |

Table 13: Comparison of method performance over time on the 50D CITE dataset. Best results are in bold.

| Method | t=1 | | t=2 | | t=3 | |
|--------|------|-----|------|-----|------|-----|
| | $\mathcal{W}_1$ | RME | $\mathcal{W}_1$ | RME | $\mathcal{W}_1$ | RME |
| MMFM | 33.971 | — | 36.854 | — | 43.721 | — |
| Metric FM | 28.314 | — | 28.617 | — | 33.212 | — |
| SF2M | 29.543 | — | 32.655 | — | 36.265 | — |
| MIOFlow | 28.290 | — | 28.524 | — | **32.230** | — |
| TIGON | 28.196 | 0.186 | 27.921 | 0.545 | 32.846 | 0.653 |
| DeepRUOT | 28.245 | 0.168 | 27.908 | 0.525 | 32.950 | 0.634 |
| Var-RUOT | 30.219 | 0.331 | 32.702 | 0.325 | 40.613 | 0.486 |
| UOT-FM | 33.531 | **0.009** | 32.795 | 0.046 | 49.751 | 0.097 |
| VGFM | 29.449 | 0.020 | 29.722 | 0.057 | 33.752 | **0.001** |
| WFR-FM | **27.831** | 0.043 | **27.478** | **0.045** | 34.784 | 0.022 |

different cell numbers, and plotted the scaling of time with respect to cell numbers in Figure 8. As shown, the training time scales linearly with cell numbers, in line with other competing methods, indicating that WFR-FM can be applied to larger datasets. We adopt the mini-batch strategy with the batch size of 2000 for this dataset, and set growth penalty to 15. As shown in Table 14, WFR-FM outperforms other methods by a large margin, indicating the effectiveness of WFR-FM in simultaneously modeling cell proliferation and differentiation.

## B.10 UNBALANCEDNESS IN REAL SINGLE-CELL DATASETS

We illustrate the proportions of cell types (or clusters) across different time points in several real single-cell RNA-seq datasets (Moon et al., 2019; Lance et al., 2022; Weinreb et al., 2020). As shown in Figure 9, the relative abundances of different cell types vary substantially across time points, indicating a pronounced unbalancedness in these datasets. We hypothesize that this widespread imbalance may partly explain why methods based on unbalanced optimal transport can perform better than those assuming balanced distributions. Furthermore, previous study (Sha et al., 2024) has also reported the impact of cell proliferation on dynamics in the EMT (Cook & Vanderhyden, 2020) and Mouse (Weinreb et al., 2020) datasets, highlighting the importance of accounting for growth and apoptosis in trajectory inference.

Table 14: Comparison of method performance over time on the 50D Mouse dataset. Best results are in bold.

| Method | t=1 | | t=2 | |
|---|---|---|---|---|
| | $\mathcal{W}_1$ | RME | $\mathcal{W}_1$ | RME |
| MMFM | 7.647 | — | 10.156 | — |
| Metric FM | 7.788 | — | 11.449 | — |
| SF2M | 8.217 | — | 11.086 | — |
| MIOFlow | 6.313 | — | 6.746 | — |
| TIGON | 6.140 | 0.382 | 6.973 | 0.326 |
| DeepRUOT | 6.052 | 0.062 | 6.757 | 0.041 |
| Var-RUOT | 7.951 | 0.131 | 10.862 | 0.154 |
| UOT-FM | 8.114 | 0.035 | 9.170 | **0.011** |
| VGFM | 6.274 | 0.076 | 6.796 | 0.070 |
| WFR-FM | **5.486** | **0.012** | **6.211** | **0.011** |

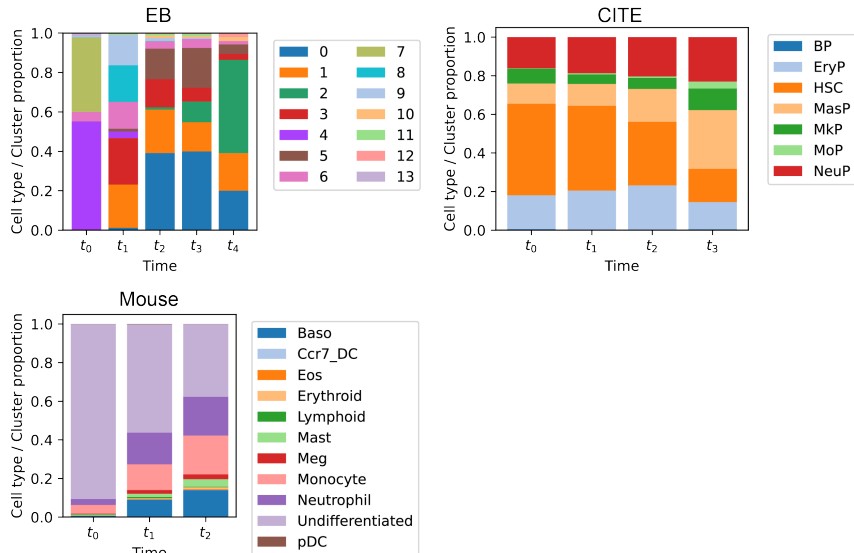

Figure 9: Proportions of cell types (or clusters) across different time points for three real scRNA-seq datasets. Top left: EB dataset Moon et al. (2019); top right: CITE dataset Lance et al. (2022); bottom left: Mouse dataset Weinreb et al. (2020).

# C   RELATION TO OTHER UNBALANCED FLOW MATCHING METHODS

## C.1   RELATION TO UOT-FM.

UOT-FM (Eyring et al., 2024) was proposed to extend flow matching to the unbalanced setting. Its formulation replaces the balanced OT map in OT-FM (Tong et al., 2024a) with an unbalanced OT map obtained from the classical static unbalanced OT problem:

$$
\begin{aligned}
\text{UOT}(\mu_0, \mu_1) := \inf_{\gamma \in \mathcal{M}_+(\mathcal{X}^2)} \Big\{ &\int_{\mathcal{X}^2} c(\boldsymbol{x}, \boldsymbol{y}) \gamma(\boldsymbol{x}, \boldsymbol{y}) \mathrm{d}\boldsymbol{x} \mathrm{d}\boldsymbol{y} \\
&+ \lambda_1 \, \text{KL}(\int_{\mathcal{X}} \gamma(\boldsymbol{x}, \boldsymbol{y}) \mathrm{d}\boldsymbol{y} \| \mu_0(\boldsymbol{x})) + \lambda_2 \, \text{KL}(\int_{\mathcal{X}} \gamma(\boldsymbol{x}, \boldsymbol{y}) \mathrm{d}\boldsymbol{x} \| \mu_1(\boldsymbol{y})) \Big\}
\end{aligned}
\tag{C.1}
$$

This provides a natural way to relax mass conservation in flow matching by leveraging static unbalanced OT formulation. In comparison, WFR-FM is built on the dynamic WFR geometry, which directly yields a continuous-time interpretation of the unbalanced dynamics. In addition, while UOT-FM focuses on learning the transport vector field, our formulation simultaneously learns both a displacement field and a growth rate function, enabling us to model single-cell dynamics where mass and state evolve jointly over time.

## C.2   RELATION TO UFM.

UFM (Corso et al., 2025) can be viewed as a natural generalization of UOT-FM. Rather than restricting the coupling to the optimal plan obtained from static unbalanced OT, UFM allows for arbitrary couplings between source and target samples. In particular, UFM focuses on arbitrary couplings $\gamma$ and provides both a formal and empirical characterization of the associated objectives. The training objective augments vector field regression with additional terms that penalize mass mismatch:

$$
\begin{aligned}
L_{\text{UFM}}(\gamma, \theta) = \alpha \mathbb{E}_{t, (\mathbf{x}, \mathbf{y}) \sim \gamma} \Big[ &\| v_t(\mathbf{x}_t; \theta) - u_t(\mathbf{x}_t | (\mathbf{x}_0, \mathbf{x}_1)) \|_2^2 \Big] \\
&+ \lambda_1 \, \text{KL}(\int_{\mathcal{X}} \gamma(\boldsymbol{x}, \boldsymbol{y}) \mathrm{d}\boldsymbol{y} \| \mu_0(\boldsymbol{x})) + \lambda_2 \, \text{KL}(\int_{\mathcal{X}} \gamma(\boldsymbol{x}, \boldsymbol{y}) \mathrm{d}\boldsymbol{x} \| \mu_1(\boldsymbol{y})) \Big\}
\end{aligned}
\tag{C.2}
$$

This formulation extends unbalanced flow matching beyond the static UOT setting by simultaneously learning transport dynamics and accounting for marginal discrepancies. Compared to this, WFR-FM is derived from the dynamic WFR geometry, which provides a principled interpretation of both displacement and growth within a unified continuous-time framework.

## C.3   RELATION TO CAO ET AL. (2025)

Cao et al. (2025) adopt a formulation that is conceptually similar to UOT-FM, but specializes it to the concrete application of pansharpening. On top of the basic unbalanced flow matching framework, they introduce additional design choices tailored to the data characteristics of this imaging task. While effective for pansharpening, such a task-specific adaptation faces challenges in generalizing to the reconstruction of single-cell dynamics.

## C.4   RELATION TO VGFM

VGFM (Wang et al., 2025) is a related work to ours as both approaches extend OT-based flow matching to the unbalanced setting by jointly learning a transport vector field and a growth rate. VGFM is built upon a semi-constrained OT formulation:

$$
\begin{aligned}
\text{SC-UOT}(\mu_0, \mu_1) := \inf_{\gamma \geq 0} &\int_{\mathcal{X}^2} \| \mathbf{x} - \mathbf{y} \|_2^2 \gamma(\mathbf{x}, \mathbf{y}) \mathrm{d}\mathbf{x} \mathrm{d}\mathbf{y} + + \lambda_1 \, \text{KL}(\int_{\mathcal{X}} \gamma(\boldsymbol{x}, \boldsymbol{y}) \mathrm{d}\boldsymbol{y} \| \mu_0(\boldsymbol{x})) \\
\text{subject to} \quad &\int_{\mathcal{X}} \gamma(\boldsymbol{x}, \boldsymbol{y}) \mathrm{d}\boldsymbol{x} = \mu_1(\boldsymbol{y}),
\end{aligned}
\tag{C.3}
$$

This formulation leads to a dynamic interpretation where mass variation and transport are separated into two distinct phases—first birth–death dynamics without transport, followed by transport without mass change. For single-cell applications, this corresponds to cells proliferating or dying

without altering gene expression in one phase, and changing gene expression without proliferation or apoptosis in the other. However, such separation remains biologically challenging to justify. To address this, VGFM introduces additional heuristics to couple transport and growth; however, this modification moves the method away from a strict OT formulation. Furthermore, VGFM is primarily employed as a pretraining strategy and still requires simulating ODE trajectories (albeit fewer than without pretraining) to achieve strong performance.

By contrast, WFR-FM is derived directly from the dynamic WFR metric, couples transport and growth within a principled geometric framework, and achieves fully simulation-free training without any additional ODE integration.

## D    INTUITION AND LIMITATIONS OF MINI-BATCH OT

In WFR-FM, we adopt a mini-batch approach to approximate the optimal transport coupling during training. The convergence of mini-batch optimal transport to the true coupling remains an open problem. For the balanced case, some theoretical and numerical studies have explored how batch size and sample number affect convergence (e.g., Sommerfeld et al. (2019)), however, to our knowledge, analogous results for the unbalanced setting are not yet available. Intuitively, one can view an independent coupling as the extreme case of mini-batch OT with batch size 1, while full-batch OT corresponds to the other extreme. Selecting an appropriate batch size thus reflects a trade-off between computational efficiency and adherence to the optimal transport principle. Importantly, even with mini-batch OT, our method remains a flow-matching approach: it continues to match the marginals to the data distributions. Although mini-batch OT may result in slight deviations from the globally optimal transport plan, it provides a practical and effective way to learn the underlying transport and growth dynamics.

In addition, we conducted an ablation study on the choice of OT batch size (see Table 8). The results show that our method is robust to the selection of batch size: performance remains stable across a wide range of batch sizes, indicating that mini-batch OT does not introduce significant sensitivity in practice.

## E    PSEUDOCODE FOR INFERENCE IN WFR-FM

---
**Algorithm 2** WFR-FM Inference Workflow

---
**Require:** Initial samples $\{\boldsymbol{x}_0^{(i)}\}_{i=1}^N \sim \mu_{t_0}$, trained vector field $\boldsymbol{v_\theta}(\boldsymbol{x}, t)$, trained growth rate $g_\phi(\boldsymbol{x}, t)$, target time $t_{\text{end}}$, integration step size $\Delta t$.

1: Initialize $\boldsymbol{x}^{(i)} \leftarrow \boldsymbol{x}_0^{(i)}, m^{(i)} \leftarrow 1$ for $i = 1, \ldots, N$
2: $t \leftarrow t_0$
3: **while** $t < t_{\text{end}}$ **do**
4:     $\Delta t_{\text{step}} \leftarrow \min(\Delta t, t_{\text{end}} - t)$        $\triangleright$ ensure last step reaches exactly $t_{\text{end}}$
5:     **for** $i = 1 \rightarrow N$ **do**
6:         $\boldsymbol{u} \leftarrow \boldsymbol{v_\theta}(\boldsymbol{x}^{(i)}, t)$
7:         $g \leftarrow g_\phi(\boldsymbol{x}^{(i)}, t)$
8:         Update position: $\boldsymbol{x}^{(i)} \leftarrow \boldsymbol{x}^{(i)} + \boldsymbol{u}\Delta t_{\text{step}}$
9:         Update mass: $m^{(i)} \leftarrow m^{(i)} \cdot \exp(g\Delta t_{\text{step}})$
10:     **end for**
11:     $t \leftarrow t + \Delta t_{\text{step}}$
12: **end while**
13: **return** Evolved samples $\{\boldsymbol{x}^{(i)}, m^{(i)}\}_{i=1}^N$

---

## F    THE USE OF LARGE LANGUAGE MODELS (LLMS)

We used LLMs solely for the purpose of improving grammar and clarity of the text in manuscript.

