# OpenReview forum: "WFR-FM: Simulation-Free Dynamic Unbalanced Optimal Transport"
_ICLR.cc/2026/Conference — ICLR 2026 Poster_

### Official Review · Reviewer_Tvju · 2025-10-22

**Soundness:** 3
**Presentation:** 3
**Contribution:** 3
**Rating:** 8
**Confidence:** 4

**Summary:**

This paper tackles the important and challenging problem of modeling continuous dynamics from sparse, unbalanced temporal snapshots --- a problem of significant interest in single-cell biology, where cell proliferation and apoptosis are key factors. The authors propose WFR Flow Matching (WFR-FM), a novel, simulation-free framework that unifies dynamic unbalanced optimal transport (under the WFR geometry) with flow matching.

The core technical contribution is a method that, unlike standard flow matching, simultaneously regresses both a transport vector field for particle displacement AND a scalar growth rate function for birth-death dynamics. The authors provide theoretical guarantees that this approach exactly recovers WFR geodesics. Empirically, the paper demonstrates that WFR-FM achieves state-of-the-art performance on several single-cell trajectory inference benchmarks, outperforming a wide range of baselines in accuracy, efficiency, and stability.

**Strengths:**

1. **Clarity**: The paper is well-written, clear, and easy to follow. The problem, the proposed method, and its theoretical underpinnings are all explained well.
2. **Thorough Related Work**: The authors provide a comprehensive review of related work with broader context of optimal transport, flow matching, and single-cell trajectory inference.
3. **Empirical Validation**: The experimental evaluation is a significant strength. The authors compare WFR-FM against a wide and appropriate range of state-of-the-art baselines (including both balanced and unbalanced OT/FM methods).
4. **Scalability and Efficiency Analysis**: The inclusion of detailed scalability analysis, including training time and memory usage (e.g., Figure 3), is good.
5. **Straightforward generalisation to multiple time points**: The method's design, which naturally extends to multiple time points by concatenating successive intervals (Proposition 5.1), is practical and well-justified for real-world snapshot data.

**Weaknesses:**

1.  **Positioning wrt to VGFM**: The paper currently relegates the detailed comparison with VGFM to Appendix C.4. While this appendix clearly articulates the important differences, VGFM appears to be the most direct competitor that also jointly models velocity and growth (and as reviewer believe was the first extension of flow matching to that). For fairness and clarity, a more prominent discussion of both the similarities and differences in the main body would be beneficial. This would also help readers better attribute the significant performance gains (e.g., in Table 1) to the novel methodological contributions of WFR-FM rather than potentially superior hyperparameter tuning
2.  **Baseline Tuning**: The experimental results show some performance variations among baseline methods (e.g., MFM/SF2M vs. MIOFlow) that appear inconsistent across different experiments. The text lacks a detailed description of the hyperparameter tuning procedure used for these baselines. Without this, it is difficult to ascertain whether the reported baseline results represent their optimal performance.

**Questions:**

1. **Practicality of Weighted Samples**: Regarding the practical application: As I understand from the inference workflow (Algorithm 2), the number of simulated particles/samples remains fixed from $t_0$. The 'growth' is modeled by increasing the mass $m^{(i)}$ of each particle. Is this interpretation correct? If so, the final output is a set of weighted points, where the number of points equals the initial number. How does this weighted-point representation practically benefit downstream analysis in single-cell biology, beyond simply matching benchmark distribution measures (like $\mathcal{W}_1$ and RME)?

2. **Coupling of Growth and Transport**: In the governing ODEs (Eq. 3.1), the evolution of the position $\phi_t$ (driven by $u_t$) and the mass $m_t$ (driven by $g_t$) appear to be two separate, parallel processes, evaluated along the same path. Does the learned growth term $g_t$ (or the WFR-OET coupling) implicitly change the learned transport field $u_t$ for a given sample, compared to a model (like UOT-FM) that uses a similar unbalanced OT coupling but only models the vector field $u_t$?

3. **Baseline Tuning**: This question refers to 2nd weakness.  Could the authors elaborate on the hyperparameter tuning procedure used for the baseline methods? Specifically, what steps were taken to ensure a fair and robust comparison, and to ensure that each baseline was performing optimally on these specific datasets?

---

> ### Author Response · Authors · 2025-11-27
> **Response to Reviewer Tvju (1/3)**
>
> > **Summary**:
>
> > This paper tackles the important and challenging problem of modeling continuous dynamics from sparse, unbalanced temporal snapshots --- a problem of significant interest in single-cell biology, where cell proliferation and apoptosis are key factors. The authors propose WFR Flow Matching (WFR-FM), a novel, simulation-free framework that unifies dynamic unbalanced optimal transport (under the WFR geometry) with flow matching.
>
> > The core technical contribution is a method that, unlike standard flow matching, simultaneously regresses both a transport vector field for particle displacement AND a scalar growth rate function for birth-death dynamics. The authors provide theoretical guarantees that this approach exactly recovers WFR geodesics. Empirically, the paper demonstrates that WFR-FM achieves state-of-the-art performance on several single-cell trajectory inference benchmarks, outperforming a wide range of baselines in accuracy, efficiency, and stability.
>
> - We sincerely thank the reviewer for the careful reading and positive assessment of our work. We are glad that you found our formulation of WFR Flow Matching clear and technically sound, and we appreciate your recognition of the simultaneous regression of transport and growth rate fields as well as the theoretical guarantees for recovering WFR geodesics. We are encouraged that you found our method to achieve strong empirical performance, outperforming a range of baselines in accuracy, efficiency, and stability. Your thoughtful summary and acknowledgment of the contributions of our work are greatly appreciated.
> - Below you will find our detailed point-by-point responses. We have revised our manuscript accordingly to address the comments, and all changes made to the manuscript are indicated in **blue** throughout the revised main text and appendix. Thank you once again for your insightful review and encouraging feedback.
>
>
> > **Strengths**:
> > 1. **Clarity**: The paper is well-written, clear, and easy to follow. The problem, the proposed method, and its theoretical underpinnings are all explained well.
> > 2. **Thorough Related Work**: The authors provide a comprehensive review of related work with broader context of optimal transport, flow matching, and single-cell trajectory inference.
> > 3. **Empirical Validation**: The experimental evaluation is a significant strength. The authors compare WFR-FM against a wide and appropriate range of state-of-the-art baselines (including both balanced and unbalanced OT/FM methods).
> > 4. **Scalability and Efficiency Analysis**: The inclusion of detailed scalability analysis, including training time and memory usage (e.g., Figure 3), is good.
> > 5. **Straightforward generalisation to multiple time points**: The method's design, which naturally extends to multiple time points by concatenating successive intervals (Proposition 5.1), is practical and well-justified for real-world snapshot data.
>
> We sincerely thank the reviewer for highlighting these strengths of our work. We are glad that you found the paper well-written and clear, and that the problem formulation, proposed method, and theoretical foundations were easy to follow. We appreciate your acknowledgment of our comprehensive review of related work and the broader context of optimal transport, flow matching, and single-cell trajectory inference. It is also encouraging that you recognized the thorough empirical evaluation of WFR-FM against a wide range of state-of-the-art baselines, as well as the detailed scalability and efficiency analysis presented in Figure 3. Finally, we are pleased that you found the extension of our method to multiple time points, as formalized in Proposition 5.1, to be practical and well-justified. Your thoughtful feedback is greatly appreciated and reinforces the value of our contributions.

---

> > ### Author Response · Authors · 2025-11-27
> > **Response to Reviewer Tvju (2/3)**
> >
> > > **Weaknesses**:
> > > 1. **Positioning wrt to VGFM**: The paper currently relegates the detailed comparison with VGFM to Appendix C.4. While this appendix clearly articulates the important differences, VGFM appears to be the most direct competitor that also jointly models velocity and growth (and as reviewer believe was the first extension of flow matching to that). For fairness and clarity, a more prominent discussion of both the similarities and differences in the main body would be beneficial. This would also help readers better attribute the significant performance gains (e.g., in Table 1) to the novel methodological contributions of WFR-FM rather than potentially superior hyperparameter tuning.
> >
> > Thank you for the thoughtful comment. We have added a more prominent and thorough discussion of VGFM in the main text (**Introduction** and **Related Work**) to improve clarity and fairness. We fully agree that VGFM represents an important advance in the joint modeling of velocity and growth, and it has opened an influential line of research on unbalanced flow matching that our work builds upon, which has been now included in the related work section. In the revised manuscript, we also explicitly present a clearer discussion of the conceptual, theoretical, and algorithmic differences between the two methods. This expanded discussion highlights the unique capabilities contributed by WFR-FM within this emerging research direction.
> >
> > > 2. **Baseline Tuning**: The experimental results show some performance variations among baseline methods (e.g., MFM/SF2M vs. MIOFlow) that appear inconsistent across different experiments. The text lacks a detailed description of the hyperparameter tuning procedure used for these baselines. Without this, it is difficult to ascertain whether the reported baseline results represent their optimal performance.
> >
> > Thank you for raising this important point. Ensuring a fair comparison is critical to our evaluation. We would like to clarify our tuning procedure and highlight a specific update we made regarding the CITE dataset to ensure fairness.
> >
> > 1. **Re-evaluation of the CITE Dataset**. During a careful examination of our experiments for the rebuttal, we identified an inconsistency between our data preprocessing pipeline for the CITE-seq dataset and the one used in the original Metric FM paper. To address this and ensure fair comparison, we re-processed the CITE data following Metric FM's exact preprocessing pipeline and re-ran all experiments on this dataset. As shown in the Table below (also Updated table 3 in the revised manuscript), WFR-FM still achieves the best performance, confirming the robustness of our method. Also, we would like to clarify that the original Metric FM codebase evaluates performance step-by-step (simulating from $t_{i-1}$ to predict $t_i$). To ensure a consistent comparison across all methods, we standardized the evaluation to simulate trajectories starting from the initial time point ($t=0$) to the target time point for all baselines, including Metric FM.
> >
> > | Method | CITE (50D) |
> > | :--- | :--- |
> > | MMFM | 38.521 |
> > | Metric FM | 37.342 |
> > | SF2M | 38.333 |
> > | MIOFlow | 39.574 |
> > | TIGON | 38.159 |
> > | DeepRUOT | 37.892 |
> > | Var-RUOT | 38.393 |
> > | UOT-FM | 38.649 |
> > | VGFM | 37.386 |
> > | **WFR-FM (Ours)** | **37.221** |
> >
> > 2. **General Hyperparameter Tuning Strategy**. Regarding the general tuning for other baselines, we adopted the following strategies:
> >     - For VGFM, as most datasets used in our work are included in their original codebase, we directly adopted their hyperparameters.
> >     - For other simulation-free methods, we adjusted the training budget based on data complexity. For high-dimensional real-world datasets (such as EB, CITE, Mouse), we increased the training duration to 3,000 epochs and incorporated learning rate schedulers (Cosine Annealing) to ensure stability and convergence.
> >     - For simulation-based methods (MIOFlow, TIGON and DeepRUOT), we adopted a distinct but unified hyperparameter configuration. To ensure a consistent and fair comparison among these methods, we set their hyperparameters to the values recommended by DeepRUOT, which represents the current state-of-the-art in this category.
> >
> > We have included these detailed discussions in the updated **Appendix B.2**.

---

> > > ### Author Response · Authors · 2025-11-27
> > > **Response to Reviewer Tvju (3/3)**
> > >
> > > > **Questions**:
> > > > 1. **Practicality of Weighted Samples**: Regarding the practical application: As I understand from the inference workflow (Algorithm 2), the number of simulated particles/samples remains fixed from $t_0$. The 'growth' is modeled by increasing the mass $m^{(i)}$ of each particle. Is this interpretation correct? If so, the final output is a set of weighted points, where the number of points equals the initial number. How does this weighted-point representation practically benefit downstream analysis in single-cell biology, beyond simply matching benchmark distribution measures (like $W_1$ and $RME$)?
> > >
> > > Thank you for the insightful comment. Your understanding is completely correct. The final mass $m^{(i)}$ of each particle represents the number of cells at that location, although it may not be an integer. In another work called stVCR [1] that models single-cell trajectories using dynamic unbalanced optimal transport, cell proliferation and apoptosis were simulated during inference via birth-death processes (even though the training still relies on weighted particles), which ensures that the final cell counts are integers. Their inference implementation could be naturally applied to the current work. Meanwhile, we would like to point out that when ignoring stochastic noise or cell-cell interactions, there is no fundamental difference between the two inference approaches, since simulating with weighted particles directly corresponds to the expected cell counts obtained from the birth-death process, and the major difference was merely about visualization effects. Using weighted particles rather than explicitly simulating cell divisions and deaths also allows for stable training and straightforward backpropagation. For biological downstream analyses, we primarily rely on the inferred cell growth rates $g$ instead of the inferred weights.
> > >
> > >
> > > > 2. **Coupling of Growth and Transport**: In the governing ODEs (Eq. 3.1), the evolution of the position $\phi_t$ (driven by $u_t$) and the mass $m_t$ (driven by $g_t$) appear to be two separate, parallel processes, evaluated along the same path. Does the learned growth term $g_t$ (or the WFR-OET coupling) implicitly change the learned transport field $u_t$ for a given sample, compared to a model (like UOT-FM) that uses a similar unbalanced OT coupling but only models the vector field $u_t$?
> > >
> > > Thank you for the insightful question. The evolution of $\phi_t(x)$ and $m_t(x)$ in Eq. 3.1 are not independent. Although the ODE is written in a parallel form, $m_t(x)$ is driven by $g_t$, and $g_t$ itself depends on $\phi_t(x)$. This creates an intrinsic coupling between the position and mass dynamics, so the learned growth term $g_t$ and the transport field $u_t$ influence each other. Furthermore, from Eq. 3.5, which provides an analytical solution for Dirac-to-Dirac transport, we can see that the evolution of the weighted particles follows an impulse-conserving form (i.e., $u(t) m(t)$ is constant). This implies that particles with larger mass move more slowly, while lighter particles move faster, which is fundamentally different from UOT-FM.
> > >
> > > > 3. **Baseline Tuning**: This question refers to 2nd weakness. Could the authors elaborate on the hyperparameter tuning procedure used for the baseline methods? Specifically, what steps were taken to ensure a fair and robust comparison, and to ensure that each baseline was performing optimally on these specific datasets?
> > >
> > > Thank you for raising this important point. To ensure fairness, we re-evaluated results on the CITE dataset, and detailed our hyperparameter tuning strategy for different types of baselines as in our response to Weakness 2 (**Appendix B.2** in the revised manuscript).
> > >
> > >
> > >
> > > **References**
> > >
> > > [1] Peng Q, Zhou P, Li T. stVCR: Spatiotemporal dynamics of single cells. bioRxiv, 2024: 2024.06. 02.596937.

---

> > > > ### Comment · Reviewer_Tvju · 2025-11-27
> > > >
> > > > I thank the authors for their detailed response. I am grateful for the following updates and clarifications: **VGFM Discussion**, **CITE Re-evaluation**, **Hyperparameters**,  **Dynamics** which i believe further strenghten the work.
> > > >
> > > >
> > > > I have maintained my positive acceptance score. Very interesting work!

---

### Official Review · Reviewer_V2He · 2025-10-31

**Soundness:** 3
**Presentation:** 2
**Contribution:** 3
**Rating:** 6
**Confidence:** 2

**Summary:**

The paper introduces WFR-FM, a model for learning continuous cellular dynamics from single-cell snapshot data when population sizes change over time. WFR-FM is based on the Wasserstein–Fisher–Rao (WFR) geometry, which couples transport of cell states with birth–death mass changes. To learn these dynamics efficiently, the authors combine WFR with flow matching, producing a simulation-free training objective. WFR-FM learns two components of cell dynamics jointly: a velocity field that transports cell states through gene expression space, and a growth rate field that models local changes in cell population density. The method constructs conditional particle trajectories inspired by traveling WFR geodesics, and trains a neural network to regress both velocity and growth directly, without integrating differential equations during training. The primary use of WFR-FM is to infer trajectories from time-series scRNA-seq data, particularly when cell counts shift between time points. The authors show that WFR-FM reconstructs developmental and differentiation dynamics more accurately and efficiently than ODE-based dynamic optimal transport methods, while capturing biologically meaningful patterns of proliferation and apoptosis.

**Strengths:**

Overall, I enjoyed the paper. I think it is clean, well-written, and the math flows quite well. Broadly, I also find the research timely and compelling. While existing publications have already studied how to approach unbalancedness in flow matching, the present paper provides quite a deep formulation of the problem with updated probability paths and proper justifications. The model also seems to be working well in applied scenarios.

**Weaknesses:**

Overall, my evaluation of the paper is positive (see the score). However, some aspects where I would suggest some improvements. I will elaborate on them below.

- **Minor:** L136 -- change the citation to `\citep` for Tong et al.
- In Eq. 3.3, $\delta$ is undefined. Since later the same symbol is used to define Dirac measures, I think it would be better to differentiate the meaning of $\delta$ in different scenarios.
- **Consistency in velocity notation:** In 3.5, the velocity notation is $x'$, but this is not used anywhere else in the text, where velocity is mostly expressed in terms of differentials.
- L195: "They" does not seem like a correct pronoun here. What's the subject?
- Eq. 3.8: The OET misses the constraints.
- In Eq. 4.7: Refer to $\tau$ as the $\tau$ described above.
- The way the traveling Gaussian is introduced could benefit from more elaboration. Especially, you could add one sentence explaining why the mass $m_0=1$ (hence, cause it's a relative mass change along the flow).
- The concept figure could be bigger with a more exhaustive caption.
- In the paper, I would probably describe the RME metric with a few sentences, and then refer to the appendix. At the moment, there is no clear intuition about it in the main text, and this breaks the readability of the experimental section a bit. Also, acronyms are not expanded in the main. Similarly, I would add a few sentences about where the presence of unbalancedness plays a role in the chosen datasets and why accounting for it is expected to produce a better performance.
- Is the action in Tab. 2 calculated over multiple trajectories? If yes, can you please add error bars? Along these lines, I think that most of the tables would benefit from error bars, as the results are very close.

**Questions:**

- L 308: Could you provide more intuition on the unit velocity here and why it is equivalent to flow matching by Tong?
- Did the authors inspect any theoretical guarantees of their batchwise approach (just asking for an intuition)? How detrimental is it to use this approach for learning the true coupling?
- Why would unbalanced OT work better on the EMT, EB, and cite datasets? I expect it to work on par with other OT and FM models, but why would it overcome them? Is it related to some aspect of umbalancedness?

---

> ### Author Response · Authors · 2025-11-27
> **Response to Reviewer V2He (1/4)**
>
> > **Summary**:
>
> > The paper introduces WFR-FM, a model for learning continuous cellular dynamics from single-cell snapshot data when population sizes change over time. WFR-FM is based on the Wasserstein–Fisher–Rao (WFR) geometry, which couples transport of cell states with birth–death mass changes. To learn these dynamics efficiently, the authors combine WFR with flow matching, producing a simulation-free training objective. WFR-FM learns two components of cell dynamics jointly: a velocity field that transports cell states through gene expression space, and a growth rate field that models local changes in cell population density. The method constructs conditional particle trajectories inspired by traveling WFR geodesics, and trains a neural network to regress both velocity and growth directly, without integrating differential equations during training. The primary use of WFR-FM is to infer trajectories from time-series scRNA-seq data, particularly when cell counts shift between time points. The authors show that WFR-FM reconstructs developmental and differentiation dynamics more accurately and efficiently than ODE-based dynamic optimal transport methods, while capturing biologically meaningful patterns of proliferation and apoptosis.
>
> > **Strengths**:
>
> > Overall, I enjoyed the paper. I think it is clean, well-written, and the math flows quite well. Broadly, I also find the research timely and compelling. While existing publications have already studied how to approach unbalancedness in flow matching, the present paper provides quite a deep formulation of the problem with updated probability paths and proper justifications. The model also seems to be working well in applied scenarios.
>
> - Thank you very much for your thoughtful and encouraging assessment of our work. We are glad to hear that you found the paper clearly written and mathematically well-motivated, and that the formulation of unbalanced flow matching and WFR geometry came across as timely and compelling. We also appreciate your recognition of the depth of our formulation, the updated probability paths, and the theoretical justifications we provide. It is also encouraging that you found WFR-FM to perform well in applied scenarios, and we appreciate your positive evaluation of its practical performance. Your feedback is greatly appreciated and motivates our continued development in this direction.
> - Below you will find our detailed point-by-point responses. We have revised our manuscript accordingly to address the comments, and all changes made to the manuscript are indicated in **blue** throughout the revised main text and appendix. Thank you once again for your insightful review and encouraging feedback.
>
>
> > **Weaknesses**:
>
> > Overall, my evaluation of the paper is positive (see the score). However, some aspects where I would suggest some improvements. I will elaborate on them below.
>
> > - **Minor**: L136 -- change the citation to \citep for Tong et al.
>
> Thank you for pointing this out. We have corrected this citation and we appreciate your careful reading.
>
>
> > - In Eq. 3.3, $\delta$ is undefined. Since later the same symbol is used to define Dirac measures, I think it would be better to differentiate the meaning of $\delta$ in different scenarios.
>
> Thank you for your careful reading and helpful comment. We have clarified in the text the meaning of $\delta$ in Eq. 3.3 to indicate that it is the WFR hyperparameter, in order to avoid confusion with its later use for Dirac measures.
>
>
> > - **Consistency in velocity notation**: In 3.5, the velocity notation is $x'$, but this is not used anywhere else in the text, where velocity is mostly expressed in terms of differentials.
>
> Thank you for pointing this out. We have revised the notation and replaced $x(t)$ with $u(t)$ in Equation 3.5 to ensure consistency throughout the paper.
>
> > - L195: "They" does not seem like a correct pronoun here. What's the subject?
>
> Thank you for the comment. We have clarified the pronoun usage. Since $(\gamma_0, \gamma_1)$ denotes a pair but is referred to as a single semi-coupling, we have replaced “they” with “it” to avoid ambiguity.
>
>
> > - Eq. 3.8: The OET misses the constraints.
>
> Thank you for the comment. The OET constraints are indeed included in Eq. 3.8: the optimization is taken over $\gamma\in\mathcal{M}_+(\mathcal{X}^2)$, which specifies the admissible set of transport plans. This constraint is explicitly stated in Eq. 3.8.
>
> > - In Eq. 4.7: Refer to $\tau$ as the $\tau$ described above.
>
> Thank you for pointing this out. To avoid confusion with the $\tau$ used in the traveling Dirac in Eq. 3.6, we have replaced $\tau$ in Eq. 4.7 (and in other similar places) with $\kappa$.

---

> ### Author Response · Authors · 2025-11-27
> **Response to Reviewer V2He (2/4)**
>
> > - The way the traveling Gaussian is introduced could benefit from more elaboration. Especially, you could add one sentence explaining why the mass $m_0=1$ (hence, cause it's a relative mass change along the flow).
>
> Thank you for the suggestion. We have revised the sentence and added an explanation clarifying that $m_0=1$ is chosen because $m_t$ describes the mass evolution relative to the initial mass, so setting the initial mass to 1 provides a consistent reference.
>
> > - The concept figure could be bigger with a more exhaustive caption.
>
> Thank you for the helpful suggestion. In the revised version, we have substantially expanded the caption to provide a more informative description of the concept figure. In addition, we have improved the overall visual quality of the figure to make it clearer and more readable.
>
> > - In the paper, I would probably describe the RME metric with a few sentences, and then refer to the appendix. At the moment, there is no clear intuition about it in the main text, and this breaks the readability of the experimental section a bit. Also, acronyms are not expanded in the main. Similarly, I would add a few sentences about where the presence of unbalancedness plays a role in the chosen datasets and why accounting for it is expected to produce a better performance.
>
> Thank you for the suggestion.
> - We have revised the main text to briefly introduce the evaluation metrics before referring to the appendix. In particular, we now clarify that the 1-Wasserstein distance ($\mathcal{W}_1$​) is computed between the normalized predicted and true distributions, and that the Relative Mass Error (RME) evaluates how well the model captures population growth. All acronyms are expanded at first use.
> - In the main text (Experiment **Q3**), we have added a few sentences explaining why unbalanced methods are expected to perform better on these datasets. In addition, we provide a more detailed discussion and empirical evidence of unbalancedness in the datasets in **Appendix B.10**.
>
> > - Is the action in Tab. 2 calculated over multiple trajectories? If yes, can you please add error bars? Along these lines, I think that most of the tables would benefit from error bars, as the results are very close.
>
> Thank you for your careful observation.
>
> - Regarding the Action metric in Table 2, we calculate it by simulating trajectories for all data points at $t=0$, computing the action for each point, and then reporting the mean value across the entire population.
> - Regarding the error bars:
>     - For WFR-FM: Our inference process is governed by deterministic ODEs defined by the learned velocity field $v_{\theta}$ and growth rate $g_{\phi}$. Consequently, for a fixed set of initial points at $t=0$, the generated trajectories and their corresponding action values are deterministic, resulting in a standard deviation of zero.
>     - For Baselines: We note that several baseline methods (e.g.,DeepRUOT, Var-RUOT, SF2M) involve intrinsic randomness. Following your suggestion, the results reported are averaged over 5 inference runs.
>
> We agree that including error bars for these stochastic methods improves the rigor of the comparison. Therefore, in the revised manuscript, we have added error bars (standard deviation over multiple runs) for the baselines that exhibit stochasticity in the main text.

---

> > ### Author Response · Authors · 2025-11-27
> > **Response to Reviewer V2He (3/4)**
> >
> > > **Questions**:
> > > - L 308: Could you provide more intuition on the unit velocity here and why it is equivalent to flow matching by Tong?
> >
> > Thank you for pointing out our typo. The velocity is actually not unit, but $x_1-x_0$. We have corrected it in the manuscript. We also organized the claim in L308 into proposition 4.3, and supplemented a proof in **Appendix A.6**.
> >
> > Intuitively, though our framework focuses on dealing with the unbalancedness, the balanced case can be naturally included as a limiting case. In balanced cases, the mass induced by travelling Dirac between two Dirac with equal mass is not constant, but a quadratic function of time $m(t) = 2m(1− \frac{1}{\sqrt{(1+\tau^2})})t(1−t)+m$. Despite with balanced source and target, the optimal measure path first shrinks its total mass during travelling, and then grows back, in order to minimize the WFR cost.
> >
> > To keep the mass conserved during the whole process, it is natural to let the penalization of growth $\delta \to \infty$ to ensure no growth. With balanced source and target, and infinite growth penalization $\delta$, we proved that
> > - The dynamical WFR reduces to dynamical OT.
> > - The OET reduces to static OT. Thus, the two semi-couplings both reduce to static OT coupling.
> > - The conditional path i.e. travelling Dirac reduces to uniform rectilinear motion with conserved mass. Thus, the conditional growth rate $g_t(x|x_0,x_1)$ reduces to 0, and the conditional velocity $u_t(x|x_0,x_1)$ reduces to $x_1-x_0$.
> >
> > The details can be found in **Appendix A.6** of the revised manuscript. The core intuition here is that setting $\delta \to \infty$ forces the growth rate to vanish, in order to have a finite dynamical WFR cost. Without the growth rate g, everything is just the same as balanced OT.
> >
> > Tong’s OT-CFM first finds an OT coupling between source and target, which is the limiting case of OET. Then, OT-CFM use simplified CFM as conditional probability path  $p_t(x|x_0,x_1)=\mathcal{N}(x|tx_1+(1-t)x_0,σ^2I)$, results in a conditional velocity $u_t(x|x_0,x_1)=x_1-x_0$, which is the limiting case of travelling Dirac. Thus, the limiting case of our framework is equivalent to Tong’s OT-CFM.
> >
> >
> > > - Did the authors inspect any theoretical guarantees of their batchwise approach (just asking for an intuition)? How detrimental is it to use this approach for learning the true coupling?
> >
> > Thank you for the question. We provide some discussion and intuition regarding our batchwise approach below, which has also been added to **Appendix D** for more detailed discussion.
> > - The convergence of mini-batch optimal transport to the true coupling remains an open problem. For the balanced case, there are some theoretical and numerical studies investigating the effect of batch size and sample number on convergence [1]. To our knowledge, there is currently no analogous results for the unbalanced setting.
> >
> > - Intuitively, an independent coupling can be seen as the extreme case of mini-batch OT with batch size 1, while full-batch OT represents the other extreme. Selecting an appropriate batch size therefore represents a trade-off between computational efficiency and the fidelity to the optimal transport principle.
> > - Importantly, using mini-batch OT does not affect that WFR-FM still matches the marginals to the data distributions. As shown in batch size ablation experiments (Table 8), the method is robust to batch size choices. Even though this may lead to slight deviations from the globally optimal transport plan, the approach remains valid and practical for learning the underlying dynamics.
> > - In real applications, we in fact use full-batch OT for the vast majority of datasets; mini-batch OT is only applied to the CITE and Mouse datasets due to their large sample sizes.

---

> > > ### Author Response · Authors · 2025-11-27
> > > **Response to Reviewer V2He (4/4)**
> > >
> > > > - Why would unbalanced OT work better on the EMT, EB, and cite datasets? I expect it to work on par with other OT and FM models, but why would it overcome them? Is it related to some aspect of umbalancedness?
> > >
> > > Thank you for this insightful question.
> > > - To better understand why unbalanced OT methods may perform particularly well on datasets such as EMT, EB, and CITE-seq, we examined the degree of unbalancedness in real multi-time single-cell RNA-seq datasets used in our paper (except for EMT, which does not provide cell-type annotations in raw data). Specifically, we computed the proportions of cell types (or clusters) across different time points. Our analysis reveals substantial temporal variation in cell-type abundances, indicating pronounced unbalancedness in these datasets.
> > > - Such imbalance provides a plausible explanation for why approaches based on unbalanced optimal transport can outperform balanced OT or standard flow-matching models that implicitly assume mass preservation. Additionally, existing literature (such as TIGON[2]) has reported effects of cell proliferation on the dynamics in both the EMT and Mouse datasets, further supporting the importance of modeling growth and apoptosis when learning cellular trajectories.
> > > - We have added this discussion, together with supporting empirical evidence, to **Appendix B.10** in the revised manuscript.
> > >
> > >
> > >
> > > **References**
> > >
> > > [1] Sommerfeld M, Schrieber J, Zemel Y, et al. Optimal transport: Fast probabilistic approximation with exact solvers[J]. Journal of Machine Learning Research, 2019, 20(105): 1-23.
> > >
> > > [2] Sha Y, Qiu Y, Zhou P, et al. Reconstructing growth and dynamic trajectories from single-cell transcriptomics data[J]. Nature Machine Intelligence, 2024, 6(1): 25-39.

---

### Official Review · Reviewer_XcwL · 2025-11-03

**Soundness:** 2
**Presentation:** 3
**Contribution:** 3
**Rating:** 6
**Confidence:** 4

**Summary:**

This paper proposed use of Wasserstein-Fisher-Rao (WFR) metric to capture unbalanced transport by jointly regressing both a transport vector field and growth function, while explicitly modeling growth dynamics unlike other unbalanced transport FM methods. The method is evaluated on a range of synthetic datasets as well as commonly used real-world datasets for single-cell biology application.

**Strengths:**

* **Clarity**: The authors provide comprehensive theoretical background in main text and appendix as well as relation to other unbalanced OT related work highlighting key differences between baselines and proposed method. The authors further support theoretical background with extensive proofs in the appendix.
* **Robustness**: Robustness is tested through several ablation studies on batch size and coupling as well as controlled synthetic settings.
* **Low computational cost**: The authors suggest method that improves performance in matching end-point marginals in comparison to baselines while keeping low computational cost. Further authors also show that the computational cost does not significantly increase across different dimensions or batch sizes.

**Weaknesses:**

* **Evaluation of learned growth dynamics**: The paper demonstrates superior performance across $W_1$ and $RME$ metrics, however paper would benefit from showing it also manages to learn correct growth dynamics given it explicitly learns term $g_{\phi}(x,t)$. This should be added as Q5 in experiments section and validated in synthetic or real-world setting.

* **Adding further overview of use of WFR for FM-based algorithms**: It would be good to see more extensive discussion and if possible empirical comparison to unbalanced transport action matching (AM) by Neklyudov (2023) which also uses WFR distance to construct unbalanced OT. It would be good to see trade-off between using WFR to explicitly learn growth dynamics and using a single network to learn both transport and growth dynamics as presented in AM

**Questions:**

* In line 115 and 118, authors cite work by Neklyudov (2023, 2024) [1,2] and Sun (2025) [3] providing commentary that existing unbalanced approaches lack framework that can jointly recover velocity and growth dynamics or require costly ODE simulations. Would it be possible to evaluate the quality of learnt growth dynamics in a synthetic setting and compare to other unbalanced transport baselines such as Neklyudov (2023, 2024) and Sun (2025)?
* Proposition 5.1 states that the solution to the multi-time WFR problem is equivalent to the concatenation of the solutions of consecutive time points. From algorithm 1 it also seems that the shared time-continuous velocity network is trained on consecutive pairs of points? However recent multi-marginal methods such as 3MSBM by Theodoropoulos et al (2025) [4] jointly enforce all marginals achieving better global coherence. Does proposition 5.1 then still hold?

**References**

[1] Neklyudov, Kirill, et al. "Action matching: Learning stochastic dynamics from samples." International conference on machine learning. PMLR, 2023.

[2] Neklyudov, Kirill, et al. "A computational framework for solving wasserstein lagrangian flows." arXiv preprint arXiv:2310.10649 (2023).

[3] Sun, Yuhao, et al. "Variational Regularized Unbalanced Optimal Transport: Single Network, Least Action." arXiv preprint arXiv:2505.11823 (2025).

[4] Theodoropoulos, Panagiotis, et al. "Momentum Multi-Marginal Schr\" odinger Bridge Matching." arXiv preprint arXiv:2506.10168 (2025).

---

> ### Author Response · Authors · 2025-11-27
> **Response to Reviewer XcwL (1/2)**
>
> > **Summary**:
>
> > This paper proposed use of Wasserstein-Fisher-Rao (WFR) metric to capture unbalanced transport by jointly regressing both a transport vector field and growth function, while explicitly modeling growth dynamics unlike other unbalanced transport FM methods. The method is evaluated on a range of synthetic datasets as well as commonly used real-world datasets for single-cell biology application.
>
> > **Strengths**:
>
> > - **Clarity**: The authors provide comprehensive theoretical background in main text and appendix as well as relation to other unbalanced OT related work highlighting key differences between baselines and proposed method. The authors further support theoretical background with extensive proofs in the appendix.
>
> > - **Robustness**: Robustness is tested through several ablation studies on batch size and coupling as well as controlled synthetic settings.
>
> > - **Low computational cost**: The authors suggest method that improves performance in matching end-point marginals in comparison to baselines while keeping low computational cost. Further authors also show that the computational cost does not significantly increase across different dimensions or batch sizes.
>
> - Thank you very much for your careful reading and for the positive feedback on our work. We appreciate your clear and thoughtful summary of our paper. As you noted, our approach leverages the Wasserstein-Fisher-Rao (WFR) metric to jointly model transport and growth dynamics, which allows us to capture unbalanced transport in a way that distinguishes our method from other approaches in the literature. In addition, flow matching, a key component of our approach, plays an important role in enabling us to model these dynamics in a flexible and scalable manner. We are pleased that you found our method’s evaluation on both synthetic and real-world datasets, particularly in single-cell biology applications, to be compelling. Your recognition of the theoretical clarity, robustness, and efficiency of our approach further reinforces the value of our work, and we are grateful for your thoughtful acknowledgment of these aspects. Finally, we would like to acknowledge that methods such as Action Matching (AM) and Wasserstein Lagrangian Flow (WLF) are highly influential and provide elegant and powerful formulations for learning dynamics. We therefore appreciate your noting the distinctions and connections between these approaches and our proposed method.
>
> - Below you will find our detailed point-by-point responses. We have revised our manuscript accordingly to address the comments, and all changes made to the manuscript are indicated in **blue** throughout the revised main text and appendix. Thank you once again for your insightful review and encouraging feedback.
>
> > Weaknesses:
> > - **Evaluation of learned growth dynamics**: The paper demonstrates superior performance across $W_1$ and $RME$ metrics, however paper would benefit from showing it also manages to learn correct growth dynamics given it explicitly learns term $g_{\phi}(x,t)$. This should be added as Q5 in experiments section and validated in synthetic or real-world setting.
>
> - Thanks for your insightful suggestion. We agree that validating the learnt growth rate is essential. As the ground truth growth rate is typically unavailable in real-world single-cell data, we performed this validation using the Simulation Gene dataset, where the dynamics are governed by known stochastic differential equations. We calculated the Pearson correlation coefficient ($g_{\text{corr}}$) between the predicted growth rate $g_{\phi}(x,t)$ and the ground truth growth rate on the data points from the dataset. The results are summarized in the table below:
>
> | Method | Correlation ($g_{\text{corr}}$) |
> | :--- | :--- |
> | **WFR-FM (Ours)** | **0.9913** |
> | DeepRUOT | 0.9688 |
> | TIGON | 0.9705 |
> | Var-RUOT | 0.9214 |
> | AM |0.5851  |
>
> - Action Matching and Wasserstein Lagrangian Flow are both simulation-free methods for solving Unbalanced Optimal Transport. Both approaches involve fitting an action, from which the velocity field and growth field are subsequently derived. To handle multi-timepoint biological data, we re-implemented Action Matching following the original paper.
>
> - As shown in the table (Table 4 in the revised manuscript), WFR-FM achieves the highest correlation (0.9913). This indicates that our method allows for an accurate recovery of the underlying growth rate. We have added these results as **Q5** in the experiment section of the revised manuscript.

---

> > ### Author Response · Authors · 2025-11-27
> > **Response to Reviewer XcwL (2/2)**
> >
> > > - **Adding further overview of use of WFR for FM-based algorithms**: It would be good to see more extensive discussion and if possible empirical comparison to unbalanced transport action matching (AM) by Neklyudov (2023) which also uses WFR distance to construct unbalanced OT. It would be good to see trade-off between using WFR to explicitly learn growth dynamics and using a single network to learn both transport and growth dynamics as presented in AM.
> >
> > We thank the reviewer for this suggestion. We agree that Action Matching (AM) and Wasserstein Lagrangian Flows (WLF) are indeed innovative simulation-free trajectory inference methods, distinct from flow matching, and that both have been extended to handle unbalanced dynamics. As these works are closely related to ours, we have added a more detailed discussion of AM and WLF in the **Introduction** to highlight their relevance and contextualize our contributions.
> >
> >
> > > **Questions**:
> >
> > > - In line 115 and 118, authors cite work by Neklyudov (2023, 2024) [1,2] and Sun (2025) [3] providing commentary that existing unbalanced approaches lack framework that can jointly recover velocity and growth dynamics or require costly ODE simulations. Would it be possible to evaluate the quality of learnt growth dynamics in a synthetic setting and compare to other unbalanced transport baselines such as Neklyudov (2023, 2024) and Sun (2025)?
> >
> > Thanks for your insightful suggestion. We have calculated the Pearson correlation coefficient ($g_{\text{corr}}$) between the predicted growth rate $g_{\phi}(x,t)$ and the ground truth growth rate on the data points from the dataset, and compared against other baselines including Action Matching and Var-RUOT. The results are summarized in our response to Weakness 1 (Table 4 in the revised manuscript). WFR-FM achieves the highest correlation (0.9913). This indicates that our method allows for an accurate recovery of the underlying growth rate. We have added these results as **Q5** in the experiment section of the revised manuscript.
> >
> > > - Proposition 5.1 states that the solution to the multi-time WFR problem is equivalent to the concatenation of the solutions of consecutive time points. From algorithm 1 it also seems that the shared time-continuous velocity network is trained on consecutive pairs of points? However, recent multi-marginal methods such as 3MSBM by Theodoropoulos et al (2025) [4] jointly enforce all marginals achieving better global coherence. Does proposition 5.1 then still hold?
> >
> > - Thank you for your insightful comment and for bringing up the recent multi-marginal methods like 3MSBM by Theodoropoulos et al. (2025). We agree that these methods aim to jointly enforce all marginals, which can result in better global coherence. This is indeed an important direction, and it is something we plan to explore in future work.
> > - As you correctly pointed out, Proposition 5.1 assumes that the solution to the multi-time WFR problem can be obtained by concatenating the solutions of successive time points. This holds true under the assumption that the transport between consecutive time points (t_{k-1}, t_{k}) is independent and does not influence other time intervals. The method is based on minimizing the total action over the entire time span, and in the case of independent pairs, minimizing the action for each pair is equivalent to minimizing the total action.
> > - However, when considering multi-marginal cases, the situation changes. Multi-marginal methods, like 3MSBM, aim to enforce smoothness or continuity across all marginals at the observed time points. This introduces dependencies between different adjacent time pairs, meaning that each consecutive pair is no longer independent of the others. Therefore, under a multi-marginal framework, Proposition 5.1 would no longer hold in the same form, as the independence assumption is no longer valid.
> > We have added a remark to clarify its relationship with multi-marginal methods and to explain why the proposition does not hold under a multi-marginal framework at the proof of Proposition 5.1 in the **Appendix A.7**.
> >
> >
> > > **References**
> >
> > > [1] Neklyudov, Kirill, et al. "Action matching: Learning stochastic dynamics from samples." International conference on machine learning. PMLR, 2023.
> >
> > > [2] Neklyudov, Kirill, et al. "A computational framework for solving wasserstein lagrangian flows." arXiv preprint arXiv:2310.10649 (2023).
> >
> > > [3] Sun, Yuhao, et al. "Variational Regularized Unbalanced Optimal Transport: Single Network, Least Action." arXiv preprint arXiv:2505.11823 (2025).
> > [4] Theodoropoulos, Panagiotis, et al. "Momentum Multi-Marginal Schr" odinger Bridge Matching." arXiv preprint arXiv:2506.10168 (2025).

---

### Author Response · Authors · 2025-11-21
**Quick and Initial Response to Reviewers**

We sincerely thank all reviewers for their careful reading of our manuscript and for providing constructive feedback and valuable suggestions. We are grateful for the positive comments regarding the novelty, clarity, theoretical grounding, and empirical validation of our work, and we fully agree with the reviewers on the aspects that can be further strengthened.

Over the past few days we have carefully evaluated each point raised by the reviewers and have begun implementing substantial revisions. Below we outline the key updates we are undertaking; most of these are already in progress or partially completed.

1. **Evaluation of Learned Growth Dynamics**: We will add experiments as Q5 in the results section to explicitly evaluate the quality of the learned growth dynamics. Particularly, this will include more thorough comparisons and discussions with state-of-the-art unbalanced transport methods such as Action Matching (AM) and Var-RUOT, which are highly relevant for their strong theoretical foundations and performance in modeling unbalanced stochastic dynamics. We will also expand the accompanying discussion to situate WFR-FM within the broader methodological landscape.

2. **Discussion of Multi-Marginal Methods**: We will include a detailed discussion of the relationship between our approach and recent multi-marginal methods, highlighting both connections and key differences.

3. **Figure 1 and Caption Update**: Figure 1 will be redrawn to offer a clearer and more intuitive overview of the WFR-FM workflow. The caption will also be revised to better explain the components, inputs, and outputs involved in the framework.

4. **Mini-Batch OT Discussion**:  We will add a more explicit discussion of mini-batch OT, along with ablations evaluating batch-size sensitivity in WFR-FM. This addresses the reviewers’ question about training stability and potential bias introduced by mini-batching.

5. **Real Biological Data and Unbalanced Dynamics**: We will add a discussion on the widespread presence of unbalance effects in real biological datasets, providing a more detailed rationale why WFR-FM performs better than balanced OT methods in these contexts.

6. **Comparison to VGFM**: We will provide a more thorough and prominent discussion of VGFM into the main text and present a clearer, more direct discussion. We fully agree that VGFM represents an important advance in the joint modeling of velocity and growth, and it has opened an important line of research on unbalanced flow matching that our work builds upon. In the revised main text, we will explicitly acknowledge how WFR-FM is developed within this intellectual trajectory, and also detailing the conceptual, theoretical, and algorithmic distinctions that characterize our approach, highlighting how WFR-FM contributes unique capabilities within this emerging research direction.

7. **Baseline Hyperparameter Tuning**: We will provide a detailed explanation of the hyperparameter tuning procedures for all baseline methods to ensure fair and robust comparisons.

We appreciate the reviewers’ detailed comments, many of which have already led us to improve both the clarity and rigor of the manuscript. We are finalizing the new experiments and analyses, and we will soon provide a complete, point-by-point response together with the updated manuscript. In the meantime, we welcome any additional suggestions from the reviewers and will do our best to incorporate them in the final revision.

---

### Meta-Review · Area_Chair_ebqt · 2026-01-04

**Summary:**

The reviewers unanimously advocate acceptance of the paper, with one reviewer championing it. All reviewers highlight the importance of the work for the field and the novelty of the proposed methodology. The raised concerns are mostly minor, requiring either conceptual or empirical comparisons with relevant works and suggesting improvements to the presentation and empirical evaluation.

**Reviewer Concerns:**

The authors addressed all of the concerns during the rebuttal, except for a few minor ones, e.g., the comparison against WLF in the table provided in the response to XcwL (https://openreview.net/forum?id=1nqu7bK1mm&noteId=lRB2YrClN0).

**Reviewer Scores:**

The paper was initially evaluated positively with no major concerns raised. Thus, the rebuttal would not result in a significant change in score except for a slight score increase.

---

### Decision · Program_Chairs · 2026-01-26

Accept (Poster)